# Quantifying CO emissions from boreal wildfires by assimilating TROPOMI and TCCON observations

Sina Voshtani<sup>1</sup>, Dylan B. A. Jones<sup>1</sup>, Debra Wunch<sup>1</sup>, Drew C. Pendergrass<sup>2</sup>, Paul O. Wennberg<sup>3,4</sup>, David F. Pollard<sup>5</sup>, Isamu Morino<sup>6</sup>, Hirofumi Ohyama<sup>6</sup>, Nicholas M. Deutscher<sup>7</sup>, Frank Hase<sup>8</sup>, Ralf Sussmann<sup>9</sup>, Damien Weidmann<sup>10</sup>, Rigel Kivi<sup>11</sup>, Omaira García<sup>12</sup>, Yao Té<sup>13</sup>, Jack Chen<sup>14</sup>, Kerry Anderson<sup>14,15</sup>, Robin Stevens<sup>16,17,18</sup>, Shobha Kondragunta<sup>19</sup>, Aihua Zhu<sup>20</sup>, Douglas Worthy<sup>21</sup>, Senen Racki<sup>22</sup>, Kathryn McKain<sup>23</sup>, Maria V. Makarova<sup>24</sup>, Nicholas Jones<sup>7</sup>, Emmanuel Mahieu<sup>25</sup>, Andrea Cadena-Caicedo<sup>26</sup>, Paolo Cristofanelli<sup>27</sup>, Casper Labuschagne<sup>28</sup>, Elena Kozlova<sup>29</sup>, Thomas Seitz<sup>30</sup>, Martin Steinbacher<sup>30</sup>, Reza Mahdi<sup>31</sup>, and Isao Murata<sup>32</sup>


<sup>1</sup>Department of Physics, University of Toronto, Toronto, Ontario, Canada

<sup>2</sup>School of Engineering and Applied Sciences, Harvard University, Cambridge, Massachusetts, USA

<sup>3</sup>Division of Geological and Planetary Sciences, California Institute of Technology, Pasadena, CA, USA

<sup>4</sup>Division of Engineering and Applied Science, California Institute of Technology, Pasadena, CA, USA

New Zealand Institute for Earth Science Limited, Lauder, NZ

<sup>6</sup>National Institute for Environmental Studies (NIES), Onogawa 16-2, Tsukuba, Ibaraki, Japan

<sup>7</sup>Centre for Atmospheric Chemistry, School of Earth, Atmospheric and Life Sciences, University of Wollongong, Wollongong, Australia

<sup>8</sup>Institute of Meteorology and Climate Research (IMK-ASF), Karlsruhe Institute of Technology (KIT), Karlsruhe, Germany

<sup>9</sup>Karlsruhe Institute of Technology (KIT), IMK-IFU, Garmisch-Partenkirchen, Germany

<sup>10</sup>Space Science and Technology Department, STFC Rutherford Appleton Laboratory, Didcot, OX11 0QX, UK

<sup>11</sup>Space and Earth Observation Centre, Finnish Meteorological Institute, Sodankylä, Finland

<sup>12</sup>Izaña Atmospheric Research Centre (IARC), State Meteorological Agency of Spain (AEMET), Santa Cruz de Tenerife, Spain

<sup>13</sup> Sorbonne Université, CNRS, MONARIS, UMR8233, F-75005 Paris, France

<sup>14</sup>Air Quality Research Division, Environment and Climate Change Canada, Toronto, Ontario, Canada

<sup>15</sup>Natural Resources Canada (emeritus)

<sup>16</sup>Department of Chemistry, Faculty of Arts and Sciences, Université de Montréal, Montréal, Quebec, Canada

<sup>17</sup>School of Public Health, Université de Montréal, Montréal, Quebec, Canada

<sup>18</sup>Climate Research Division, Environment and Climate Change Canada, Victoria, British Columbia, Canada

<sup>19</sup>Center for Satellite Application and Research, NOAA/NESDIS, College Park, USA

<sup>20</sup>IM Systems Group Inc., College Park, USA

<sup>21</sup>Climate Research Division, Environment and Climate Change Canada, Toronto, Ontario, Canada

<sup>22</sup>Climate Chemistry Measurements and Research, Environment and Climate Change Canada, Toronto, Ontario, Canada

<sup>23</sup>NOAA Global Monitoring Laboratory, Boulder, CO, USA

<sup>24</sup>Department of Atmospheric Physics, Faculty of Physics, St. Petersburg State University, Saint Petersburg, Russia

<sup>25</sup>Institute of Astrophysics and Geophysics, University of Liège, Liège 4000, Belgium

<sup>26</sup>Instituto de Ciencias de la Atmósfera y Cambio Climático Universidad Nacional Autonoma de Mexico, UNAM, Mexico City, Mexico

<sup>27</sup>National Research Council of Italy, Institute of Atmospheric Sciences and Climate (ISAC), Bologna, Italy

<sup>28</sup>South African Weather Service, Stellenbosch, South Africa

<sup>29</sup>University of Exeter, Hatherly Labs, Exeter, EX4 4PS, UK

<sup>30</sup>Empa, Swiss Federal Laboratories for Materials Science and Technology, CH-8600 Duebendorf, Switzerland

<sup>31</sup>GAW Bukit Kototabang Station, BMKG, Indonesia

<sup>32</sup>Graduate School of Environmental Studies, Tohoku University, Sendai, Japan

Correspondence to: Sina Voshtani (voshtani.sina@gmail.com; sina.voshtani@utoronto.ca)

**Abstract.** We perform a global inverse modelling analysis to quantify biomass burning emissions of carbon monoxide (CO)

from the extreme wildfires in Canada between May and September 2023. Using the GEOS-Chem model, we assimilated

observations at 3-day temporal and  $2^{\circ} \times 2.5^{\circ}$  horizontal resolution from the Tropospheric Monitoring Instrument (TROPOMI) separately and then jointly with Total Carbon Column Observing Network (TCCON) measurements. We also evaluated prior emissions from the Quick Fire Emissions Dataset (QFED), Blended Global Biomass Burning Emissions Product eXtended (GBBEPx), Global Fire Assimilation System (GFAS), and Canadian Forest Fire Emissions Prediction System (CFFEPS). The assimilation of TROPOMI-only measurements estimated posterior North America emissions for QFED, GBBEPx, GFAS, and CFFEPS of 110.4 $\pm$ 20, 112.8 $\pm$ 20, 127.2 $\pm$ 17, and 125.6 $\pm$ 18 Tg CO compared to prior estimates of 37.1, 42.7, 91.0, and 90.2 Tg CO, respectively. The joint assimilation of TROPOMI+TCCON reduced the posterior 1 $\sigma$  uncertainty on the North American emission estimates by up to about 30%, while showing only a modest impact (< 5%) on the mean estimate of the inferred emissions. An evaluation against independent measurements reveals that adding TCCON data increases the correlations and slightly lowers the biases and standard deviations. Additionally, including an experimental TCCON product at East Trout Lake with higher surface sensitivity, we find better agreement of the assimilation results with nearby in situ tall tower and aircraft measurements. This highlights the potential importance of vertical sensitivity in these experimental data for constraining local surface emissions. Our results demonstrate the complementarity of the greater temporal coverage provided by TCCON with the spatial coverage of TROPOMI when these data are jointly assimilated.

#### 60 1 Introduction





Biomass burning (BB) from wildfires is a major source of carbon emissions released into the atmosphere with large climate and air quality impacts, exerting a significant influence on human health, ecosystems, and the environment (Cascio, 2018; Chen et al., 2017; O'Neill et al., 2021; Wu et al., 2022). Over the past few years, wildfires have become more frequent and destructive in different regions of the world (Jegasothy et al., 2023; Mataveli et al., 2024; You and Xu, 2023). More specifically, in Canada in 2023, the total area burned by wildfires surpassed the previous record in 1989 (75,596 km<sup>2</sup>) by nearly a factor of 2.5 while the amount of emitted carbon also dramatically increased by more than 11 times compared to the 1998-2022 average (Jain et al., 2024; Jones et al., 2024; Kolden et al., 2024). Therefore, reactive trace gases (e.g., CO) and greenhouse gases (GHGs) (e.g., CH<sub>4</sub> and CO<sub>2</sub>) had an unprecedent increase during the 2023 wildfire emissions (Byrne et al., 2024). In addition to perturbing the carbon budget, these emissions also have implications for air quality. However, obtaining reliable estimates of wildfire emissions is a challenging task due to several factors, such as the episodic and localized nature of these emissions (Sokolik et al., 2019; Zhao et al., 2025). Here, we focus on estimating emissions from carbon monoxide (CO) from wildfires during the summer 2023. With a lifetime of up to several months (Holloway et al., 2000), which is sufficiently long to track long-range transport on intercontinental scales, CO is an ideal tracer of combustion. CO plays an important role in both air quality and climate as it is a precursor of ozone (O<sub>3</sub>) and the dominant sink of the hydroxyl radical (OH), which is the main atmospheric oxidant (Aschi and Largo, 2003; Fowler et al., 2008). Observations of CO have provided information on combustion sources on a range of scales, from urban to regional and global scales (Borsdorff et al., 2020; Cristofanelli et al., 2024; Pommier et al., 2013; Schneising et al., 2020; Tang et al., 2019).

CO emissions from wildfires can be estimated either from bottom-up or top-down approaches. In the bottom-up approach, emissions are represented as the product of an emission factor, which is the amount of trace gas emitted per unit of fuel consumed, and the amount of dry matter burned. Bottom-up inventories use either observations of burned area to determine the mass of dry matter burned (Liu et al., 2024; van der Werf et al., 2017; Wiedinmyer et al., 2023) or estimates of fire radiative power (FRP) to quantify the rate of fuel consumption (Filizzola et al., 2023; Kaiser et al., 2012). There are typically large uncertainties in these inventories (Andreae, 2019; Hundal et al., 2024) arising from discrepancies in the emission factors and the estimated mass of dry matter burned, resulting in significant differences in emission estimates (Chen et al., 2022; Nguyen et al., 2023; Saikawa et al., 2017; Zhang et al., 2023). The top-down approach makes use of CO observations to optimize emissions through an inverse modelling method, but this approach depends on the use of an atmospheric chemistry-transport model and a priori emission estimates, which are typically obtained from a bottom-up inventory.








Over the past two decades, global CO observations have been provided by several satellite sensors, including the Measurements of Pollution in the Troposphere (MOPITT) instrument (Deeter et al., 2003; Edwards et al., 2006), the Infrared Atmospheric Sounding Interferometer (IASI) (Pope et al., 2021; Turquety et al., 2004), and the Tropospheric Emission Spectrometer (TES) (Lopez et al., 2008), and these data have been used in numerous inverse modelling studies to quantify CO emissions (e.g., Kasibhatla et al., 2002; Arellano Jr. et al., 2006; Warner et al., 2007; Jones et al., 2009; Kopacz et al., 2010; Miyazaki et al., 2012; Miyazaki, Eskes, and Sudo 2015; Jiang et al., 2017; Zheng et al., 2019). However, large discrepancies between the inversion results have been reported, which may arise from differences between spatiotemporal coverage of the observations, the vertical sensitivity of the measurements, and observation biases (Deeter et al., 2015; Jiang et al., 2017; Jones et al., 2009; Miyazaki et al., 2015; Warner et al., 2010). Nonetheless, a few recent studies attempted to address some of the challenges by reducing potential biases in the model (Gaubert et al., 2023; Miyazaki et al., 2020) or by improving the quality of the assimilated data (Tang et al., 2024). The Tropospheric Monitoring Instrument (TROPOMI) (Borsdorff et al., 2018), launched in 2017, has provided CO retrievals with improved accuracy, higher spatial resolution, and significantly greater observational coverage (Landgraf et al., 2016; Schneising et al., 2020). In particular, it offers higher sensitivity to near-surface CO compared to earlier thermal infrared measurements from instruments such as IASI and TES. These factors make it wellsuited for inverse modelling of CO emissions, as demonstrated in many recent studies (Borsdorff et al., 2023; Byrne et al., 2024; Goudar et al., 2023; Griffin et al., 2024; Inness et al., 2022; Shahrokhi et al., 2023; Stockwell et al., 2022; Wan et al., 2023).

Measurements from surface in situ networks and aircraft campaigns have been used for CO trend determination (Patel et al., 2024) and CO inversion studies in the past (Palmer et al., 2003; Yumimoto & Uno, 2006; Koohkan & Bocquet, 2012; Tang et al., 2013; Feng et al., 2020). However, mainly due to limited spatiotemporal coverage and/or vertical distribution, they are typically incapable of sufficiently constraining emission estimates on fine spatial scales; therefore, model errors such as those from vertical transport, the OH field, and the a priori emissions, can significantly impact the inferred emission estimates (Hooghiemstra et al., 2011). Only a few studies have attempted to use both satellite and surface observations together to exploit the complementarity of these observations to reduce the influence of errors, such as those that arise from the sensitivity to the

vertical distribution of CO (Tang et al., 2022) and long-range transport (Kim et al., 2024). However, most of these studies focused on inversions over a limited area, where sufficient surface CO observations are available. There are also ground-based total column measurements from networks such as the Total Carbon Column Observing Network (TCCON, Wunch et al. 2011), which was designed in part to validate satellite observations (Borsdorff et al., 2019; Bukosa et al., 2023; Hedelius et al., 2021; Sha et al., 2021; Tang et al., 2024). TCCON provides time-resolved and accurate column-averaged dry-air mole fractions of CO (XCO) under sunny skies. Although TCCON observations are spatially sparse, they are of high temporal density and therefore could provide valuable information in constraining episodic CO emissions from wildfires. However, a standard method of integrating TCCON measurements with satellite data in a data assimilation or inversion system is still lacking, as most current studies assimilate satellite data, while reserving the TCCON data to evaluate the performance of the assimilation.

In this study, we quantify biomass burning CO emissions between May and September 2023 using the CHemistry and Emissions REanalysis Interface with Observations (CHEEREIO) assimilation toolkit (Pendergrass et al., 2023), which employs the GEOS-Chem model and an ensemble Kalman filter (EnKF) scheme. We conduct a global analysis, but our focus is on quantifying boreal emissions associated with the 2023 fires in Canada. We jointly assimilate TCCON and TROPOMI data and conduct a comparison with a TROPOMI-only assimilation to assess the added value of TCCON observations in the assimilation and to determine the additional constraints that TCCON data provide for optimizing CO emissions from localized and episodic wildfires. We include two distinct types of TCCON data with different vertical sensitivities in our inversion, while using independent total column and in situ surface and aircraft vertical profile observations to characterize the success of our analysis. Additionally, we evaluate the following three global and one regional biomass burning inventories in the context of the assimilation: the Quick Fire Emissions Dataset (QFED) (Koster et al., 2015), the Blended Global Biomass Burning Emissions Product eXtended (GBBEPx) (Zhang et al. 2012; 2019), the Global Fire Assimilation System (CFFEPS) (Chen et al., 2019).

We begin in Section 2 with a description of the observations and model configurations, including the a priori emissions used in the inversion. Section 3 presents the inversion methods, the main assumptions, and sensitivity experiments using simulated observations to tune the inversion performance. Section 4 provides the main results and discussion, and finally, the study concludes with a few summary points and a suggestion for future works.

#### 2 Observations and model







This section describes the observational datasets and modelling framework and inputs used in this study. We first describe in Sections 2.1-2.2 the two types of observations that are assimilated in the inversion framework: (i) TROPOMI satellite CO products and (ii) TCCON ground-based CO measurements. Then, in Section 2.3, we present several independent datasets (not assimilated) used for evaluation, including the Network for the Detection of Atmospheric Composition (NDACC,

De Mazière et al., 2018) ground-based total column observations, in situ surface CO measurements from the World Data Centre for Greenhouse Gases (WDCGG) network and from Environment and Climate Change Canada's (ECCC) tall tower at East Trout Lake (ETL), and vertical profiles from in situ aircraft measurements. The assimilated TCCON data are also used for comparisons between different experiments, although they are no longer independent information for the joint inversion. Finally, Section 2.4 provides a description of the GEOS-Chem model and emissions inventories that are used as the a priori estimate in our inversion setup. The priors include three global inventories, including QFED, GBBEPx, and GFAS, and one regional inventory, CFFEPS, for North America.

# 2.1 TROPOMI








The TROPOMI instrument is on board the Copernicus Sentinel 5 Precursor (S5P) satellite, which was launched in October 2017. Total column abundances of carbon monoxide (XCO) are retrieved from spectra measured in the shortwave infrared (SWIR) band at 2305-2385 nm, with daily global coverage, a local overpass solar time of 13:30 UTC, and high spatial resolution of 5.5 × 7 km<sup>2</sup> (Veefkind et al., 2012). We use the operational XCO product publicly available from the European Space Agency (ESA) Sentinel-5P data hub at https://scihub.copernicus.eu/ (last access: 4 June 2024) (Landgraf, 2019), with a reported bias of better than 15 ppb in comparison with TCCON GGG2014 data product (Sha et al., 2021). The XCO data are published together with the total column averaging kernels to account for the sensitivity of the retrieved total column to the true atmosphere, thus, they can be used along with a priori vertical profiles to obtain model-equivalent total CO columns representing the observed data (Apituley et al., 2018). The TROPOMI retrieval algorithm provides clear-sky and cloudy observations over land and ocean (Borsdorff et al., 2019), however, we only use measurements with a quality flag equal to and greater than 0.7 to ensure high-quality data obtained under cloud-free or low cloud conditions. As shown in the Fig. 1a, we exclude TROPOMI observations poleward of 60°, primarily to avoid biases due to low surface albedo in the SWIR from snow cover (Hasekamp et al., 2022; Lorente et al., 2021) and biases due to the stratosphere in the chemical transport model (CTM) affecting the inversion performance (Turner et al., 2015). Since the horizontal resolution of the TROPOMI data is substantially higher than the GEOS-Chem model resolution used in this study  $(2^{\circ} \times 2.5^{\circ})$ , the observations are not spatially representative for the model grid cells, resulting in a large representativeness error in the assimilation process. To overcome this, we aggregate the observations into so-called super-observations before using them in the assimilation. In fact, for the duration of the study between May and September 2023, we compute the error-weighted median average of measurements within each grid cell, where each measurement is weighted by the inverse of its reported error standard deviation (Eskes et al., 2003; Miyazaki et al., 2012). To account for error reduction because of averaging, we follow a similar method as Pendergrass et al. (2023) to compute the associated super-observation errors. This includes average of individual measurement errors and assumptions on error correlations and transport errors. The relative weight of the super-observation error to the prior error is then estimated through parameter tuning in the inversion system to ensure robustness to possible error misspecification (see Section 3 and Appendices A and B). Finally, a total of 1,744,682 number of observations are processed.

Several previous studies evaluated TROPOMI XCO observations and found reasonable agreement with satellite and ground-based measurements. For example, Sha et al. (2021) reported a bias of  $2.45 \pm 3.38\%$  against the unscaled TCCON and a bias of  $6.50 \pm 3.45\%$  against NDACC, which remains within the range of TROPOMI's precision and accuracy. In addition, the TROPOMI validation report (https://mpc-vdaf.tropomi.eu/, last access: 28 June 2024; Lambert et al., 2024) shows that operational TROPOMI XCO data are in good agreement with collocated measurements from NDACC, TCCON, and the Collaborative Carbon Column Observing Network (COCCON; Alberti et al., 2022; Frey et al., 2019) monitoring networks.

# 2.2 TCCON







TCCON (https://www.tccon.caltech.edu/) is a ground-based network of solar-viewing Fourier transform spectrometers (FTS) that collect atmospheric transmission spectra every 2-3 minutes. The spectra range covers the near and short-wave infrared region, and measurements collected under clear-sky conditions are used to retrieve column-averaged dry-air mole fractions of trace gases, including carbon monoxide (i.e., XCO) (Wunch et al., 2011). We use data from 15 sites around the world, shown in Fig. 1 and listed in Table 1, derived from the standard GGG2020 retrieval software (Laughner et al., 2024). The difference between the GGG2020 and GGG2014 XCO retrieval data is 6.3 ppb, with GGG2020 larger than GGG2014. This is because TCCON XCO data are no longer scaled to the WMO trace gas scale (Wunch et al., 2025). The accuracy and precision of the standard TCCON XCO product is reported to be around 8 ppb. These data are publicly available and can be accessed via https://tccondata.org/ (last access: 1 June 2024). In addition to the standard XCO retrievals, which use spectra measured using an InGaAs detector and a CO window centred at 4290 cm<sup>-1</sup>, we use retrievals of XCO available from spectra collected at the East Trout Lake TCCON station from an additional InSb detector. The spectral range of the InSb detector includes two mid-infrared windows (centred at 2111 cm<sup>-1</sup> and 2160 cm<sup>-1</sup>) that contain strong CO absorption features that result in an XCO retrieval with markedly different averaging kernels (orange) from the standard XCO retrieval (blue), as shown in Fig. 1c-d. These mid-infrared spectral windows were used in a previous study together with the standard TCCON CO window to extract vertical information from the TCCON measurements (Parker et al., 2023). The XCO retrievals from the InSb spectra have higher sensitivity to the surface and lower sensitivity to the higher altitudes than the standard XCO retrievals. The impact of including these mid-infrared XCO retrievals on the inversion performance to constrain CO emissions is discussed in Section 4.2.2.

We filter all TCCON datasets to include only data with a quality flag = 0 (i.e., the highest quality data). To prepare for the assimilation, first, all the measurements are aggregated in time, based on the model output hourly timestep, weighted by the measurement reported errors. This produced 213,784 quality-controlled data points that were then mapped on the GEOS-Chem grid resolution, providing 19,733 median hourly averaged observations for the period of May–September 2023 in this study.

Figure 1: (a) Number of quality-controlled and aggregated TROPOMI and TCCON observations (XCO) used in the global inversion as described in Sections 2.1 and 2.2. (b) Variability of the number of non-aggregated TROPOMI (blue) and TCCON (red) observations at the East Trout Lake (ETL) TCCON station from April-September 2023. (c) Time series (MM-DD) of XCO column retrievals from standard TCCON GGG2020 data (blue) and XCO measurements from the InSb detector (orange) at ETL between May-September 2023. (d) Column averaging kernels for standard TCCON GGG2020 XCO (blue) and for XCO measurements derived from an alternative CO absorption window on the ETL InSb detector (orange) between May-September 2023.

# 2.3 NDACC and in situ surface and aircraft data


This study employs two independent ground-based data sources for validation purpose. We utilize measurements from NDACC, a global network of ground-based stations equipped with Fourier transform infrared (FTIR) spectrometers that provide long-term total column measurements of XCO (De Mazière et al., 2018). NDACC XCO measurements, similar to those from TCCON (i.e., same spectral domain as the InSb TCCON data), are of high-quality and well-suited for validating models, satellite observations, and assimilation system performance (Kerzenmacher et al., 2012; Lutsch et al., 2020; Sha et al., 2021). In this study, we include mid-infrared NDACC total column data from seven stations covering the study period (see Table 1). The data are publicly available at http://www.ndacc.org (last access: 30 July 2024).

Additionally, continuous and discrete surface in situ CO measurements obtained from WDCGG serve as a second independent dataset to evaluate surface CO concentrations obtained from our experiments. In situ measurements compiled by the WDCGG have been widely used in previous inverse modelling studies for validating results and testing model performance

(Chevallier et al., 2011; Jiang et al., 2017; Miyazaki et al., 2020; Tanimoto et al., 2008). We use the archived data from nine sites over the study period, which are publicly available and accessed from https://gaw.kishou.go.jp/ (last access: 1 July 2024). We also use in situ tall tower measurements from the ETL site provided by ECCC (Chen et al., 2014) to assess the impact of using the XCO retrievals from the InSb spectra on the inversion results. The evaluation results are presented in Section 4.2.2.

We use in situ aircraft CO measurements from the National Oceanic and Atmospheric Administration (NOAA) air sampling network (McKain et al., 2024) taken as another independent source to evaluate our inversion results. The data product is freely available to public via https://gml.noaa.gov/ccgg/aircraft/. The aircraft program aims to capture temporal variability (i.e., seasonal and interannual changes) of the greenhouse gases in the lower atmosphere. Dry air mole fractions of CO are measured using flask air samples at different fixed altitude levels. It provides measurements at different sites across the United States and Canada and at different altitudes, descending from a maximum of 8000 m to the lowest sampling level at ~750 m (a.s.l). These data have been commonly used in previous studies to explore the large-scale changes in horizontal and vertical distribution of CO and greenhouse gases (Sweeney et al., 2015), to serve as benchmark for validating forward and inverse modelling analysis (Stephens et al., 2007; Yang et al., 2007), and to calibrate remote sensing retrievals (Wunch et al., 2010). Focusing on the impact of the experimental TCCON InSb data used in the inversion to constrain surface CO emissions, we use aircraft profiles at ETL during multiple time events (details are discussed in Section 4.2.2). Table 1 shows the list and geographical information of all observations used for evaluation.

| Measurement | Site (ID)                                            | Latitude | Longitude | Altitude<br>(km a.s.l) | Reference                                        |  |  |
|-------------|------------------------------------------------------|----------|-----------|------------------------|--------------------------------------------------|--|--|
| TCCON       | Sodankylä, Finland                                   | 67.4° N  | 26.6° E   | 0.188                  | (Kivi et al., 2022)                              |  |  |
| TCCON       | Karlsruhe, Germany                                   | 49.1° N  | 8.4° E    | 0.116                  | (Hase et al., 2023)                              |  |  |
| TCCON       | Garmisch, Germany                                    | 47.4° N  | 11.1° E   | 0.740                  | (Sussmann and Rettinger, 2023)                   |  |  |
| TCCON       | Paris, France                                        | 48.8° N  | 2.4° E    | 0.060                  | (Té et al., 2022)                                |  |  |
| TCCON       | Harwell, UK                                          | 51.6° N  | 1.3° W    | 0.142                  | (Weidmann et al., 2023)                          |  |  |
| TCCON       | Izana, Tenerife, Spain                               | 28.3° N  | 16.5° W   | 2.370                  | (García et al., 2022)                            |  |  |
| TCCON       | East Trout Lake, Canada                              | 54.3° N  | 104.9° W  | 0.502                  | (Wunch et al., 2022)                             |  |  |
| TCCON       | Lamont, USA                                          | 36.6° N  | 97.5° W   | 0.320                  | (Wennberg et al., 2022b)                         |  |  |
| TCCON       | Park Falls, USA                                      | 45.9° N  | 90.3° W   | 0.440                  | (Wennberg et al., 2022c)                         |  |  |
| TCCON       | Caltech, USA                                         | 34.1° N  | 118.1° W  | 0.230                  | (Wennberg et al., 2022a)                         |  |  |
| TCCON       | Edwards, USA                                         | 34.0° N  | 117.8° W  | 0.700                  | (Iraci et al., 2022)                             |  |  |
| TCCON       | Rikubetsu, Japan                                     | 43.5° N  | 143.8° E  | 0.380                  | (Morino et al., 2022a)                           |  |  |
| TCCON       | Tsukuba, Japan                                       | 36.1° N  | 140.1° E  | 0.030                  | (Morino et al., 2022b)                           |  |  |
| TCCON       | Wollongong, Australia                                | 34.4° S  | 150.9° E  | 0.300                  | (Deutscher et al., 2023)                         |  |  |
| TCCON       | Lauder, New Zealand                                  | 45.0° S  | 169.7° E  | 0.370                  | (Pollard et al., 2022)                           |  |  |
| NDACC       | Tsukuba, Japan                                       | 36.1° N  | 140.1° E  | 0.030                  | (Morino et al., 2022b)                           |  |  |
| NDACC       | Wollongong, Australia                                | 34.4° S  | 150.9° E  | 0.300                  | (Jones et al., 2009)                             |  |  |
| NDACC       | Lauder, New Zealand                                  | 45.0° S  | 169.7° E  | 0.370                  | (Bègue et al., 2024)                             |  |  |
| NDACC       | Arrival Heights, Antarctica                          | 77.8° S  | 66.67° E  | 0.184                  | (Smale et al., 2021)                             |  |  |
| NDACC       | St. Petersburg, Russia                               | 59.9° N  | 29.8° E   | 0.020                  | (Makarova et al., 2024)                          |  |  |
| NDACC       | Jungfraujoch, Switzerland                            | 46.5° N  | 7.9° E    | 3.580                  | (Zander et al., 2008)                            |  |  |
| NDACC       | Altzomoni, Mexico                                    | 19.1° N  | 98.6° W   | 3.985                  | (Grutter et al., 2008)                           |  |  |
| In Situ     | Bukit Kototabang (BKT), Indonesia                    | 0.2° S   | 100.3° E  | 0.864                  | (Eko Cahyono et al., 2022), Reza Mahdi, BMKG     |  |  |
| In Situ     | Minamitorishima (MNM), Japan                         | 24° N    | 153.9° E  | 0.007                  | (Takatsuji, S., 2024a)                           |  |  |
| In Situ     | Ryori (RYO), Japan                                   | 39.3° N  | 141.8° E  | 0.260                  | (Takatsuji, S., 2024b)                           |  |  |
| In Situ     | Yonagunijima (YON), Japan                            | 24.4° N  | 123° E    | 0.030                  | (Takatsuji, S., 2024c)                           |  |  |
| In Situ     | Capo Granitola (CGR), Italy                          | 37.6° N  | 126° E    | 0.005                  | (Cristofanelli et al., 2017)                     |  |  |
| In Situ     | Cape Point (CPT), South Africa                       | 34.3° S  | 18.5° E   | 0.230                  | (Labuschagne et al., 2018)                       |  |  |
| In Situ     | Cape Verde Atmospheric Observatory (CVO), Cabo Verde | 16.9° N  | 24.9° W   | 0.010                  | (Kozlova, E. et al., 2021)                       |  |  |
| In Situ     | Jungfraujoch (JFJ), Switzerland                      | 46.5° N  | 7.9° E    | 3.580                  | (Hueglin et al., 2024), Martin Steinbacher, Empa |  |  |
| In Situ     | Mt. Kenya (MKN), Kenya                               | 0.06° S  | 37.3° E   | 3.678                  | (Kirago et al., 2023), Martin Steinbacher, Empa  |  |  |
| Tall tower  | East Trout Lake, Canada                              | 54.3° N  | 104.9° W  | $0.502^{a}$            | (Chen et al., 2014), Douglas Worthy, ECCC        |  |  |
| Aircraft    | East Trout Lake, Canada                              | 54.3° N  | 104.9° W  | -                      | (McKain et al., 2024)                            |  |  |

This is the surface altitude, and the measurement intake are at four levels (95, 55, 33, 22 m) installed on a 105 m SaskTel communication tower.

# 2.4 GEOS-Chem and prior estimates

245


The GEOS-Chem model (http://www.geos-chem.org, last access: 1 July 2024) is a global 3D CTM that uses assimilated meteorological observations as input from the NASA Global Modelling and Assimilation Office (GMAO). We use version 14.1.1 of the GEOS-Chem CTM driven by meteorological input from the Modern-Era Retrospective analysis for Research and

Applications, Version 2 (MERRA-2; Gelaro et al., 2017). The meteorological fields have a native resolution of 0.25° × 0.3125° with 72 vertical levels from the surface to 0.01 hPa, which is degraded to 2° × 2.5° horizontal grid and 47 vertical levels (Bey et al., 2001). For the purpose of global CO assimilation in this study, the linear CO-only simulation of GEOS-Chem, also known as "tagged CO", is used with prescribed monthly mean OH fields from a 10-year archived full chemistry simulation based on version 14 of the model. The tagged CO simulation reduces the computation cost relative to the full-chemistry and has been widely applied in different studies in the past (Heald et al., 2004; Jiang et al., 2017; Jones et al., 2009; Kopacz et al., 2010; Lutsch et al., 2020; Tang et al., 2023; Wunch et al., 2019). Version 14.1.1 of the tagged CO simulation incorporated the improved secondary CO production scheme for the tagged CO simulation, which reduces the differences between full chemistry and tagged CO simulations, especially in regions strongly influenced by biogenic emissions and chemistry (Fisher et al., 2017). The biogenic source of CO in the full chemistry simulation is based on the oxidation of volatile organic compounds (VOCs) produced by the Model of Emissions of Gases and Aerosols from Nature (MEGAN version 2.1) inventory (Guenther et al., 2012).

For the results presented here, we specify fossil fuel emissions of CO from the Community Emissions Data System (CEDS) inventory (Hoesly et al., 2018). Biomass burning (BB) emissions are based on the four different BB inventories described below. These BB emissions have been used in various studies (Griffin et al., 2020; Jin et al., 2024; Li et al., 2020; Zhang et al., 2022), and are used as our prior emissions in the inversion analyses conducted here.

# 2.4.1 QFED







The Quick Fire Emissions Dataset version 2.5r1 (QFED v2.5r1) (Koster et al., 2015), is a global product of biomass burning emissions which was developed for the NASA GEOS model. It applies the fire radiative power (FRP) method with a cloud correction technique (Koster et al., 2015), where the location of fires and FRP are derived from the polar orbiting Moderate Resolution Imaging Spectroradiometer (MODIS) instrument onboard the NASA's Terra and Aqua satellites. QFED provides daily emissions with a horizontal spatial resolution of 0.1° x 0.1°, and GEOS-Chem applies a climatological profile based on WRAP (Western Regional Air Partnership) method (WRAP, 2005) to distribute the emissions over the diurnal cycle. In the QFED implementation in GEOS-Chem, the setup of the plume injection height follows Fischer et al. (2014) and Travis et al. (2016), where the 65% of the biomass burning emissions are allocated to the planetary boundary layer (PBL) and the remaining 35% belongs the troposphere. The **OFED** to free data can be accessed from http://geoschemdata.wustl.edu/ExtData/HEMCO/QFED/v2023-05/ (last access: 1 July 2024).

# **2.4.2 GBBEPx**

The Blended Global Biomass Burning Emissions Product eXtended (GBBEPx v4, Zhang et al., 2012; 2019), developed by NOAA National Environmental Satellite, Data, and Information Service (NESDIS), produces daily global biomass burning emissions. GBBEPx blends information from QFED and fire emissions estimated from the Visible Infrared Imaging Radiometer Suite (VIIRS) instrument onboard Suomi National Polar-orbiting Partnership (SNPP) and Joint Polar Satellite

System (JPSS). VIIRS fire emissions are obtained using FRP derived from VIIRS data in an approach that is similar to the use of MODIS FRP data in QFED (Csiszar et al., 2016), but with a different fire detection scheme (Zhang et al., 2019). The blended GBBEPx emissions are produced daily at a resolution of  $0.25^{\circ} \times 0.3125^{\circ}$ , and the same profile is applied to distribute the emissions over the diurnal cycle as is used for QFED. The implementation of GBBEPx in GEOS-Chem assumes the same injection height scheme as for QFED. The GBBEPx v4data can be accessed from https://www.ospo.noaa.gov/pub/Blended/GBBEPx/ (last access: 1 July 2024).

# 2.4.3 **GFAS**


The Global Fire Assimilation System (GFAS v1.2, Kaiser et al. 2012; Di Giuseppe et al., 2021), utilized by the Copernicus Atmosphere Monitoring Service (CAMS), provides daily estimate of biomass burning emissions by assimilating FRP observations from MODIS instruments on the Terra and Aqua satellites. GFAS estimates emissions by conversion of FRP to the dry matter burned and the use of biome-specific emission factors. GFAS utilizes the vegetation type prescribed by the Global Fire Emissions Database (GFED). The daily data are globally available at a resolution of 0.1° × 0.1° from 2003 to the present time, which can be accessed from https://ads.atmosphere.copernicus.eu/datasets/cams-global-fire-emissions-gfas?tab=overview, (last access: 11 June 2024). In the GEOS-Chem simulations, we use the same diurnal cycle as used in QFED and GBBEPx. Additionally, GFAS provides information about the daily injection height (i.e., mean altitude of maximum injection (MAMI)) of the emissions. In GEOS-Chem, it is assumed that the emissions are injected uniformly from the surface to the MAMI.

# 295 **2.4.4 CFFEPS**


The Canadian Forest Fire Emissions Prediction System (CFFEPS v4.0, Chen et al., 2019) produces biomass burning emissions for North America using information from the Canadian Forest Service (CFS), Canadian Wildland Fire Information System (CWFIS), and meteorological inputs from ECCC's Global Environmental Multiscale (GEM) model. The product provides hourly fire emissions and smoke plume injection height at individual hotspot locations. In implementing the CFFEPS emissions in GEOS-Chem, the emissions were aggregated to the GEOS-Chem grid resolution with a weighted average plume height based on the CO<sub>2</sub> emission level. The CO<sub>2</sub> emission level was used in determining the injection height for all species in CFFEPS, even though here we focus only on the CO emissions. The plume injection height estimates from the FireWork plume rise model (Anderson et al., 2011; Chen et al., 2019) on which CFFEPS is based, have been validated against satellitedriven (e.g., TROPOMI) aerosol plume heights (Griffin et al., 2020).

#### 305 3 Inversion methodology

The inversion framework utilizes the CHEEREIO assimilation toolkit (Pendergrass et al., 2023), which employs a localized ensemble transform Kalman filter (LETKF). A detailed description of the LETKF algorithm used is provided in Hunt

et al. (2007). We use CHEEREIO to derive optimized estimates of globally gridded emissions of CO between May and September 2023 at a spatial resolution of  $2^{\circ} \times 2.5^{\circ}$  and a temporal resolution of three days by assimilation of TROPOMI satellite and TCCON observations. The solution state vector of emissions,  $x^{a}$ , is given by

$$\overline{\mathbf{x}^a} = \overline{\mathbf{x}^b} + \gamma \mathbf{X}^b \widetilde{\mathbf{P}^a} (\mathbf{Y}^b)^T \mathbf{R}^{-1} (\mathbf{y}^o - \overline{H(\mathbf{x}^b)}), \tag{1}$$

where the overbar represents the ensemble mean,  $x^b$  is the background state vector,  $y^o$  is the observation vector,  $H(x^b)$  is the model simulation of the observations with observation operator H, R is the observation error covariance matrix,  $P^a$  is the analysis error covariance matrix where tilde represents the ensemble space,  $X^b$  is the background perturbation matrix,  $Y^b$  is the observation perturbation matrix (see the detailed description of LETKF variables in Hunt et al., 2007).  $\gamma$  is a regularization factor used to prevent overfitting or underfitting to observations by balancing the relative influence of the a priori estimate and the measurements in the inversion. It serves as a pragmatic correction for uncertainties that are hard to quantify, such as unaccounted observation error correlations (Hakami et al., 2005; Lu et al., 2022; Voshtani et al., 2023).







To simulate the observations, we have developed an observation operator for each measurement type. The observation operator maps the model emission fields (i.e., state space) into the observation space as follow:

$$H(\mathbf{x}^b) = h_v[\mathbf{c}^{a\ priori} + \mathbf{A}(\mathbf{M}(\mathbf{x}^b) - \mathbf{c}^{a\ priori})],\tag{2}$$

where A denotes the averaging kernel of the retrieval, capturing the vertical sensitivity of the retrieval profiles relative to the real atmosphere,  $c^{a priori}$  is the a priori profile provided by measurement data,  $M(x^b)$  represents a forward operator that operates on the emissions state vector and produces CO profiles that are spatially and temporally interpolated at the locations and times of the measurements, and  $h_v$  describes the vertical summation operator based on pressure weighting for computing model-equivalent column retrievals. To obtain a posteriori estimate of CO emissions, we begin by initializing the ensemble scaling factors using a multiplicative random perturbation to the prior estimates from all emission sources at the grid scale (i.e., spatially varying perturbations), sampled from a multivariate lognormal distribution (see the detailed description of ensemble generation in Pendergrass et al. (2023), Section 3.2). We assume lognormal errors on the prior emissions to ensure the positivity of the solution (i.e., prevent unrealistic negative scaling factors) and to better capture the skewed tails of the emissions distribution (Maasakkers et al., 2019; Plant et al., 2022). In the next step, CHEEREIO first runs GEOS-Chem for each ensemble member over the assimilation window and then applies Eq. 2 to those ensembles in the LETKF process. This process further scales the emissions based on the observation increments (i.e.,  $y^0 - H(x^b)$ ), and the observation and the prior error covariances (Eq.1). Note that during the construction of error covariances, a logarithmic transform of the scaling factor distributions to a normal distribution is required to satisfy the assumptions of LETKF (Hunt et al., 2007), which can be transformed back to the lognormal distribution after the LETKF process. Finally, gridded total CO emissions from all sources, including biomass burning (BB), fossil fuel, and biogenic emissions, are updated through the inversion process. Note that the analysis here will focus on regions where BB plays a dominant role in the attribution of CO emissions between May and September 2023, and where BB emissions are spatially distinct from fossil fuel emissions. As a result, misattribution of CO emissions to BB from other sources will be minor and likely falls within the uncertainty bounds of the a posteriori estimates.

To obtain an efficient performance of the inversion using TCCON data, it is important to tune the assimilation parameters with the available configuration in CHEEREIO. This was accomplished through a series of observing system simulation experiments (OSSEs), which are described in Appendices A and B. We use 24 ensemble members following previous inversion studies with the same approach (Liu et al., 2019; Pendergrass et al., 2023). Our sensitivity test with a larger ensemble size of 36 produced nearly identical posterior error estimates, with negligible improvements; thus, for saving on the computational cost of the analysis, we used the smaller ensemble size.

Before starting the inversion, we first conducted a 1-year model spin-up in 2022 (January-December) for all experiments to minimize the impact of the initial conditions on the analysis. Then, an ensemble spin-up without assimilating observations is performed, where emissions are randomly perturbed at each grid point based on a lognormal distribution to create an ensemble spread. We assume a lognormal standard deviation of  $\sigma = 0.2$ , centered around 1, that provides ensemble member scaling factors between 0.55 and 1.82 with 99% confidence. This perturbation level is sufficient to generate meaningful ensemble spread while avoiding unrealistically high or low values for constructing the prior error covariance. The scaling factors are assumed to be spatially correlated using an exponential decay function with a correlation distance of 500 km, while no explicit temporal correlations are imposed. We use a spin-up of about three months, comparable to the CO lifetime during summer, not only to provide a reasonable spread in the ensemble members but also to ensure the concentrations will reflect the perturbations in emissions. At the start of the assimilation, we adjusted the ensemble members by a global multiplicative factor, making the ensemble mean equivalent to the TROPOMI and TCCON observations. This maintains a globally unbiased field of concentrations relative to the observations. Because the LETKF is sequential, it takes some time for the observations to provide sufficient information to update the emissions. To account for this lag, we use a one-month burn-in period in CHEEREIO (Pendergrass et al., 2023), for which the inversion output is discarded in postprocessing.

The performance of the inversion can be also enhanced using different LETKF parameters, such as localization and inflation parameters (Bisht et al., 2023; Miyazaki et al., 2012, 2020). For the TROPOMI and TCCON data, we use regularization factors,  $\gamma_{TROPOMI} = 0.2$  and  $\gamma_{TCCON} = 5$ , respectively, estimated separately for each observation type through the OSSE experiments (see Appendix B). These factors scale the observation error covariances to balance the weight given to the measurements relative to the prior weight in the inversion. For the high spatial density of satellite observations, a factor of  $\gamma < 1$  is required to prevent overfitting, while for sparse measurements like TCCON, a factor of  $\gamma > 1$  is typically suitable to prevent underfitting of the observations. We also use an inflation factor ( $\Delta = 0.08$ ) to compensate for a rapid reduction of the ensemble spread, which may otherwise prevent the inversion from being updated by subsequent observations. We set a localization radius of 500 km following previous CO inversion studies (Gaubert et al., 2023; Miyazaki et al., 2015) to avoid the impact of distant observations, which may be affected by sampling errors and spurious correlations (see Fig. S1). We generate super-observations for both TROPOMI and TCCON by aggregating measurements to the model time and grid. This aggregation helps mitigate spatiotemporal representativity errors and facilitates LETKF computations. The associated super-observation errors are calculated as described in Section 2.1. We further apply an inflation factor to the observation errors, while initially assuming an observation error correlation of 0.28, following previous studies (Chen et al., 2023; Pendergrass et

al., 2023). We do not explicitly account for model transport errors in calculating the super-observations, but the influence of these errors will be captured by the error inflation in the inversion.

We conducted a series of assimilation experiments, which are listed in Table 2, to assess the impact of the use of TCCON and TROPOMI data and the choice of a priori emission inventories on the inferred CO emissions. Each experiment pairs a specific emission inventory—QFED, GBBEPx, GFAS, or CFFEPS—with either only TROPOMI observations or both TROPOMI and TCCON data in the assimilation process. Although the emissions are optimized at the grid box scale, we aggregate emissions for the following five regions (shown in Fig. 2), where there are typically significant fire emissions between May and September: North America (NA), Siberia (SI), South America (SA), Africa (AF), and South Asia and Australia (SA&A). Any emission from outside of these five regions are captured in the Rest of the World (ROW) category. The emissions from the five regions account for 90%–95% of biomass burning emissions globally.



**Figure 2:** Major source regions of the biomass burning CO emissions used in the inversion analysis between May-September 2023, with an example of the a priori emission estimate using the GFAS bottom-up inventory. The a priori estimates for these regions from other inventories are listed in Table 3.

| Experiment | Biomass burning emission inventory | Assimilated observations <sup>b</sup> |  |  |  |
|------------|------------------------------------|---------------------------------------|--|--|--|
| 1          | QFED                               | -                                     |  |  |  |
| 2          | QFED                               | TROPOMI                               |  |  |  |
| 3          | QFED                               | TROPOMI+TCCON                         |  |  |  |
| 4          | GBBEPx                             | -                                     |  |  |  |
| 5          | GBBEPx                             | TROPOMI                               |  |  |  |
| 6          | GBBEPx                             | TROPOMI+TCCON                         |  |  |  |
| 7          | GFAS                               | -                                     |  |  |  |
| 8          | GFAS                               | TROPOMI                               |  |  |  |
| 9          | GFAS                               | TROPOMI+TCCON                         |  |  |  |
| 10         | CFFEPS <sup>a</sup>                | -                                     |  |  |  |
| 11         | CFFEPS                             | TROPOMI                               |  |  |  |
| 12         | CFFEPS                             | TROPOMI+TCCON                         |  |  |  |

<sup>&</sup>lt;sup>a</sup> The inversions based on CFFEPS emission inventory in North America uses GFAS global emissions for the regions outside of North America.

#### 4 Results and discussions





This section presents our main results by first assessing the CO emissions on a global scale (Section 4.1) and then focusing on North American emissions (Section 4.2), where the most extreme fire events took place during the study period in summer 2023. A series of experiments are conducted as listed in Table 2 for both global and regional analysis. For the global analysis (Sections 4.1.1-4.1.3), we use three biomass burning emission inventories (QFED, GBBEPx, GFAS) as priors, and compare how assimilating TROPOMI satellite observations alone or jointly with TCCON ground-based measurements affects our emissions estimates. A method to measure and compare error variance and information content is presented. For evaluation, two types of independent data, including NDACC total column and surface WDCGG measurements, in addition to the same TCCON observations (non-independent) are used. Focusing on North America analysis (Sections 4.2.1-4.2.2), we include an additional regional prior emissions inventory from CFFEPS (Chen et al., 2019) provided by ECCC and additional experimental InSb XCO data from TCCON at ETL in our analysis. First, our analysis explores the spatiotemporal variability of a posteriori and a priori CO fields during extreme fire episodes. Then, as a case study at ETL, we assess the impact of assimilating additional information from experimental TCCON data at ETL, with unique retrieval characteristics, on local emissions constraints. The evaluations are performed using independent aircraft and tall tower measurements. The discussion presented

b No observations are assimilated (-), corresponding to the model a priori or control run with a particular biomass burning emission inventory.

in these two sections allows us to highlight both the broad and local impacts of our approach and the specific improvements achieved in areas most affected by fires.

# 4.1 Global analysis





# 4.1.1 Comparison between prior and posterior emissions

Table 3 shows the total regional BB CO emissions from the a priori bottom-up inventories and the a posteriori emission estimates (i.e., ensemble mean) obtained from the TROPOMI-only assimilation and the joint TROPOMI and TCCON (i.e., TROPOMI+TCCON) assimilation. The standard deviations shown in this table are posterior ensemble spread based on the LETKF assimilation, which is referred to as posterior uncertainty throughout the discussion in this study. The vector of posterior uncertainties has the same size as the state vector, computed for each grid cell after each assimilation window. Accordingly, the posterior uncertainty in an inversion region (i.e., regional or global scale) is the ensemble standard deviation of the total emissions. The total emissions per ensemble is obtained by summing the emissions across all grid cells in that region. For the inversion period between May and September 2023, there is a large discrepancy in the a priori emissions between the three global inventories, with GFAS producing the highest global emissions of 230.3 Tg CO, which is a factor of 1.3 times GBBEPx (182.6 Tg CO) and 1.4 times QFED (164.5 Tg CO) emissions. These differences are more substantial in the regions of North America (NA) and Siberia (SI), where the boreal wildfires play an important role. In NA, the GFAS emissions are greater than the emissions from GBBEPx and QFED by a factor of 2.1 and 2.5, respectively. In SI, GFAS is a factor of 3.1 and 2.1 greater than GBBEPx and QFED, respectively. Although all GFAS, GBBEPx, and QFED are based on estimates of FRP derived from satellite products (i.e., MODIS and/or VIIRS), there are many driving factors that cause the differences in the total emissions (Li et al., 2020; Liu et al., 2024), which can reach up to an order of magnitude in a finer spatiotemporal scale (Stockwell et al., 2022). In contrast, the CFFEPS emissions in NA are comparable to GFAS despite using a different approach for estimating the emissions.

**Table 3:** Biomass burning CO emissions estimates of a priori and a posteriori in the major source regions based on Fig. 2 for the period between May and September 2023. The a posteriori estimates are denoted by the ensemble mean and standard deviation.

| Region                                     | A priori <sup>a</sup> (Tg CO) |        |       |                     | A posteriori TROPOMI-only (Tg CO) |                |                | A posteriori TROPOMI+TCCON (Tg CO) |              |                |                |         |
|--------------------------------------------|-------------------------------|--------|-------|---------------------|-----------------------------------|----------------|----------------|------------------------------------|--------------|----------------|----------------|---------|
|                                            | QFED                          | GBBEPx | GFAS  | CFFEPS <sup>b</sup> | QFED                              | GBBEPx         | GFAS           | CFFEPS                             | QFED         | GBBEPx         | GFAS           | CFFEPS  |
| North                                      |                               |        |       |                     | 440.4                             |                |                |                                    |              |                | 4004           |         |
| America                                    | 37.1                          | 42.7   | 91.0  | 90.2                | 110.4 ±                           | 112.8 ±        | 127.2 ±        | 125.6 ±                            | 111.2 ±      | 115.3 ±        | 130.1 ±        | 129.3 ± |
| (NA)                                       |                               |        |       |                     | 19.8                              | 19.9           | 17.2           | 17.7                               | 15.5         | 15.1           | 12.9           | 12.5    |
| Africa (AF)                                | 65.1                          | 87.4   | 53.5  |                     | 104.6 ±                           | 117.2 ±        | 100.3 ±        |                                    | 103.4 ±      | 117.8 ±        | 100.8 ±        |         |
|                                            |                               |        |       |                     | 17.3                              | 18.8           | 17.4           |                                    | 16.2         | 17.7           | 16.4           |         |
| Siberia (SI)                               | 16.5                          | 11.1   | 34.5  |                     | 29.4 ± 9.8                        | 24.1 ± 8.5     | 44.5 ± 12.6    |                                    | 29.1 ± 8.2   | 22.8 ± 6.9     | 46.4 ± 9.2     |         |
| South America (SA)                         | 22.1                          | 22.1   | 19.1  |                     | 27.2 ± 10.3                       | 27.4 ± 10.3    | 24.9 ± 10.0    |                                    | 28.1 ± 9.7   | 27.0 ± 9.4     | 25.0 ± 9.1     |         |
| South Asia<br>& Australia<br>(SA&A)        | 13.8                          | 12.1   | 16.3  |                     | $17.2 \pm 6.4$                    | $15.6 \pm 6.3$ | $19.5 \pm 6.8$ |                                    | 17.2 ± 5.7   | $16.1 \pm 5.9$ | 20.2 ± 6.1     |         |
| Rest of the<br>World<br>(ROW) <sup>c</sup> | 9.8                           | 7.2    | 5.8   |                     | 14.1 ± 3.0                        | $12.5 \pm 3.0$ | 12.2 ± 3.2     |                                    | 18.9 ± 2.7   | 12.6 ± 2.2     | $16.2 \pm 2.7$ |         |
| Global                                     | 164.5                         | 182.6  | 230.3 |                     | 302.9 ± 66.6                      | 309.6 ± 66.8   | 328.6 ± 67.1   |                                    | 307.9 ± 58.0 | 311.6 ± 57.1   | 338.7 ± 56.4   |         |

<sup>&</sup>lt;sup>a</sup> A priori Fossil fuel and biofuel emissions are provided by CEDS inventory and biogenic emissions are obtained from MEGAN version 2.1 in all the cases.

After assimilating the TROPOMI data, the a posteriori emissions suggest a large increase in global emissions from the a priori, with global a posteriori emissions of all inventories in close agreement with each other. We estimate global fire emissions of 303±67 Tg CO, 310±67 Tg CO, and 329±67 Tg CO for QFED, GBBEPx, and GFAS, respectively. In North America, the a posteriori emission estimates are also consistent, with estimates of 126±18 Tg CO, 127±17 Tg CO, 113±20 Tg CO, and 110±20 Tg CO for CFFEPS, GFAS, GBBEPx, and QFED, respectively. For most other regions, the inferred emissions all agree with within the a posteriori uncertainty except for Siberia, where the a posteriori emission estimates are 45±13 Tg CO, 24±9 Tg CO, and 29±10 Tg CO for GFAS, GBBEPx, and QFED, respectively. The discrepancy between the a posteriori Siberian emissions could be because the assimilation did not ingest TROPOMI observations poleward of 60°N, and there are large emission sources poleward of 60°N, as can be seen in Fig. 4. Note that although the emissions beyond 60° may not be directly corrected from local observations at the current assimilation step, they can still be updated using observations between 60°S-60°N and through model transport and cycling of the assimilation that propagate information globally. The overall

<sup>&</sup>lt;sup>b</sup> The global inversions with CFFEPS BB emissions uses GFAS global BB emissions for the regions outside of North America.

<sup>&</sup>lt;sup>c</sup> The source of CO in the rest of the world also includes the oxidation of methane and non-methane hydrocarbons.

agreement between the a posteriori emission estimates obtained with the different a priori emissions suggest that TROPOMI provides sufficient information to constrain the regional emission estimates.





The joint TCCON and TROPOMI inversion produces a posteriori emission estimates that agree to within 5% of the a posteriori emissions from the TROPOMI-only inversion, but these differences vary among the source regions. At the global scale, the a posteriori estimates remain within  $1\sigma$  uncertainty, implying that the increment from the joint inversion closely matches that from the TROPOMI-only inversion. Previous inversion studies with CO, and CO2 measurements showed that combining satellite total column observations with surface in situ measurements (Byrne et al., 2020; Kim et al., 2024; Wang et al., 2018) could benefit from the complementarity between the two types of measurements and provide posterior fluxes that are better informed by measurements. We will discuss later in this section the spatiotemporal distributions of the a posteriori emissions in comparison with the a priori across all inventories. Table 3 shows that the joint inversion reduces the uncertainties in the posterior relative to the inversion using only the TROPOMI data. We find a global uncertainty reduction of nearly 15% in all the a posteriori emission estimates when including the standard TCCON XCO in the inversion. This reduction is likely due to the additional temporal information and higher accuracy provided by TCCON, compared to TROPOMI. The reduction of uncertainty varies between the source regions. In the Northern Hemisphere extratropics, where most TCCON stations are located, the uncertainty reduction reaches 29% in NA with CFFEPS emissions, followed by Siberia with GFAS emissions (26% reduction). On the other hand, for the tropics and subtropics in the Southern Hemisphere, where there are fewer TCCON stations, we find smaller reductions in uncertainties (between 5% and 10%). Goudar et al. (2023) investigated the uncertainties in estimating CO emissions from isolated fires using TROPOMI assimilation. They reported that these estimated uncertainties primarily arise from errors due to spatial under-sampling of the CO field by TROPOMI observations and errors due to assumptions about the temporal variability of the emissions. Although we did not examine individual fire events, the lower overall uncertainty from the joint TCCON and TROPOMI assimilation suggests an improved handling of the spatial undersampling error in TROPOMI-only assimilation, which is reflected in the uncertainty estimates. This improvement could be particularly important, as it shows the isolated effect of adding TCCON data to the inversions, when other factors such as a priori emissions and their errors were kept fixed between the joint and TROPOMI-only assimilations.

**Figure 3:** Comparison of the temporal variability of the CO emissions estimate for the priors (blue), posteriors using TROPOMI-only assimilations (green), and posteriors using joint TROPOMI and TCCON assimilations (red), among the three global biomass burning inventories, including (a-e) QFED, (f-j) GBBEPx, and (k-o) GFAS used as the priors, and for the major inversion source regions as shown in Fig. 2.

Figure 4: Comparison of the spatial distribution of the time-averaged biomass burning CO emissions in the (a-c) a priori, (d-f) a posteriori – a priori using TROPOMI and TCCON assimilation, and among three global biomass burning inventories.


To better understand the influence of assimilating TCCON XCO together with the TROPOMI XCO, we examine the temporal and spatial variability of the estimated emissions. Fig. 3 shows a priori emissions in blue, a posteriori emissions from the TROPOMI-only assimilation in green, and the a posteriori emissions from the joint inversion in red between May and September. In North America, QFED shows only slight variations in the a priori emissions (Fig. 3a). GBBEPx (Fig. 3f) shows some degree of improvement over QFED relative to TROPOMI posterior during a few fire episodes. For GFAS, however, this improvement over QFED (Fig. 3k) is significant, so that the GFAS prior exhibits reasonable agreement in both magnitude and temporal variability with the a posteriori emissions from the TROPOMI-only assimilation. The posteriors from different priors, but with the same set of assimilated observations, show overall stronger temporal variability compared to their respective priors. For instance, in North America, while the QFED and GBBEPx priors are relatively flat compared to the GFAS prior, all their posteriors exhibit enhanced and more consistent variability (Fig. 3a,f,k). Still, differences to some extent remain

noticeable between these posterior time series, likely driven by differences in the spatial and temporal distribution of the priors, and their interaction with observational constraints through model transport and mixing. Comparison of the spatial distribution of the a priori emissions (Fig. 4a-c) indicates that two main regions of boreal wildfires in (eastern) Quebec and (western) Alberta and British Columbia, Canada, correspond to the large differences in regional emissions among the inventories, although their overall global spatial distributions are similar. The a posteriori – a priori emissions from the TROPOMI-only inversions confirm the underestimate of CO emissions in QFED and GBBEPx, in those regions in Canada (Fig. 4d-f). GFAS, unlike the other two inventories, has significantly larger emissions from wildfires across Canada, so that they are comparable with the magnitude of the a posteriori emissions in that region (see Table 3 for a comparison of total emissions and Fig. S2 in the supplements for the separate a posteriori emission maps associated with Fig. 4).








In Siberia, the a posteriori emissions provide an enhancement on the a priori in a few episodes in July and August (Fig. 3c,h,m), indicating an overall low level of emissions from all inventories. GFAS, followed by QFED, has not only higher emissions in the same locations as GBBEPx, but also has a greater area of BB emissions. However, the a posteriori emissions suggest that the GFAS inventory required larger adjustments in the central and western part of SI, suggesting errors in the spatial allocation of the a posteriori fire emissions in Siberia. In Africa and South America, the emissions enhancement occurs in late July through the end of the inversion period in September, which are shown in Fig. 3b,g,l,d,i,n. For these regions, GBBEPx, followed by QFED, provide a closer estimate to the a posterior than GFAS, while the posterior emissions from the TROPOMI-only and joint inversions remain nearly identical (Fig. 4d-f versus 4g-i). This could be because there are very few TCCON measurements in the Southern Hemisphere near that region. In South Asia and Australia, the emissions enhancement occurs only in two episodes in early May and early September, which remain nearly consistent among the inventories. Finally, comparing the regional a posteriori emissions from the TROPOMI-only and joint inversions (Fig. 3k,l,m,n,o), we find different temporal variability of the updated emissions, which are distinctive for several wildfire episodes. Additionally, we observed improvements in the correlation between analogous posterior timeseries across different BB inventories. For example, the temporal correlation between the OFED and GFAS posterior emissions in North America increased from r = 0.85 in the TROPOMI-only assimilation to r = 0.90 in the joint TROPOMI and TCCON assimilation. These findings suggest that the higher temporal resolution of the TCCON measurements provide additional constraints on temporal variability of the emissions in the inversion, which is consistent with findings from previous studies on CO<sub>2</sub> inverse modelling (Byrne et al., 2020, 2024; Chevallier et al., 2011).

Overall, according to the discussion above and the results from the Observing System Simulation Experiments (OSSEs) demonstrated in Appendices A and B, we find that although TCCON alone may not significantly constrain spatiotemporal variability in the major inversion regions—likely due to the limited number of measurement sites—it is still clear that adding TCCON to TROPOMI in the joint inversion reduces the posterior uncertainty estimates everywhere compared to the TROPOMI-only inversion. We found that the reduction of the uncertainty by adding TCCON measurements becomes more significant during high BB emission episodes or wildfire events. Later, in Section 4.1.3, we evaluate the a priori together with the a posteriori from both TROPOMI-only and joint inversions using independent measurements.

#### 4.1.2 Error variance reduction and the information content




We evaluate the performance of the inversion for constraining BB CO emissions by quantifying the information content provided by the TROPOMI and TCCON data. To achieve this, we use two approaches: (i) computing the reduction of uncertainty in the model space and (ii) computing the degree of freedom for signal (DOFS) in the ensemble subspace. Using these methods will quantify the information provided by the two observing systems individually. In the first approach, we compute the a priori and a posteriori error variance for each grid point, which is obtained as part of the solution for LETKF processing in CHEEREIO. The reduction of error variance can be used as a metric for evaluating the inversion performance (Feng et al., 2009); as such, a greater error variance reduction at a grid point indicates that more reliable information from observing system is available to constrain emissions for that grid. Accordingly, we define a normalized error reduction ( $\varepsilon$ ) for each grid point as follows:

$$\varepsilon_i = 1 - \frac{(\sigma_i^a)^2}{(\sigma_i^b)^2},\tag{3}$$

where  $(\sigma_i^a)^2$  and  $(\sigma_i^b)^2$  denote the error variance of the a posteriori and the a priori, respectively, for the i<sup>th</sup> emissions in the state vector.  $\varepsilon_i$  varies between 0 to 1, with greater values indicating a higher reduction of a priori uncertainties. Although the LETKF method approximates the a posteriori uncertainty due to the reduced rank representation associated with the limited number of ensembles used to construct the error covariances (Livings et al., 2008), it still provides useful information with which to evaluate the analyses.

**Figure 5:** Comparison of the time-averaged (May-September 2023) a posteriori error variance reduction,  $\varepsilon$ , of (a-c) TROPOMI-only assimilation and (d-f) joint TROPOMI and TCCON assimilation for three global biomass burning inventories used as a prior.

Fig. 5 shows the error variance reduction,  $\varepsilon_i$ , for the a posteriori emissions based on the three different global BB inventories (i.e., QFED, GBBEPx, and GFAS) using TROPOMI-only measurements (Fig. 5a-c) and using joint TROPOMI and TCCON measurements (Fig. 5d-f). The greater reduction in error variances implies higher confidence in the posterior estimate at those locations. We find greater reduction of error variance in the joint inversion compared to the inversion with TROPOMI-only data, primarily in NA and in the vicinity of TCCON stations. In fact, the reductions correlate with the weight of observations compared to the model a priori, so that increasing the weight of observations with respect to the model a priori could result in higher reductions of the error variances. For example, with QFED as the a priori emissions, the rate of reduction is greater in boreal Canada, central and southern part of the United States, western Europe, eastern Asia, Siberia, and Australia. This indicates where TCCON provides additional information to further constrain the emissions based on the QFED BB inventory. In addition, comparing the cases with different inventories suggests that there could be differences in error variance reduction due to the model a priori. Accordingly, the slightly greater reduction with GFAS, compared to QFED or GBBEPx,

is likely due to its higher spatiotemporal variability, which enables the inversion to better exploit the information from observations.

**Figure 6:** Computed degree of freedom for signal in ensemble subspace (DOFS<sub>k</sub>) associated to the TROPOMI-only (blue) and joint TROPOMI and TCCON (red) inversions for three global biomass burning inventories (QFED, GBBEPx, GFAS) used as a priori.

In the second approach, following Zupanski et al. (2007), we compute the degree of freedom for signal (DOFS) approximated for the ensemble-based assimilation method. The DOFS, as defined by Rodgers (2000), quantifies the number of pieces of independent information in an observing system toward constraining the state vector of dimension n (also equivalent to the total number of grids in the model). It is defined as

$$DOFS_n = tr(\mathbf{I}_{n \times n} - \mathbf{P}_{n \times n}^a (\mathbf{P}_{n \times n}^b)^{-1}) = tr(\mathbf{A}_{n \times n}), \tag{4}$$

where  $P_{n\times n}^a$  and  $P_{n\times n}^b$  are analysis and background error covariances,  $I_{n\times n}$  denotes the identity matrix, and  $A_{n\times n}$  represents the averaging kernel matrix. To compute  $A_{n\times n}$  and then DOFS<sub>n</sub>, a Jacobian matrix must be constructed in full rank, requiring extensive computational cost (e.g., Varon et al., 2022). In an ensemble-based approach, computing the Jacobian matrix in the state space is impractical since the limited number of ensembles are not sufficient to describe full rank error covariances. However, those quantities can be approximated for the ensemble subspace with reduced rank error covariance matrices, so that the information provided by the observation system is measured relative to the maximum independent pieces of information determined by the ensembles size, k. Therefore, the DOFS<sub>k</sub> is defined as,

$$DOFS_k = tr((\mathbf{I}_{k \times k} + \mathbf{C}_{k \times k})^{-1} \mathbf{C}_{k \times k}) = tr(\mathbf{A}_{k \times k}),$$
(5)

where  $C_{k\times k}$  denotes the symmetric information matrix,  $I_{k\times k}$  is the identity matrix, and  $A_{k\times k}$  represents an equivalent averaging kernel or influence matrix, all obtained in the ensemble subspace. The derivation of Eq. (5) started from Eq. (4) is described in Zupanski et al. (2007) and Zupanski (2005). Subsequently, we can compute the information matrix,  $C_{k\times k}$ , either within the LETKF calculation of CHEEREIO or as a postprocessing step if all the outputs from the ensemble members and the control run are already stored. Accordingly, every element of matrix C, are computed as

$$\mathbf{C}_{ij} = \mathbf{z}_i^T \mathbf{z}_i, \tag{6}$$

where z is a vector of dimension m (the number of observations) and is defined as

570

$$\mathbf{z}_{i}(m) = \mathbf{R}_{m \times m}^{-\frac{1}{2}} \left( H_{m}(\mathbf{x}_{i}) - H_{m}(\mathbf{x}) \right),$$
 (7)

where the observation operator  $H_m$  is applied to the perturbed,  $x_i$ , and unperturbed state vector x, and weighted by the inverse of square root of observation error covariance  $R_{m\times m}$ . Migliorini (2013) used the same method in the square root filter, where the forecast error covariance matrix is approximated by the sample covariance matrix, which is produced by the forecast of each ensemble member. Once matrix C is constructed, one can use Eq. (5) to obtain  $A_{k\times k}$  and the DOFS<sub>k</sub> for the ensemble subspace. Note that using this approach, there are at most k-1 independent pieces of information for the entire assimilation period. Thus, with k=24 in this work, the computed DOFS may vary between 0 and 23, mainly depending on the characteristics of the assimilated observations, such as their density and error statistics. Although this method does not produce DOFS in the state space of the emissions (Žagar et al., 2016), it enables a straightforward comparison of the information content across different experiments.

The computed DOFS for different inversions are shown in Fig. 6. Adding TCCON data to the inversion increases the DOFS from the TROPOMI-only inversion for all the cases. These values increase from 11.1, 11.4, and 12.7 for the TROPOMI-only inversions to 15.5, 15.6, and 16.9 for the inversions with TROPOMI and TCCON data using QFED, GBBEPx, and GFAS emissions, respectively. The higher DOFS from GFAS compared to the other BB inventories is also in agreement with its higher reduction of uncertainty in Fig. 5. This likely implies that the difference between the perturbed and unperturbed forecast of the state vector, which defines the elements of matrix  $\boldsymbol{\mathcal{C}}$ , correlates with the spatiotemporal variability of the prior emissions. Thus, GFAS prior, with greater variability than the other priors (e.g., Fig. 3a,f,k and Fig. 4a-c), may result in higher DOFS.

### 4.1.3 Evaluation using ground-based observations

We evaluate the inversion against TCCON, NDACC total column retrievals, and in situ WDCGG measurements to better understand the constraint from each measurement type used in the inversion. NDACC and WDCGG are independent data while TCCON are the same data as those used in the assimilation, so they are not independent for evaluating the joint inversions. Table 1 shows the measurement sites with their geographical information (latitude, longitude, and altitude above sea level). First, we evaluate the results against all the TCCON measurements from May to September 2023. In Fig. 7, the model is evaluated against hourly averaged TCCON data, with the a priori shown in blue and the a posteriori in red from either the TROPOMI-only inversions (in the left) or the joint TROPOMI and TCCON inversions (in the right). The statistics indicate that the inversions significantly improve on the prior for all the cases. Examination of the a priori models shows that the GBBEPx simulation slightly improves on the QFED simulation, with a coefficient of determination of  $R^2 = 0.31$  compared to  $R^2 = 0.27$  with QFED. The GFAS simulation has the highest a priori correlation ( $R^2 = 0.54$ ), resulting in the best a posteriori agreement with TCCON, with  $R^2 = 0.82$  and  $R^2 = 0.87$  for the TROPOMI-only and TROPOMI+TCCON assimilations, respectively. The time series plots in Fig. 7 show the improvement of GBBEPx on QFED, with better agreement between the simulation and the TCCON measurements when there are wildfire enhancements of XCO. The GFAS simulation shows a significant improvement during those peaks, indicating that the GFAS simulation better captures the time variability in the

measurements. The evaluation of the joint inversion using TROPOMI and TCCON XCO data against the TCCON measurements shows a further improvement of the inversions; both the slope and the  $R^2$  are closer to 1.0 than the results from the TROPOMI-only assimilation. This is expected because the same TCCON data are used in the inversion and the evaluation (Fig. 7d,h,l).

605

Figure 7: Evaluation of the model a priori (blue) and a posteriori from TROPOMI-only assimilation (red) against TCCON measurements (green) for all sites together using time series and scatter plots based on (a, b) QFED, (e, f) GBBEPx, and (i, j) GFAS biomass burning emissions (left side). A similar evaluation of the model a priori against TCCON, but with a posteriori from joint TROPOMI and TCCON assimilation (red) using (c, d) QFED, (g, h) GBBEPx, and (k, l) GFAS as the prior biomass burning emissions estimates (right side).

Figure 8: Evaluation of the time-averaged model a priori (blue), a posteriori using TROPOMI-only assimilation (orange), and a posteriori using joint TROPOMI and TCCON observations (red) against independent NDACC and in situ measurements between May-September 2023. Each panel is associated with a particular prior biomass burning emissions inventory, including (a, b) QFED, (c, d) GBBEPx, and (e, f) GFAS. The top row of each panel shows the coefficient of determination (R<sup>2</sup>) of the model using the prior or assimilation using the posterior with respect to the measurements, whereas the bottom row represents the mean bias and the standard deviation of the model or assimilation with respect to the measurements. Tsukuba, Lauder, and Wollongong are collocated stations for both TCCON and NDACC.






To assess the impact of TCCON on the performance of the inversion more objectively, we also compare the inversion results with independent NDACC column measurements and surface in situ measurements in Fig. 8. The R<sup>2</sup>, mean bias, and standard deviation relative to the measurements are shown for the model a priori (QFED, GBBEPx, and GFAS) in blue, the a posteriori from the TROPOMI-only assimilation in orange, and the a posteriori from the joint TROPOMI and TCCON assimilation (i.e., TROPOMI+TCCON) in red. We find a higher R<sup>2</sup> for both a posteriori estimates (i.e., TROPOMI-only and TROPOMI+TCCON) relative to the a priori estimates in almost all the cases, while there is an additional increase in correlation for the joint inversion compared to the TROPOMI-only inversion. The a posteriori from the TROPOMI-only assimilation provides a small reduction of the mean bias and standard deviation, and by adding TCCON to the assimilation, there is a further reduction at several sites (Tsukuba, St. Petersburg, MNM, RYO, YON). The added improvement in the posterior XCO obtained by adding the TCCON data to the inversion differs between sites. For the NDACC sites collocated or downwind of TCCON sites, such as Tsukuba, Lauder, Wollongong (collocated stations), and St. Petersburg (~2000 km downwind of European stations), the  $R^2$  increases more by adding TCCON to the inversion. The evaluation at the Arrival Heights NDACC station located in a remote area in Antarctica, far from both the TROPOMI and TCCON assimilated observations, shows an improvement in the a posteriori that suggests that the assimilation improves global background concentrations of CO. However, at the Altzomoni NDACC site, located about 75 km southeast of Mexico City and almost 2 km higher in altitude, we found little improvement after the assimilation. It is likely that the local topography cannot be captured in our global model, which has a 2° × 2.5° spatial resolution. In addition, the local ambient atmospheric conditions, such as stability and humidity in this region, cause most of the fire emissions to stay within the boundary layer, and neither the model nor the assimilation is capable of capturing such effects (Sha et al., 2021).

For the surface in situ measurements, we also find an increase in the correlation between the a priori and the a posteriori estimates using the TROPOMI+TCCON assimilation, but with a smaller improvement than observed at the NDACC stations, with the exception of the JFJ and MKN stations, where there is a larger improvement. The relatively smaller improvement compared to NDACC may primarily be attributed to the fact that the surface sites have a larger representativeness error given our  $2^{\circ} \times 2.5^{\circ}$  grid resolution, which limits the ability of the assimilation to significantly correct for them. For example, at the CGR station, the local atmosphere is influenced by a land-sea wind regime that cannot be resolved by the relatively coarse grid resolution of the model. In addition, the vertical sensitivity of the TCCON and NDACC data based on their averaging kernels may partly impact these evaluations.

Similar to the evaluation against NDACC, we find slight improvements in the mean bias and standard deviation at WDCGG surface stations. In most cases, adding TCCON to the assimilation reduces the error standard deviations (i.e., posterior—measurement errors), while the mean bias remains almost identical to that of the TROPOMI-only inversion. Comparison among the three BB cases shown in Fig. 8 indicates that, for equivalent a priori or a posteriori emissions estimates (e.g., for TROPOMI-only), their statistics are not significantly different at most measurement sites.

#### 4.2 North America analysis





#### 4.2.1 Assimilation performance for constraining boreal wildfires emissions

Our posterior emissions from the inversions indicated that North America has the highest level of BB emissions and contributed one-third of the global total in summer 2023. The emissions primarily came from the boreal forest across Canada, which were poorly estimated by the bottom-up emissions inventories, with a 31%–67% underestimation in the a priori relative to the a posteriori from both TROPOMI-only and joint inversions in this region during the study period (see Section 4.1 and Table 3). Thus, we take a closer look at the spatial and temporal characteristics of the XCO over North America to better understand the localized and episodic behavior of the fire emissions with respect to the different inventories. In addition to the three global BB emissions inventories discussed in the previous sections, here, we also include a regional bottom-up emissions inventory in North America from CFFEPS (Chen et al., 2019) provided by ECCC to compare with those global emissions. Note that for our global simulation with CFFEPS emissions in North America, we use GFAS for the global emissions which are replaced by CFFEPS in North America.

Figure 9: Evaluation of the domain-averaged CO concentrations (XCO) of the a priori model (dashed lines) and a posteriori using TROPOMI assimilations (solid lines) for four different inventories in North America, including QFED (blue), GBBEPx (green), GFAS (red), and CFFEPS (purple), against TROPOMI XCO measurements (black). The a priori model refers to a model forecast using prior emissions and the a posteriori is equivalent to the ensemble mean from LETKF. Three extreme wildfire episodes across boreal regions are chosen for comparison between the assimilation results using different inventories and for comparison of the assimilation with the a priori model.

We first focus on the temporal variability of the domain-averaged XCO in North America from the a priori model and a posteriori estimate using TROPOMI inversion with the four inventories. In Fig. 9, the model a priori for each emission inventory is shown in dashed lines with an 'x' marker, the a posteriori is shown with solid lines with a square marker, and the TROPOMI measurements themselves are indicated by the black line with circles. Fig. 9 shows that the a priori using QFED

and GBBEPx emissions have similar XCO in the entire period, except a slightly higher level of XCO with GBBEPx during May and June. On the other hand, the a priori XCO estimates with GFAS and CFFEPS are both greater in magnitude (~10 ppb higher) and more variable than those with QFED and GBBEPx. Although, the model-estimated CO shows similar trends between CFFEPS and GFAS, the two inventories produce XCO with different temporal variability and both underestimate the XCO observed by TROPOMI. In fact, during the large emissions episodes from wildfires in mid-May and late June, CFFEPS has higher emissions and better captures the variability in TROPOMI XCO, whereas from July to September, they produce comparable levels and variability of CO, with slightly higher CO for GFAS at the peaks in late July and early August.

Nevertheless, the inversion using TROPOMI suggests that all the a posteriori emission estimates are a significant improvement from the a priori after about a month, and show reasonable agreement with the temporal variability of the TROPOMI measurements. The a posteriori XCO also suggests that the seasonal variability that is usually characterized by decreasing CO in summer, due to the higher rate of oxidation with OH radical, is balanced by the higher rate of BB CO emissions estimated in the inversions, resulting in an almost uniform XCO during the summer-fall 2023. Despite their poorer a priori estimates, QFED and GBBEPx provide a posteriori XCO that agrees with TROPOMI measurements better than the a priori of GFAS and CFFEPS. However, comparing the XCO between the four inversions suggests that CFFEPS, followed by GFAS, perform better at the XCO peaks, and thus can better capture the variability in the TROPOMI measurements.






We looked closely at three extreme wildfire episodes that occurred across the Canadian boreal forest at different times and regions in summer 2023 to better examine the spatial characteristics of the a priori and a posteriori estimates at the time of the fires. As shown in Fig. 9, the first episode covers five days of large wildfires in Alberta between 19<sup>th</sup> and 23<sup>rd</sup> of May, the second episode occurred in Nova Scotia and Quebec between the 22<sup>nd</sup> and 26<sup>th</sup> of June, and the third episode was in British Columbia and the Northwest Territories between the 17th and 19th of July. To evaluate the inversion results from these events, we compare in Fig. 10 the model a priori (M) and a posteriori analysis (A) with the TROPOMI observations to obtain analysis minus observations (A – O) and model minus observations (M – O) differences for the four emissions inventories during the three extreme wildfire episodes in North America. A comparison of M – O between the different inventories for all the episodes reveals that CFFEPS, followed by GFAS, has a smaller underestimation of CO concentrations compared to QFED and GBBEPx. Although the reduction of this bias occurred over a large domain, including downwind of the emissions, the reduction is more significant in the vicinity of the wildfire emissions. Our results comparing different inventories are consistent with our findings in Fig. 9, which show that CO concentrations from the a priori are underestimated due to the lower emissions in the inventories (Fig. 4), in the same order as observed here. We find similar improvements in our a posteriori analysis between the different inventories, in which A – O exhibits lower bias with CFFEPS, followed by GFAS, in comparison with OFED and GBBEPx. Although the CFFEPS a posteriori XCO is significantly closer to the TROPOMI observation, the OFED a posteriori still shows a slight improvement on the CFFEPS a priori, indicating the larger impact of assimilating TROPOMI observations compared to providing a better prior.

**Figure 10:** Comparison of the difference between model (M) or assimilation (A) and TROPOMI XCO observations (O) (i.e., M – O or A – O), for three extreme wildfire episodes across boreal regions. Episode 1: May 19–23; episode 2: June 22–26; episode 3: July 17–19. Assimilation is based on TROPOMI observations and uses different inventories in North America as prior CO emission estimates. The model and assimilation fields were transformed using the TROPOMI a priori profiles and averaging kernels.

Figure 11: Taylor diagram for evaluation of the assimilation and a priori model using four different biomass burning inventories against TCCON XCO measurements at East Trout Lake (ETL) between May and September 2023. (a) Assimilation is performed using TROPOMI-only (square) and joint TROPOMI and TCCON (triangle) data, while their correlations and standard deviations are compared with the model a priori (coloured circles) and TCCON XCO measurements (black circle). (b) Evaluation of the mean bias and error of the priors (blue), assimilation using TROPOMI-only data (green), and assimilation using joint TROPOMI and TCCON data (red).


We also evaluated the a priori and a posteriori XCO with TCCON data from ETL for the entire simulation period between May and September 2023. Fig. 11a presents a Taylor diagram comparing the standard deviation and correlation of the a priori or a posteriori against TCCON measurements at ETL, and Fig. 11b shows the mean bias and standard deviation of the prior/posterior — measurement residuals. The a priori XCO are shown in circles, the a posteriori from the TROPOMI-only assimilation in squares, and the a posteriori from the joint TROPOMI+TCCON assimilation in triangles. It shows that the a priori estimates with QFED and GBBEPx have not only low correlation with the measurements, but also low variability. The a priori estimates with CFFEPS and GFAS improve on the correlation but more significantly on the variability, in addition to the mean bias (Fig. 11b) that is reduced by more than a factor of 2. The a posteriori for all cases provides significant

improvements on the a priori by increasing the correlation and lowering the mean bias and standard deviation, resulting in closer estimates to the measurements. Among all the a posteriori cases, the joint TROPOMI and TCCON inversion has a noticeable level of improvement with increased correlation and slightly smaller mean bias and standard deviation, in addition to adjusting the variability towards the measurements variability. A comparison among the inventories suggests that the joint inversion using CFFEPS provides the highest correlation, lowest standard deviation, and nearly unbiased estimates with variability matching the measurements. The best agreement between the joint inversion and the TCCON measurements at ETL is with CFFEPS, followed by GFAS, then GBBEPx, and QFED.

# 4.2.2 Implications for vertical sensitivities in the inversion

In this section, we examine the potential use of the experimental TCCON XCO product from the mid-infrared (InSb) detector available at ETL for the inversions of BB CO emissions. Since the measurements provide us with an independent set of XCO with distinct averaging kernels, we take those as separate pieces of information into our joint TROPOMI and TCCON inversion. Similar to the standard TCCON, we assume uncorrelated errors for the InSb data, however, the effect of possible error correlations can be approximated and taken into account by re-tuning the regularization factor. The InSb CO measurements are processed in a similar way to produce a separate set of hourly gridded TCCON InSb data. Thus, when these data are added to the joint inversion, the size of the observation vector and observation error covariance increases accordingly. The vertical profile of the averaging kernels in the InSb CO product (Fig. 1d) has higher sensitivity to the surface and lower troposphere, and lower sensitivity to higher altitudes compared with the standard XCO. Thus, we aim to understand the added benefit of assimilating these data for constraining CO emissions in the inversion. To achieve this, we conducted inversions using joint TROPOMI and TCCON data that incorporate three variations of the TCCON product, including the standard XCO, the InSb XCO, and the combined InSb and standard TCCON product. The inversions show nearly identical improvements between the posteriors and the prior when using the CFFEPS inventory, indicating a less than 1% discrepancy of total BB emissions in North America. However, there is a noticeable difference in their spatial distributions, especially in the wildfire hotspots in British Columbia, Alberta, and Quebec (see Fig. S3 in the supplements).

**Figure 12:** Evaluation of the assimilation and a priori model using the CFFEPS biomass burning inventory against tall tower measurements at ~60 m above sea level at East Trout Lake (ETL). (a-b) a priori model (blue), (c-d) assimilation using joint TROPOMI and standard TCCON data (red), (e-f) assimilation using joint TROPOMI and InSb TCCON data (yellow), and (g-h) assimilation using joint TROPOMI and the standard and InSb TCCON data (orange) are compared with tall tower measurements (green).


To evaluate the performance of these inversions, we use two independent in situ datasets: tall tower measurements from Environment and Climate Change Canada (Chen et al., 2014) and aircraft profiles from the National Oceanic and Atmospheric Administration (McKain et al., 2024; https://gml.noaa.gov/aftp/data/trace\_gases/co/pfp/aircraft/) at ETL. We compare the tall tower measurements with the a priori (blue) in Fig. 12a,b and the a posteriori (red) from the joint TROPOMI and TCCON assimilation using standard TCCON in Fig. 12c,d, the a posteriori (yellow) from the joint assimilation using the TROPOMI and InSb TCCON data in Fig. 12e,f, and the a posteriori (orange) from the combined TROPOMI plus the standard and InSb TCCON data in Fig. 12g,h. We find a significant improvement in all inversion cases compared to the a priori, especially at the concentration peaks, resulting in an increase in  $R^2$  and the slope of the regression.

Figure 13: Evaluation of vertical profiles from the joint inversion using TCCON standard CO (red), InSb CO (yellow), combined standard and InSb CO (orange), and the a priori model (blue) against aircraft measurement profiles from NOAA at ETL (green squares) on (a) July 7, (b) July 16, (c) August 13, and (d) August 20, 2023.

The evaluation of the inversions shows that the assimilation with TROPOMI and the InSb XCO data improves the correlation from the case with standard TCCON XCO (from  $R^2 = 0.68$  to  $R^2 = 0.78$ ), although the slope of the regression has slightly decreased (from slope = 0.72 to slope = 0.68). However, the inversion using the combined standard and InSb TCCON product improves on both correlation and the slope of the regression (slope = 0.73,  $R^2 = 0.77$ ) with respect to the inversion using the standard TCCON. This suggests that the inversion using the InSb XCO better captures the variability of CO near the surface, likely associated with the greater sensitivity of these data to lower altitudes, which improves the sensitivity of the data to surface emissions in localized regions with short-range transport. The slightly lower slope is likely due to the greater level of underestimation of CO at the peak concentrations compared to the standard TCCON inversion. This might be because the lower sensitivity of the InSb XCO to the mid-troposphere than the standard CO, could reduce the measurement sensitivity to CO plumes at higher altitudes or in the background, which is normally captured through longer range transport. However, the combined standard TCCON and InSb CO assimilation captures the variability slightly better than the standard

TCCON CO assimilation, with improved correlations relative to independent data, and also provides us with more representative estimates of the background CO. Thus, adding the InSb XCO dataset potentially benefits the inversion by providing a better constraint on the surface BB CO emissions.

Furthermore, an evaluation of the vertical profiles of CO from the a priori and a posteriori simulations against aircraft in situ measurements by NOAA at ETL in Fig. 13 shows that there is a consistent improvement with the joint inversion using different variations of TCCON data (the inversion using standard TCCON CO is shown in red, whereas the inversion using the InSb TCCON CO is shown in yellow), and the combined standard and InSb TCCON CO (in orange). Note that through the inversion process, we update only CO emissions, without directly updating concentrations. The results indicate that replacing the standard TCCON with the InSb product improves the agreement with the measurements at lower altitudes (1-2.5 km), while at higher altitudes (2.5-3.5 km) the standard TCCON assimilation performs slightly better. Despite the fact that a perfect constraint on vertical profiles cannot be obtained by assimilating only total column measurements, due to the limited vertical sensitivity, using both the standard and InSb CO data together in the assimilation maintains a balanced and reasonable agreement with the independent measurements at both lower and higher altitudes. This suggests that using all the TCCON standard and InSb CO measurements in the inversion provides an improved constraint on the fire plumes at a broader range of altitudes. This is likely associated with uniformly larger sensitivities with altitude compared to the inversion using each of the TCCON XCO datasets individually. Note that we have not found a similar level of improvement from adding the ETL InSb XCO dataset to our inversions when we evaluate against aircraft in situ data at the Park Falls TCCON station (not shown), which is about 1700 km to the southeast of ETL. This suggests that, although adding the InSb XCO to the inversion benefits the inversion results, it has a more local effect and may not provide a substantial additional constraint on the regional/global scale. Therefore, providing the InSb product at other TCCON locations is recommended for a better constraint on the emissions on larger scales.

# 5 Summary and conclusions







We used total column measurements from the TROPOMI satellite and the TCCON ground network to infer CO biomass burning emissions during the extreme North American fire season between May and September 2023. Using the CHEEREIO toolkit, we optimized CO emissions globally at a 2° × 2.5° grid resolution every 3 days. One objective of this work is to better understand the influence of the TCCON measurements in providing additional constraints for quantifying CO emissions through a joint TCCON and TROPOMI inversion. Despite the limited spatial coverage, TCCON has substantially more observations in time with high accuracy on column-averaged dry mole fraction measurements. This motivates the evaluation of the joint inversion in comparison with the TROPOMI-only inversion to constrain emissions from localized and episodic wildfires. A second objective is to evaluate the global QFED, GBBEPx, and GFAS a priori BB emission inventories, as well as the regional North American CFFEPS emissions, and to assess their impact on the inversion analyses.

All of the inversion results indicate that the priors significantly underestimate the BB CO emissions. Based only on TROPOMI observations, the global posterior emission estimates for QFED, GBBEPx, and GFAS are 302.9±67, 309.6±67, and 328.6±67 Tg CO, compared to prior estimates of 164.5, 182.6, and 230.3 Tg CO, respectively. For North America, the posterior emissions for QFED, GBBEPx, GFAS, and CFFEPS are greater than the priors by a factor of 3.0, 2.6, 1.4, and 1.4, respectively. Adding TCCON through a joint inversion with TROPOMI makes little difference to the global total and regional estimates (

Figure A1: Two OSSEs within the twin experiments that start with the a priori of -50% CO emissions with respect to the true emissions as shown in (a) prior – true and (c) posterior – true emissions; and with the a priori of +50% CO emissions with respect to the true emissions as shown in (b) prior – true and (d) posterior – true emissions; time series of CO emissions in the a priori (blue), a posteriori (red), and true (green) for the OSSE with (e) –50% CO emissions and (f) +50% CO emissions.

875

880

Each OSSE setup involves multiple inversion runs. A "nature run" (without observation assimilation) is conducted to generate the "true" state of the concentration fields using GEOS-Chem with unperturbed emissions—the a priori emissions in the inversion with real observations. The true state is then mapped into the observation space by an observation operator, generating simulated observations that include added observation errors under a perfect model transport assumption. A set of "control runs" assimilates these simulated observations. Each control run may vary in terms of perturbations in the magnitude of the emissions and/or the assimilation parameter range, depending on the experiment's objectives. In our first OSSE, we use a combined set of simulated TROPOMI and TCCON observations and apply observation errors proportional to the retrieval errors, based on the ratio of simulated to retrieval XCO. Emissions in the control runs are perturbed by  $\pm 50\%$  to evaluate the system's performance in recovering the true emissions.

Fig. A1 shows that the posteriors (Fig. A1 c,d) effectively capture and recover the spatial context of the CO emissions globally for both control runs, which have ±50% CO emissions in the prior (Fig. A1 a,b). Additionally, the time series of total CO emissions reveal that approximately 1.5 months (from the start of the assimilation) are required to constrain the magnitude and temporal variability of the global CO emissions (Fig. A1 e,f). In other experiments (not shown) where perturbations are applied only to major source regions (Fig. 2), we observe similar behaviour, even though the convergence rate for recovering true emissions slightly varies. For example, emissions are recovered relatively faster in North America (within ~3 weeks), followed by Europe (~4 weeks), likely due to a higher density of TCCON observations in these regions. This trend holds despite the nearly globally uniform spatiotemporal distribution of quality-filtered TROPOMI observations (i.e., super observations). In this experiment, we employ a set of previously optimized LETKF parameters and error statistics for background and observation errors, described below in Appendix B.

## Appendix B: Comparing OSSEs with TROPOMI-only and TCCON-only inversions against joint TCCON and TROPOMI inversion

Similar OSSEs to those in Appendix A are conducted, using either only TROPOMI or only TCCON observations, with prior emissions perturbed by -50% in one region at a time, as shown in Table B1. We compute the mean bias and standard deviation of OmF, along with the convergence time from the start of assimilation to recover the true emissions. Here, the forecast represents a model field, driven by either a priori or a posteriori emissions, mapped into the observation space. For the TROPOMI-only and TCCON-only inversions, we observe that biases and standard deviations generally increase across most cases; however, the inversion remains capable of recovering the true emissions. Compared to the joint TROPOMI and TCCON inversion, the convergence rate for the TROPOMI-only inversion slows down by up to a factor of two, depending on the perturbed region in the inversion. In contrast, convergence for the TCCON-only inversion can vary significantly. Specifically, the time to recover true emissions extends to 3 months in North America and 4 months in Europe, likely due to the lower spatial coverage of TCCON observations in these areas compared to TROPOMI. Additionally, we find that the TCCON-only inversion requires considerably more time to recover the true emissions globally and, especially, in Southern Hemisphere regions, such as Africa. The delay is primarily due to the limited number of TCCON sites in the Southern Hemisphere—only two sites are available for this study—and the extended time of inter-hemispheric exchange of air, which takes about 1 year (Jacob, 1999). As a result, sufficient information to constrain emissions in the Southern Hemisphere may not be achievable, especially within the limited 5-month period of inversion in this study.

**Table B1:** Mean bias and standard deviation of OmF for prior, posterior using TCCON-only, TROPOMI-only, and joint TROPOMI and TCCON (TROPOMI+TCCON) emissions. Each OSSE starts with -50% prior emissions in the specified region. Convergence times to recover the true emissions are shown from the start of assimilation.




| Region           | OSSE                | OmF <sup>a</sup> mean bias | OmF standard deviation | Convergence time |
|------------------|---------------------|----------------------------|------------------------|------------------|
|                  | (-50% CO emissions) | (ppb)                      | (ppb)                  | (month)          |
| Global           | Prior               | -14.2                      | 8.2                    | -                |
|                  | TCCON-only          | -2.5                       | 4.7                    | >6 <sup>b</sup>  |
|                  | TROPOMI-only        | 0.9                        | 4.1                    | 3.0              |
|                  | TROPOMI+TCCON       | -0.3                       | 3.5                    | 1.5              |
| North<br>America | Prior               | -7.9                       | 4.5                    | -                |
|                  | TCCON-only          | -0.5                       | 2.6                    | 3.0              |
|                  | TROPOMI-only        | 0.7                        | 2.8                    | 2.0              |
|                  | TROPOMI+TCCON       | -0.2                       | 2.1                    | 1.0              |
| Europe           | Prior               | -3.1                       | 4.7                    | -                |
|                  | TCCON-only          | -0.4                       | 2.1                    | 4.0              |
|                  | TROPOMI-only        | -0.5                       | 2.2                    | 2.5              |
|                  | TROPOMI+TCCON       | -0.3                       | 1.9                    | 1.5              |
| Africa           | Prior               | -6.2                       | 5.5                    | -                |
|                  | TCCON-only          | -3.7                       | 3.6                    | >6               |
|                  | TROPOMI-only        | 0.6                        | 2.3                    | 2.0              |
|                  | TROPOMI+TCCON       | 0.5                        | 2.0                    | 2.0              |

<sup>&</sup>lt;sup>a</sup> Forecast (F) in OmF uses a posteriori emissions for the assimilation run and a priori emissions for the prior (control) run.

To optimize the performance of the inversion, we employ OmF diagnostics to estimate key LETKF parameters. Specifically, we use the global mean bias and standard deviation of OmF over the full assimilation period to derive optimal values for parameters, such as the regularization factor  $\gamma$ , the inflation factor  $\Delta$ , and the localization radius (r). We also configure essential setup elements, like the assimilation spin-up time, the burn-in duration, and the ensemble size to support efficient system operation. Our analysis yields the following optimal values:  $\gamma_{TROPOMI} = 0.2$  and  $\gamma_{TCCON} = 5$ ,  $\Delta = 0.08$ , r = 500 km, with a minimum of three months for spin-up, one month for burn-in, and a minimum of 24 ensemble members—each chosen to minimize OmF statistics (e.g., see Fig. S4). Although we assume no transport modelling error, we apply an additional adjustment by inflating observation errors to offset this assumption. These configurations are consistently used across all OSSEs in this study.

b It means that the true emissions are not fully recovered,  $(|E_t^{\text{posterior}} - E_t^{\text{true}}|/E_t^{\text{true}}) \neq \delta$ , within 6 months of inversion ( $\delta = 2\%$ ).

Overall, our OSSEs results indicate that assimilating TCCON observations alone may not provide us with sufficient information to fully constrain the spatial context of CO emissions in regions in the Southern Hemisphere within the limited study period. However, in regions of the Northern Hemisphere, including North America and Europe, CO emissions are fully recovered within 2–3 months, likely due to the higher density of TCCON stations in these areas. In contrast, in the joint inversion, TROPOMI observations address the larger spatial biases, which typically exist in the model a priori, while TCCON measurements contribute finer constraints that enhance the representation of spatiotemporal variability.






925

Code and data availability. TROPOMI CO data can be downloaded from https://doi.org/10.5270/S5P-bj3nry0 (Copernicus Sentinel-5P, 2024). The individual TCCON GGG2020 datasets used in this publication are cited in Table 1, and the references are included in the reference list. The TCCON data are available at https://tccondata.org/2020 (last access: 6 July 2024; Total Carbon Column Observing Network (TCCON) Team, 2022). The NDACC data are obtained as part of the Network for the Detection of Atmospheric Composition Change (NDACC) and are publicly available (see https://wwwair.larc.nasa.gov/missions/ndacc/data.html; last access: 6 July, 2024). CO in situ measurements from WDCGG are available at http://ds.data.jma.go.jp/gmd/wdcgg/ (last access: 1 July 2024). In situ aircraft CO measurements from Global Monitoring Laboratory of the National Oceanic and Atmospheric Administration (NOAA) available are https://gml.noaa.gov/aftp/data/trace\_gases/co/pfp/aircraft/ (last access: 1 July 2024). In situ tall tower measurements at ETL provided by Environment and Climate Change Canada are available at https://gaw.kishou.go.jp/search/station#4007 (last access: 1 July 2024). GEOS-Chem version 14.1.1 source code is archived at https://doi.org/10.5281/zenodo.7696651 (The International GEOS-Chem User Community, 2023), and MERRA-2 meteorology input data can be downloaded from WashU data portal at http://geoschemdata.wustl.edu/ExtData/GEOS 2x2.5/MERRA2/ (last access: 1 July 2024). The QFED emissions (version 2.5, release 1) data can be accessed from http://geoschemdata.wustl.edu/ExtData/HEMCO/QFED/v2023-05/ (last access: 1 July 2024). The **GBBEP**x version emissions data available https://www.ospo.noaa.gov/pub/Blended/GBBEPx/ (last access: 1 July 2024). GFAS emissions (version 1.2) can be downloaded from https://ads.atmosphere.copernicus.eu/datasets/cams-global-fire-emissions-gfas?tab=overview (last access: 11 June 2024). The CFFEPS output is produced for ECCC's operational air quality forecast system (D. Kornic et al., 2024). The CFFEPS emissions code and the accompanying user manual are available at https://zenodo.org/records/2579383 (last access: 1 July 2024) (Anderson and cast of thousands, 2019). The CHEEREIO source code is available at https://github.com/drewpendergrass/CHEEREIO (last access: 1 July 2024) (Pendergrass et al., 2024) and is documented at https://cheereio.readthedocs.io (last access: 22 August 2024). A forked repository of CHEEREIO used in this study, which contains the TCCON CO and TROPOMI CO observation operators and the assimilation configuration, is available at https://github.com/Sinavo/CHEEREIO (last access: 22 August 2024).

955

Acknowledgements. We would like to thank the team behind the four BB emission inventories: QFED v2.5r1, GBBEPx v4, GFASv1.2, and CFFEPSv4, for making their data available. We thank the World Data Centre for Greenhouse Gases

(WDCGG) for providing their CO data. We acknowledge all members of the Japan Meteorological Agency for operating the trace gas measuring systems at Minamitorishima, Yonagunijima, and Ryori stations. We thank the NOAA Greenhouse Gas Reference Network team and pilot Jared Chursinoff for their contributions to the aircraft data. The FTIR monitoring program at Jungfraujoch is primarily supported by the F.R.S. - FNRS (Brussels, Belgium) and the GAW-CH program of MeteoSwiss (Zürich, CH). The Paris site has received funding from Sorbonne Université, the French research center CNRS and the French space agency CNES. Financial support from DGPAPA-UNAM (IN106024, G101225) was provided for the Mexican stations as well as the postdoctoral grant of Andrea Cadena-Caicedo. Tsukuba TCCON/NDACC and Rikubetsu TCCON sites are supported in part by the GOSAT series project. We thank the Canada Foundation for Innovation and NSERC for infrastructure and data analysis support for the TCCON station at ETL. Computations were performed on the Niagara supercomputer at the SciNet HPC Consortium. SciNet is funded by Innovation, Science and Economic Development Canada; the Digital Research Alliance of Canada; the Ontario Research Fund: Research Excellence; and the University of Toronto.

Financial support. The Government of Canada's Environmental Damages Fund provided funding for this project under its Climate Action and Awareness Fund. Funding was also provided by the NOAA Global Monitoring Laboratory through Contract 1305M323PNRMJ0742.

Author contributions. The study was designed by SV, DBAJ, and DWu. SV performed the data analysis and prepared the original draft. DBAJ and DWu guided the work and edited the paper. DCP is the principal developer of CHEEREIO and SV contributed to the code development of TCCON and TROPOMI CO observation operators. JC, KA, and RSt contributed to the CFFEPS emissions. SK, and AZ contributed to GBBEPx emissions. DWu, POW, DFP, IMo, HO, NMD, FH, RSu, DWe, RK, OG, YT provided TCCON data. IMo, MVM, NJ, EM, AC, IMu contributed to NDACC measurements used. KM provided aircraft measurements. DWo and SR provided tall tower measurements at East Trout Lake. PC, CL, EK, TS, MS, RM contributed to WDCGG measurements. All authors reviewed and commented on the paper.

Competing interests. The authors declare that they have no conflict of interest

Disclaimer. Views expressed are those of authors and do not reflect those of NOAA or Department of Commerce.

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
