# Peer review of "Quantifying CO emissions from boreal wildfires by assimilating TROPOMI and TCCON observations"

_EGUsphere, 2025_

## Referee Comment (RC3)

The manuscript titled "Quantifying CO emissions from boreal wildfires by assimilating TROPOMI and TCCON observations" by Voshtani et al. with reference egusphere-2025-858 is a highly valuable scientific contribution in atmospheric air quality modeling and emission inversion for wildfires. Overall, it is very well written and provides a complete investigation of the benefit of large-scale CO emission inversion using TCCON total column retrievals. The study utilizes the CHEEREIO data assimilation toolkit for CO emission inversion and considers different wildfire emission inventories during the period between May and September 2023. In the first part, the benefit of assimilating TCCON data in addition to TROPOMI retrievals is analyzed on a global scale with 3 different global wildfire emission inventories. The results are compared with the assimilated, as well as different types of independent observations. The second part focuses on North America where the regional CFFEPS emissions are also included in the comparison. This part also includes a comparison and evaluation of different TCCON retrievals of CO.

Overall, I suggest accepting the manuscript with the following minor corrections:

**General comments:**

A) My first general comment is on the length of the manuscript and its broad range of investigations. This is not a criticism, but rather reflects the extensive information content in this study. But still, I would at least suggest dividing the results in Sec.4 into two sections including the global and North America evaluation, respectively. You may even consider dividing the paper into 2 parts because especially the evaluation of different TCCON retrievals in Sec.4.5 appears to have a quite different purpose than Sec4.1-4.3 and thus addresses a different audience (which can also be seen e.g. in the summary). This would also allow for a more detailed description and evaluation of the different TCCON retrieval data in Sec.4.5 (see related smaller comments below).

B) I would also highly appreciate a short introductory paragraph at the beginning of Sec.2 and 4 to guide the reader and provide an overview of the content of each subsection.
   ○ For Sec.2: It would be helpful to already provide the information which observations are actually used for assimilation and validation in the section overview. Otherwise the amount of different observations might be confusing. This could maybe also be included in the title of the subsections. It is also not very clear why the aircraft data are presented in Sec.2.4 while the other in situ data are in Sec.2.3.
   ○ For Sec.4: Please also describe that the global evaluation in Sec.4.1-4.3 includes the 3 global emission inventories, while the regional evaluation in Sec.4.4 and 4.5 additionally includes CFFEPS (and see general comment A above)

C) Thought the results, there are several locations where the word "uncertainty" or "error" is used without a clear specification on the actual measure. Since different measures are used in the evaluation, it is important so use specific wording.
   ○ l.53: how is the uncertainty quantified? Eg ensemble spread or difference to independent data?
   ○ l.358: How is the "a posteriori" uncertainty estimated? from LETKF analysis scheme? According to the table description for Tab.3, it is the a postriori ensemble standard deviation. Please specify in the text.
   ○ l.400: (as above?): Please define in the text how the uncertainties are estimated, ie as posterior ensemble spread after the assimilation as stated in the description of Fig.3? This is particularly important to interpret the large reduction of uncertainty when additionally assimilating TCCON compared to a small change in the posterior emission estimates themself (on a global scale). Please also define how the global and regional

uncertainty estimated are calculated from the spatially resolved ensemble spread (or equivalent).

- l.586: The word "uncertainties" is unspecific in this case. Are you referring to correlations, biases? Or to the investigation of uncertainties as given by the ensemble spread in Sec.4.2? Please formulate more specific.
- l.664, l.666, l.667: please specify "uncertainty" in terms of the quantities that are given in Fig.11 or state where this statement is coming from elsewhere.
-

D) From my knowledge figure descriptions like color-coding should only be given in the figure caption because it's hindering the reading flow in the text (eg. l.560ff, l.606ff, l.630ff, l.658ff, l.692ff, l.717ff, …, please check also other locations, especially before)

**Smaller comments:**
1. Abstract: Wildfires play an important role on a wide range of scales, from very local impacts on individual hotspots to global scale. Please state more clearly in the abstract that this study focuses on the large-scale.
2. Sec.2.4: can you please explain a bit more in detail how the aircraft data was sampled? Are the sensors mounted on commercial aircraft or was there a measurement campaign?
3. Sec.3: Please give a bit more details on the ensemble generation:
   - Are the emissions from all sources perturbed or BB-only?
   - How was the width of lognormal-distribution (ie perturbation variance) for the emission perturbations determined?
4. Sec.4.1, l.386-388: This suggests that you are including all global emissions in the comparison. Since your assimilated observations were limited to below 60°N (&S), excluding this region from the comparison would provide more isolated insights on the actual impact of assimilation. Did you look at this as well?
5. Sec.4.1: l.412f: I don't understand the importance of this sentence in the given context. Why should the a priori emissions and error estimates be different between the experiments?
6. Sec.4.1, l.427-429: Is this "improvement" w.r.t. to the posterior? Please state clearly right when mentioning the 1st time in L.427.
7. Sec.4.1, l.455-447: You mentioned the impact of TCCON already before (eg L.443). If the previous comparison refers to a priori vs joint a posteriori, please explicitly state so. From L.431ff, I was assuming you were comparing a priori vs TROPOMI a posteriori above. If this is the case, then I don't understand the explanation with few TCCON stations in L.443ff. Please clarify.
8. Sec.4.1, l.455ff: That's a reasonable conclusion, is it possible to verify that, e.g. with the OSSE where you only assimilate TCCON in Apx.B?
9. Sec.4.2: For understanding, I would suggest adding a note that the normalized error reduction (and maybe also the DOFS?) increases with increasing weight of observations compared to the model a priori, even for overfitting to the observations, and that a small error reduction could also be due to a good model a priori.
10. Sec.4.3: Does the validation use the same TCCON data as used in joint inversion? If so (as stated in L.539f) state explicitly right from the beginning (e.g. l.528) because it is an important information that the validation data is not independent for the joint inversion.
11. Sec.4.4: The title is rather long which makes it difficult to extract the specific content if this subsection: North America. Please shorten the tilte to stress it's specific purpose more clearly. (and see also general comments above)
12. Sec.4.4: Which timer period is used in the statistical evaluation inf Fig.11? The whole period or some of the episodes? Please explain in the figure caption and the main text.
13. Sec.4.5:

- Is the InSb retrieval really independent from the original retrieval, even when from same instrument? Without deeper knowledge about the retrievals: If its about different averaging kernels, it's probably more an "valuable" additional information rather than an completely independent measurement in a mathematical sense? Please explain.
- Could you give some details about the combined TCCON observations? Were the two products combined, and how, before using in the assimilation? Or were both used jointly in the assimilation? If so, how are where treated, e.g. are they assumed to be 2 completely uncorrelated datasets? And is this a reasonable assumption, and why?

14. Sec.4.5, l.701ff: To me, the differences appear to be quite small. I would suggest stressing this a bit more, e.g. by adding this information in the last sentence in l.712ff.
15. Sec.4.5, l.718ff: That's an interesting point. Did you look into how the different averaging kernels impact the a posterior emissions? Do they induce in different spatial or temporal distributions of emissions?

**Technical and formulation-related comments:**
- l. 216: "aircraft directly samples the atmosphere" Is there a word missing? Do you mean aircraft measurements (then "sample" in plural)? The aircraft itself does not sample.
- l.226 & 228: The word "driven" is used in two different contexts here.
  1. I'm not aware of using word for assimilated observations in a model. Suggest replacing here.
  2. Probably suitable word when referring to reanalysis, however formulation remains unclear. Is MERRA-2 used for initialization? Suggest more specific formulation.
- l.230-232: Please add a specific literature reference of the model description here.
- l.293, Eq.(1): tilde over analysis error covariance matrix was not defined
- l.301ff:
  - Please specify more clearly which variables are referring to emissions (ie in model space), and which to concentrations (ie in observation space), eg for apriori profile, simulated CO profile, …
  - the definition of M remains unclear: does it refer to the forecast model or emission-to-concentration mapping, or both? If it is mapped in observation space as denoted, shouldn't it include the operator H? Please clarify.
- l.306: ", and the observation and the prior error covariances": It unclear where is this part of the sentence referring to? Are the observation and error covariances also scaled based on the obs increments? Or are they also passed on the LETKF processor? Please specify.
- l.337: Spelling error? Do you mean "updated by"?
- Fig.4: The subfigures are quite small, it's especially hard to see influences of TCCON assimilation with a low spatial coverage
- l.481f: "more reliable" compared to what? The a priori?
- l.588: "difference "unclear: Do you refer to differences between TROPOMI-only and joint inversion or between the different prior emissions?
- l. 593: "underestimation" w.r.t. what? Please specify and refer to section above if applicable
- Fig.9: The in-figure legend is confusing: Is "ensemble mean" the a posieriori ensemble mean, and "model a priori" the a priori ensemble mean? From my understanding, both are ensemble means, right? Then it would be clearer to label "a priori" and "a posteriori" (for each emission model) in the legend and the information "ensemble mean" can be added in the figure caption.
- l.622: "even though the … ." What is the massage of this part of the sentence? You already compare with the a priori of GFAS and CFFEPS before. So they are closer to TROPOMI compared to what? Maybe the a priori of QFED and GBBEPx? Please reformulate and make the specific massage of the sentence more clear.
- Fig.12: in-figure labels are very small, hardly readable

---

## Author Comment (AC1)

**Response to Reviewers**

**Quantifying CO emissions from boreal wildfires by assimilating TROPOMI and TCCON observations**

We sincerely thank the referees for the time spent reviewing the manuscript and for their thoughtful comments that led to an improved manuscript. Our responses to all three referees and updates in the revised manuscript are listed below. Referee comments are in black and author responses are in blue. The line numbers in our responses refer to those in the revised manuscript.
* * *
**Referee 1 (RC1):**

This study optimized the CO emissions between May and September 2023 with the CHEEREIO toolkit. Various inversion experiments were conducted with global and regional priori emission data, TropOMI and TCCON observations and different vertical profiles of TCCON data. This study is well designed and the manuscript is in great organization. The findings could benefit future emission inversions. My comments are listed below:

1.   Fig 1(a) the TCCON observation is at hourly scale and the TROPOMI observation is at daily scale, thus please use separate color bar for plotting number of observations.

Thanks for the feedback. We updated Fig.1a with a separate color bar for number of TCCON observations.

[Figure]

**Figure 1:** (a) The number of quality controlled and aggregated TROPOMI and TCCON observations (XCO) used in the global inversion as described in sections 2.1 and 2.2. (b) The variability of the number of non-aggregated TROPOMI (blue) and TCCON observations (red) at the East Trout Lake (ETL) TCCON station from April-September 2023. (c) Temporal series (MM-DD) of XCO column retrievals of standard TCCON GGG2020 data (blue) and XCO measurements from InSb detector (orange) at East Trout Lake (ETL) between May-September 2023. (d) Column averaging kernels for the standard TCCON GGG2020 XCO (blue); column averaging kernels for XCO measurements derived from an alternative CO absorption window on the ETL InSb detector (orange) at East Trout Lake (ETL) between May-September 2023.

We also updated the caption of Fig.1 and the legend in Fig. 1a (see additional update 2 on page 26).

2. Fig 3 The posteriors from different priori showed observable differences in temporal variations. Please explain and evaluate these differences.

Thanks for this valuable comment. In Section 4.1, we primarily discussed the difference in temporal variability between the two types of posteriors (i.e., TROPOMI-only and joint TROPOMI+TCCON inversions) using the same prior. However, throughout the manuscript (e.g., lines 590-591, 605-606, 689-691, and 724-735), we also noted and explained differences in temporal variability of the posteriors driven by different priors but the same assimilated observations. As highlighted, the priors have different temporal variations, reflecting the different approaches used in constructing them, and this information can propagate into the a posteriori estimates. A detailed discussion of the causes of the discrepancies in the priors is beyond the scope of this work. However, an objective of the work is to assess the extent to which assimilating the satellite and TCCON data can correct for such discrepancies in the priors. We have clarified and expanded this discussion near Fig. 3 in the revised manuscript.

**Text added in Section 4.1.1** (lines 487-492):
"The posteriors from different priors, but with the same set of assimilated observations, show overall stronger temporal variability compared to their respective priors. For instance, in North America, while the QFED and GBBEPx priors are relatively flat compared to the GFAS prior, all their posteriors exhibit enhanced and more consistent variability (Fig. 3a,f,k). Still, differences to some extent remain noticeable between these posterior time series, likely driven by differences in the spatial and temporal distribution of the priors, and their interaction with observational constraints through model transport and mixing."

3. There are figures showing the differences between priori and posteriors, and between observations and priori/posteriors. These lack posteriors emission maps.

Thank you for this helpful comment. We added the posterior emission maps to the supplement of the revised manuscript and referenced them in the main text. The posterior emissions for QFED, GBBEPx, and GFAS (related to Fig. 4) are now shown in Fig. S2. The prior and posterior emissions for CFFEPS in North America (related to Fig. 10) were already provided in the supplements (now in Fig. S3).

**Text added in Section 4.1.1** (lines 498-499):
"(see Table 3 for a comparison of total emissions and Fig. S2 in the supplements for the separate a posteriori emission maps associated with Fig. 4)"

[Figure]

Figure S2: Comparison of the spatial distribution of the time-averaged biomass burning CO emissions in the (a-c) a posteriori using TROPOMI-only assimilation, and (d-f) a posteriori using TROPOMI+TCCON assimilation, from three global biomass burning inventories of QFED, GBBEPx, and GFAS.
* * *
**Referee 2 (RC2):**

The paper by Voshtani et al. discusses the estimate of wildfire CO emissions on a global scale by assimilating both TROPOMI and TCCON observations. The paper is very well written, with high-quality graphics and a very comprehensive set of references. The comparison of the results for 3 (4) independent CO fire inventories provides insightful new results. The sparse TCCON network, in combination with TROPOMI XCO, is convincingly shown to provide a large impact on the emission estimates and a-posteriori emission uncertainty. As a consequence I am in support of publishing these results. I only have relatively few comments, ranked as minor, for which I would ask the authors to include their responses in the manuscript.

Minor comments:
1.   l 82: Is "bottom-up" the right term? FRP is retrieved from satellite observations, and not estimated from "activity data". For instance, GFAS stands for "Global Fire Assimilation System".

We appreciate the reviewer's comment, and we agree that the distinction between bottom-up and top-down can be confusing. In the emission inventory development community, the use of satellite-derived FRP to provide constraints on the emissions results in what is referred to as "top-down" emission estimates. However, from the perspective of the atmospheric chemistry modelling, solving the inverse problem to infer emissions from concentration measurements produces "top-down" emission estimates. In this context, the prior emissions used as input to the inversion system are typically referred to as "bottom-up" estimates, regardless of whether they are constructed from activity data or satellite-based FRP and as long as they are not derived from an inverse modeling approach. For consistency with the inverse modeling literature, we prefer to keep this nomenclature. We note that in the manuscript, the usage of "bottom-up" versus "top-down" emissions is explicitly specified in the context of inverse modelling, as explained in lines 78-87.

2. l 100: A major benefit of TROPOMI is it's high sensitivity for CO close to the surface, in contrast to e.g. MOPITT or IASI. The total column measured has a strong relation to the emission, which can be estimated using basically only wind information. Please emphasise this point.

Thank you for the suggestion. Near surface sensitivity is indeed an important aspect, and we have added the information to the revised manuscript as suggested highlighting the benefit over thermal infrared (TIR) measurements. It should be noted that MOPITT also provides shortwave infrared (SWIR) measurements, and the joint SWIR and TIR MOPITT retrievals offer good sensitivity to near surface CO.

**Text revised and added in Section 1** (lines 98-102)**:**
"The Tropospheric Monitoring Instrument (TROPOMI) (Borsdorff et al., 2018), launched in 2017, has provided CO retrievals with improved accuracy, higher spatial resolution, and significantly greater observational coverage (Landgraf et al., 2016; Schneising et al., 2020). In particular, it offers higher sensitivity to near-surface CO compared to earlier thermal infrared measurements from instruments such IASI and TES. These factors make it well-suited for inverse modelling of CO emissions, as demonstrated in many recent studies"

3. l 141: Why is XCO used as acronym of the "Total column abundances of carbon monoxide" ? Where does X come from (why not TCO or TCCO)?

The term "X" is commonly used in atmospheric measurement and retrieval studies to refer to the column-averaged dry-air mole fraction of a gas. It originates from the convention in remote sensing, where satellite-based instruments (e.g., TROPOMI) and ground-based spectrometers (e.g., TCCON) retrieve gas concentrations using spectroscopic techniques. Accordingly, "XCO" refers to the column-averaged dry-air mole fraction of carbon monoxide (CO). This terminology is widely used in the literature (Borsdorff et al., 2019; Kiel et al., 2016; Knapp et al., 2021; Schneising et al., 2023; Wunch et al., 2009, 2011, 2019; Zhou et al., 2019)

4. l 149: "quality flag equal to and greater than 0.7 to ensure high-quality data obtained under cloud-free or low cloud conditions." Fires produce smoke, which may look similar to clouds. I was wondering if part of the measurements close to the fires may not be included in the quality flag > 0.7 dataset. Does this filtering impact the inversions?

It is possible that thick smoke from intense fires may be misclassified as cloud in the cloud detection algorithm and filtered out, depending on parameters such as cloud height and optical thickness (Landgraf et al., 2016). In that case, the inversion will utilize measurements downwind of the plume to infer emissions from the fire itself. This is a powerful feature of the data assimilation approach that enables us to take advantage of the transport of information throughout the domain.

5. l 154: The GEOS-Chem model resolution used in this study (2 x 2.5) is relatively coarse. Does this resolution influence the BB emission estimate?

That is true. Depending on the scale of the assimilation problem, model resolution can influence emission estimates. In this study, we present a global-scale inversion, focusing on large-scale wildfire emissions across multiple regions over a five-month period (Fig. 2 and Table 3). At this scale, the $2° \times 2.5°$ resolution used in GEOS-Chem is widely used as a standard in global inversion studies (Byrne et al., 2024a; Jiang et al., 2011; Jones et al., 2009; Pendergrass et al., 2023; Qu et al., 2021; Stanevich et al., 2020, 2021; Wunch et al., 2019). While even coarser resolutions (e.g., $4° \times 5°$) at the global scale have been frequently used in the past. A finer-resolution, nested regional inversion (e.g., $0.25° \times 0.3125°$) would likely perform better at resolving local fire episodes, assuming sufficient observational coverage. However, such high-resolution regional assimilation was outside the scope of this study, which focuses on

global-scale CO constraints from multiple observing networks and using multiple priors. Future work could more systematically assess the impact of model resolution by comparing inversions at global, regional, and local scales.

6.  l 157: super-observations "we average the observations, weighted based on their reported retrieval errors". l 158: "The super-observation errors in each grid are also obtained by averaging the reported retrieval errors combined with the error correlation between measurements following Pendergrass et al. (2023)" The way the construction of the super-observations is described here seems to contradict Pendergrass 2023, section 3.3. If I understand correctly observations are averaged, and not weighted, in this paper. Please provide the details on how the weights are constructed (the correlation factor is mentioned later). Is the weight inversely proportional to the error^2? This is important to appreciate the overfitting and gamma factor discussed later.

Thanks for the feedback. For each model grid cell, the aggregated TROPOMI observation (i.e., super-observations) is computed using an error-weighted median average, rather than a simple or unweighted mean or median. Specifically, each individual retrieval within a grid cell is assigned a weight inversely proportional to its reported retrieval error standard deviation (i.e., $w_i = \frac{1}{\sigma_i}$), so that observations with lower reported error contribute more to the aggregated value. This approach differs slightly from Pendergrass et al. (2023) that use a different averaging scheme. However, for the error associated with the super-observation, we follow a similar formulation that accounts for both the individual retrieval errors ($\sigma_i$) and their potential error correlation ($c$):

$$\sigma_{super\_observation} = \sqrt{\left[(\frac{1}{n}\sum_{i=1}^{n}\sigma_i)(\frac{1-c}{n}+c)\right]^2 + \sigma_{transport}^2}$$

with two distinctions: 1) transport error ($\sigma_{transport}$) is assumed to be zero, and 2) the error correlation, $c$=0.28, is adopted from empirical value used in TROPOMI methane inversions (Chen et al., 2023; Pendergrass et al., 2023) but its effects on the assimilation are revised using regularization parameters. This choice avoids assuming zero correlation and allows us to account for some level of spatial correlation in retrieval errors for the same type of instrument. In fact, to compensate for potential bias or sub-optimality due to these assumptions, we perform additional error tuning in the inversion by introducing a regularization factor $\gamma$ in Eq.1, which adjusts the weight of observation errors relative to prior errors. This tuning is performed using OSSEs, as described in Section 3 (lines 363–378) and Appendices A and B

It is important to note that assumptions on $c$ and $\sigma_{transport}$ do not directly affect the inversion results. Instead, they influence the initial conditions for estimating $\gamma$. For example, using a different assumption for $c$ and $\sigma_{transport}$ would lead to a different optimal value of $\gamma$, which appropriately rescales the observation error and ideally results in identical posterior emissions (please see our response to Comment 8 for details about regularization factor $\gamma$). This ensures that the inversion is neither overfit nor under-regularized with respect to the actual error structure. We have clarified this in the revised manuscript:

**Text revised and added in Section 2.1** (lines 168-175):
"In fact, for the duration of the study between May and September 2023, we compute the error-weighted median average of measurements within each grid cell, where each measurement is weighted by the inverse of its reported error standard deviation (Eskes et al., 2003; Miyazaki et al., 2012). To account for error reduction because of averaging, we follow a similar method as Pendergrass et al. (2023) to compute the associated super-observation errors. This includes an average of individual measurement errors and assumptions on error correlations and transport errors. The relative weight of the super-observation error to the prior error is then estimated through parameter tuning in the inversion system to ensure robustness to possible error misspecification (see Section 3 and Appendices A and B)."

7.  Fig. 1, panel a: What is the time period for collecting the 1.7 million TROPOMI super-observations? Is it also May-September 2023?

Yes, the 1.7 million TROPOMI super-observations shown in Fig. 1 are collected over the entire assimilation period from May to September 2023. This information is also stated in the figure caption.

8.  p12, eq 1: What does the gamma factor represent? (p13 mentions the gamma used) Why is it needed?

The gamma factor $\gamma$, also referred to as regularization parameter, balances the relative contribution of the a priori emission estimate and the observations in the a posteriori estimate, based on their respective error covariances (Eq.1). More specifically, in the cost function used in the Bayesian or Kalman filter-based inversion (Hunt et al., 2007), $\gamma$ adjusts the weighting between the a priori term and the observational terms. This parameter is commonly used in 4D-Var and analytical inversion frameworks (Hakami et al., 2005; Lu et al., 2022) to prevent overfitting or underfitting to the observations. It serves as a pragmatic correction for uncertainties that are hard to quantify, such as unaccounted error correlations in the observational error characterization (see also our response to Comment 6).
In this study, $\gamma$ estimated through a series of OSSEs (i.e., identical twin experiments), as described in Section 3 (lines 364-369) and Appendix B. The objective of this tuning is to select a $\gamma$ value that minimizes the mismatch between model forecasts (based on posterior emissions) and the observations. This approach ensures the inversion system is neither under- nor over-constrained by observational data. Similar $\gamma$-tuning methods have been used in other inversion studies (Chen et al., 2021; Varon et al., 2022; Voshtani et al., 2023). We have clarified the role of $\gamma$ in the revised manuscript.

**Text revised and added in Section 3** (lines 316-320):
"$Y^b$ is the observation perturbation matrix (see the detail description of LETKF variables in Hunt et al. 2007). $\gamma$ is a regularization factor used to prevent overfitting or underfitting to observations by balancing the relative influence of the a priori estimate and the measurements in the inversion. It serves as a pragmatic correction for uncertainties that are hard to quantify, such as unaccounted observation error correlations (Hakami et al., 2005; Lu et al., 2022; Voshtani et al., 2023)."

9.  l 314: Is this an OSSE? For me the term OSSE refers an assessment of the impact of a future (satellite) observational dataset on an existing analysis system. It is typically using two independent models/systems where one is generating the Nature run and synthetic observations and the other performs the analysis.

We acknowledge the reviewer's definition of OSSE as a tool primarily used to assess the potential value of future observations in an existing data assimilation system. However, OSSEs are also widely used in a broader context, beyond the evaluation of future missions. As discussed in Lahoz and Schneider (2014), OSSEs serve as a standard process for assessing the influence of observations and (assimilation) system configurations in a controlled environment, even when existing (rather than future) observation types are used. For example, Stanevich et al. (2021) used OSSEs to evaluate the capability of new assimilation developments (based on Weak-Constrained 4D-Var) with existing GOSAT satellite observations to reduce model errors in $CH_4$. Voshtani et al. (2023) employed OSSEs to explore the impact of error covariances in a coupled source-state estimation (based on coupling 4D-Var inversion and parametric Kalman filter assimilation) using existing GOSAT methane observations. Byrne et al. (2024b) used OSSEs with existing in situ and OCO-2 data to assess the effect of observational coverage gaps on detecting extreme-event-driven $CO_2$ flux anomalies.
In this study, we conducted a set of identical twin experiments as simplified OSSEs, using synthetic observations generated from a Nature run within the same model framework, to configure and assess the assimilation of TROPOMI CO and TCCON CO data. While the observations are existing datasets, the observation operators used to assimilate them are newly developed in this study as part of the CHEEREIO system. This setup enables systematic evaluation and tuning of key assimilation parameters, such as observation error inflation and the regularization factor ($\gamma$), across different observation types and a priori emissions estimates. By conducting these experiments in a controlled

environment where the true emissions are known, we can optimize the inversion system before applying it to real-world observations. Appendix A and B and our response below to Comment 20 provide an example of how we applied OSSEs to estimate parameters, which is based on a brute force minimization of the mismatch between model forecasts and the observations (OmF).

10. l 338: "We set a localization radius of 500 km" Given the model resolution of 2 x 2.5 degree, this is only two grid cells. Also, for CO with a long lifetime, the plumes (emission and concentration location spatial distances) remain visible over thousands of km. Could you please discuss this in some more detail. Were tests performed with longer correlation lengths?

Thanks for the insightful comment. We agree that CO plumes can be transported over long distance, and thus we should expect that satellite observations can constrain emissions over a wider spatial extent.
We apply a localization radius of 500 km, following previous CO inversion studies (Gaubert et al., 2023; Miyazaki et al., 2012, 2015) that used a same range of distance. This choice is primarily motivated by the need to reduce spurious long-range correlations that arise due to the limited ensemble size in the LETKF framework (Hunt et al., 2007). We also tested smaller and larger values of localization radius (i.e., 250 km, 1000 km, 1500 km) using simple OSSEs with −50% and +50% emissions in the prior relative to the true emissions. We added Fig. S1 to the supplements. It compares observation minus forecasts (OmF) mean and standard deviations between these cases and shows that the system with a shorter localization distance (250 km) underperformed (i.e., larger OmF mean and standard deviations) while for a longer distance (1000 km and 1500km), mean bias results were nearly identical to the 500 km case, but standard deviations is slightly increasing. This might be due to the effect of spurious correlations and when the system starts becoming less stable. Accordingly, 500 km is a reasonable compromise, which is broad enough to capture local influence while minimizing the risk of spurious long-range correlation and ensuring robust performance of the inversion. We also added this information to the revised manuscript:

**Text revised and added in Section 3** (lines 370-372)
"We set a localization radius of 500 km following previous CO inversion studies (Gaubert et al., 2023; Miyazaki et al., 2015) to avoid the impact of distant observations, which may be impacted by sampling errors and spurious correlations (see Fig. S1)"

[Figure]

Figure S1: Comparison of OmF between OSSEs with different localization distances used in the inversion. Using (a) −50% prior emissions relative to the true emissions and (b) +50% prior emissions relative to true emissions, based on TROPOMI+TCCON inversion.

11. l 340: The correlation factor of 0.28 was reported for CH4, while the present study is for CO. Please comment.

As noted earlier, the super-observation errors are refined through an estimation of the regularization factor $\gamma$. Please see our response to Comments 6 and 8.

In this study, instead of directly estimating the observation error correlation factor $c$ for CO, we adopted a value of 0.28 reported for TROPOMI $CH_4$ (Chen et al., 2023) as initial condition when estimating $\gamma$. This choice avoids assuming zero correlation and allows us to account for some level of spatial correlation in retrieval errors for a same type of instrument. We note that assumption on $c$ does not directly affect the inversion results. Instead, they influence the initial conditions for estimating $\gamma$. Using a different assumption for $c$ would lead to a different optimal value of $\gamma$, which respectively rescales the observation error and ideally results in identical posterior emissions.

12. l 387: "the assimilation did not ingest TROPOMI observations poleward of 60°N". Since there is a clear focus on Boreal fires, and since this choice impacts the results, I was wondering if the sharp cut-off at 60 degree could be justified a bit more. It is mentioned that snow-covered land should be avoided. But most of the fires will occur after the snow has melted. Is snow-cover provided in the TROPOMI product? If so, could this be used to refine the filtering and provide an option to use observations north of 60 degree?

We first clarify that while TROPOMI observations poleward of 60° latitude were not directly assimilated, the assimilation system still adjusts emissions and concentrations beyond this limit through model transport and assimilation cycling (using a 3-day assimilation window). As emissions influence downwind concentrations, and updates propagate through subsequent assimilation windows, improvements can extend beyond the observation limits. For instance, as shown in Figure 8, the inversion significantly improves CO concentrations at the NDACC Arrival Heights station in Antarctica (77.8°S, 166.67°E), which lies well outside the 60°S–60°N band. Similarly, in the eastern part of the Siberian inversion region (poleward of 60°N), posterior emissions still show clear updates (see Fig. 4d-i), due to the influence of (nearby) assimilated observations. At St. Petersburg (59.9°N, 28.6°E), close to the 60°N cutoff, posterior CO estimates show a noticeable improvement relative to the prior, that further support this.

Regarding the justification for the 60° cutoff: the primary reason for this filtering is to avoid potential biases in the data associated with surface albedo, especially over snow-covered land. This approach is consistent with previous CO inversion studies using MOPITT data (Jiang et al., 2011, 2017) and also with practices in TROPOMI $CH_4$ inversions (e.g., Hasekamp et al., 2022; Lorente et al., 2021), which similarly exclude high-latitude data to avoid albedo-related retrieval biases. We also conducted a simple sensitivity test comparing inversion results at the TCCON ETL that showed that including observations up to 90°N/S resulted in a slight degradation in correlation and slope.

Also, at the time of our study, we only had access to the operational TROPOMI CO Level 2 dataset (Copernicus Sentinel-5P, 2024), which provides the quality flag (i.e., qa_value variable) but does not include snow cover or surface albedo as separate flag. While stricter qa_value filtering could remove some low-quality observations from snow-covered regions, this would also lead to a significant reduction in data coverage and density, which could negatively impact inversion performance (see also our response to Comment 4).

13. l 402: "higher accuracy provided by TCCON". The surface column observations are point-like while the model represents 2x2.5 degree box averages. There may be a considerable representativity uncertainty, bigger than the measurement uncertainty, especially for fire plumes passing over the station. Was such a representativity term considered/included, and is there a way to estimate it? Please add a discussion.

In this statement, we were referring specifically to the higher instrumental accuracy of TCCON observations compared to satellite retrievals. The addition of TCCON data to the assimilation system, alongside TROPOMI, leads to a noticeable reduction in posterior standard deviation, as demonstrated in Table 3 and Fig. 5. For instance, regions with TCCON coverage exhibit stronger uncertainty reduction compared to areas without TCCON data.

It should be noted that because the TCCON instruments are measuring the total column, the measurements should not be considered point-like. As shown by Belikov et al. (2017), due to the influence of transport in the free troposphere, TCCON sites can have footprints that are much larger than the model grid box.

Regarding the issue of representativeness error, a common approach to account for representativeness uncertainty is to include it as part of the observation error covariance (e.g., Janjić et al., 2018). In our study, we address these errors indirectly through observation error inflation, described in Section 3. While this method may not explicitly and fully quantify representativeness errors, it compensates for their overall impact by tuning the total observation error during the inversion process. We have modified the text to explain our approach for capturing representativeness errors.

**Text revised and added in Section 3** (lines 372-376)
"We generate super-observations for both TROPOMI and TCCON by aggregating measurements to the model time and grid. This aggregation helps mitigate spatiotemporal representativity errors and facilitates LETKF computations. The associated super-observation errors are calculated as described in Section 2.1. We further apply an inflation factor to the observation errors, while initially assuming an observation error correlation of 0.28, following previous studies (Chen et al., 2023; Pendergrass et al., 2023)."

14. Figure 5: I was wondering if the extra reduction due to TCCON occurs close to the TCCON stations? I would find it interesting to have extra plots of the difference between the right and left panels, with the TCCON locations also indicated.

The extra reduction in error variance due to TCCON is not strictly limited to the vicinity of the stations. Instead, while the effect is generally stronger near TCCON locations, particularly in North America and Europe where the TCCON network is denser, it also extends across broader regions. This broader impact reflects the effect of transported plumes combined with the cycling nature of our 3-day assimilation window and the high temporal density of TCCON measurements. This allows regions beyond the station vicinity and in longer distance to be indirectly constrained (see our response to Comments 10 and 12 for details). The improvement away from the TCCON sites can be seen in Fig. 8, where we compared the model at the NDAAC sites. The improvement is greatest at NDACC sites that are near TCCON sites, but there is improvement everywhere.

15. l 521: "This likely implies that the difference of the perturbed and unperturbed forecast of the state vector, which approximates their covariances, correlates better with the actual emissions, such that the greater variations in GFAS emissions result in higher DOFS." I do not understand this argument. What is the relation between the presence of variations and DOFS?

Given the same model configuration and observation set across the three experiments, we obtain slightly larger DOFS for GFAS compared to QFED and GBBEPx. Our findings also indicate that GFAS exhibits larger overall emission magnitudes and greater spatiotemporal variability. It also better captures the variability observed in independent datasets (see Fig. 7, Fig. 9, and Fig. 11a).
Since DOFS is computed based on the sensitivity of the model forecast to emission perturbations (using the same perturbation size, model, and observations), the remaining factor contributing to the differences between the three inventories is the inherent spatial and temporal variability in the prior emissions. We argued that this variability probably has some large influences on the structure of the $C$ matrix (Eq. 6). That might be why GFAS with higher variability has higher DOFS values (Eq. 5).

**Text revised and added in Section 4.1.2** (lines 589-591)
"This likely implies that the difference between the perturbed and unperturbed forecasts of the state vector, which defines the elements of matrix $C$, correlates with the spatiotemporal variability of the prior emissions. Thus, GFAS prior, with greater variability than the other priors (Fig 3a-c and Fig 4a,f,k), may result in higher DOFS."

16. Fig.8: In the text on p 24 the $R^2$ is reported as "correlation". This is normally called the "Coefficient of determination", or explained variance. Does figure 8 report the correlation, or the square of the correlation?

Thanks for mentioning this point. Fig. 8 shows $R^2$ as well, not correlation. We have updated the label of Fig. 8 to "$R^2$" in the revised manuscript.

**Text revised in the caption of Fig. 8** (lines 620-621)**:**
"The top row of each panel shows the coefficient of determination ($R^2$) of the model using prior or assimilation using posterior with respect to the measurements,"

17. Fig.8: Are there stations shown in this plot which are equipped with both NDACC and TCCON instruments? If so, please indicate those, or note this is not the case.

Yes, three of the stations are shared between TCCON and NDACC. This information has been updated.

**Text added to the caption of Fig. 8** (line 622)**:**
"Tsukuba, Wollongong, and Lauder are collocated stations for both TCCON and NDACC."

18. l 568: "sites close to or downwind of TCCON sites" It would be useful to mention the distance between the sites. Is it more or less than a grid box?

We added the information to the revised manuscript:

**Text revised and added in Section 4.1.2** (lines 633-635):
"For the NDACC sites collocated or downwind of TCCON sites, such as Tsukuba, Lauder, Wollongong (collocated stations), and St. Petersburg (~2000 km downwind of European stations),"

19. l 764: "there is stronger agreement at the NDACC and in situ sites that are located in close proximity to the TCCON measurements used in the inversion". This is not really a surprise.

We agree that this statement may be generally expected; however, it is not always guaranteed in practice. The system is complex, and many factors, such as local meteorology, topography, emission patterns, and the way observations and their associated error statistics are assimilated, can influence the resulting analysis.

20. l 853: "Our analysis yields the following optimal values: gamma_tropomi = 0.2 and gamma_tccon = 5, delta = 0.08, r = 500 km, with a minimum of three months for spin-up, one month for burn-in, and a minimum of 24 ensemble members - each chosen to minimize OmF statistics." This is quite a number of parameters to obtain from one set of OmFs, which are also influencing each other. Please provide a plot to document this analysis. In particular it is interesting to learn how the two gamma values are obtained.

Thanks for the feedback. We agree that estimating several parameters simultaneously using one set of OmFs, especially if the parameters influence each other, may not even be possible, as it can be an underdetermined problem that does not provide sufficient information to constrain all parameters together. Indeed, the estimation process has to be involved with trade-offs and assumptions. Here is a brief explanation:
The estimation was performed using a brute-force sensitivity method (±50% perturbations) rather than through a formal optimization framework. Specifically, for each parameter, we tested a small number of values and selected the one that minimized the OmF statistics. To simplify the estimation and reduce computational cost, we assumed the parameters to be approximately independent. While we acknowledge that these parameters may influence each other, estimating them jointly would require a complex optimization system that is not only computationally expensive but also difficult to constrain with available OmFs. To address this, we first relied on values commonly used in the literature as default configurations, then perturbed one parameter at a time around its default, keeping the others fixed. For example, while estimating the regularization factor γ for TROPOMI, we fixed the localization distance at 500 km

as supported by prior studies (e.g., Gaubert et al., 2023; Miyazaki et al., 2012, 2015; see our response to Comment 10) and assumed $\gamma = 1$ for TCCON. Similarly, when estimating the localization radius, we fixed $\gamma = 1$. Although we initially assumed no interaction between parameters, we later tested for the interdependence by repeating the sensitivity analysis using the optimal $\gamma$ of one observation type while estimating the $\gamma$ of the other. The resulting OmF values slightly changed, but the optimal $\gamma$ remained unchanged. For computational efficiency, all sensitivity tests were performed using one month of data representative of the inversion period, and we adjusted the threshold to ensure a full recovery within a month of assimilation (see Table B1). The estimation followed a binary search-like refinement process: we began with a few candidate values and refined the search by comparing OmF results and selecting the value that minimized the error. Specifically, for $\gamma$, the estimation was guided by the relative size of the observations and state vector (Lu et al., 2022): $\gamma < 1$ was generally preferred when the number of observations exceeded the state vector size (e.g., TROPOMI), and $\gamma > 1$ when the observations were sparse (e.g., TCCON). Figure S4 has been added to the supplements to document this process. It shows the $\gamma$ estimation for both TROPOMI and TCCON; as explained, each was estimated separately by assuming $\gamma = 1$ for the other, and we verified that this assumption did not alter the resulting optimal values.

**Text added in Appendix B (**line 924):
"(e.g., see Fig. S4)."

[Figure]

Figure S4: Comparison of OmFs between OSSEs with different regularization factors ($\gamma$) used in the inversion. Using (a-b) $\pm50\%$ prior emissions relative to the true emissions, based on TROPOMI inversion and (c-d) $\pm50\%$ prior emissions relative to the true emissions, based on TCCON inversion.

---------------------------------------------------------------------------------------------------------------------------- --------------

**Referee 3 (RC3):**

The manuscript titled "Quantifying CO emissions from boreal wildfires by assimilating TROPOMI and TCCON observations" by Voshtani et al. with reference egusphere-2025-858 is a highly valuable scientific contribution in atmospheric air quality modeling and emission inversion for wildfires. Overall, it is very well written and provides a complete investigation of the benefit of large-scale CO emission inversion using TCCON total column retrievals. The study utilizes the CHEEREIO data assimilation toolkit for CO emission inversion and considers different wildfire emission inventories during the period between May and September 2023. In the first part, the benefit of assimilating TCCON data in addition to TROPOMI retrievals is analyzed on a global scale with 3 different global wildfire emission inventories. The results are compared with the assimilated, as well as different types of independent observations. The second part focuses on North America where the regional CFFEPS emissions are also included in the comparison. This part also includes a comparison and evaluation of different TCCON retrievals of CO.
Overall, I suggest accepting the manuscript with the following minor corrections:

**General comments:**
A)  My first general comment is on the length of the manuscript and its broad range of investigations. This is not a criticism, but rather reflects the extensive information content in this study. But still, I would at least suggest dividing the results in Sec.4 into two sections including the global and North America evaluation, respectively. You may even consider dividing the paper into 2 parts because especially the evaluation of different TCCON retrievals in Sec.4.5 appears to have a quite different purpose than Sec4.1-4.3 and thus addresses a different audience (which can also be seen e.g. in the summary). This would also allow for a more detailed description and evaluation of the different TCCON retrieval data in Sec.4.5 (see related smaller comments below).

We agree with the reviewer's suggestion to better distinguish the Global and North America analysis. To provide further clarity, we have restructured Section 4 as described below, making it easier for readers to follow the results and evaluations:

**Results and discussions (Section 4) structure revised:**

  4. Results and discussions
    4.1 Global analysis
        4.1.1 Comparison between prior and posterior emissions
        4.1.2 Error variance reduction and the information content
        4.1.3 Evaluation using ground-based observations
    4.2 North America analysis
        4.2.1 Assimilation performance for constraining boreal wildfires emissions
        4.2.2 Implications for vertical sensitivities in the inversion

**Additional revisions:**

• We have provided short introductory paragraphs in Sections 2 and 4 (see our response to General comment B), clarifying the existing descriptions and guiding the reader.

• In Section 4.2.2, we have added clarification to explain the rationale for separately examining the impact of TCCON InSb retrievals at ETL and the use of aircraft data (see our response below to Smaller comment 13)

We do not believe that it would be useful to split the paper into two parts. The evaluation in Section 4.2.2, although focused on using the InSb TCCON retrievals, is tightly connected to our main goal: improving constraints on CO emissions by jointly assimilating satellite and ground-based datasets. The intention here is not to evaluate the TCCON

retrievals and their measurement characteristics, such as averaging kernels, but rather to provide an example (a case study of ETL) of whether and how additional information (even at the same observation location) has the potential to improve on the emission estimates from the inversion results. In fact, the TCCON InSb retrievals, described in Section 2.2, are not a new measurement type, but an identical retrieval to the standard TCCON measurements, just in a different spectral range with a different vertical sensitivity. Note that from an assimilation/inversion point of view, this dataset can be assumed to be independent. To examine their impact separately on the emissions inversion, we provide Section 4.2.2, that has the same purpose as adding the standard TCCON data to TROPOMI CO inversion.

B) I would also highly appreciate a short introductory paragraph at the beginning of Sec.2 and 4 to guide the reader and provide an overview of the content of each subsection.
   o For Sec.2: It would be helpful to already provide the information which observations are actually used for assimilation and validation in the section overview. Otherwise the amount of different observations might be confusing. This could maybe also be included in the title of the subsections. It is also not very clear why the aircraft data are presented in Sec.2.4 while the other in situ data are in Sec.2.3.
   o For Sec.4: Please also describe that the global evaluation in Sec.4.1-4.3 includes the 3 global emission inventories, while the regional evaluation in Sec.4.4 and 4.5 additionally includes CFFEPS (and see general comment A above)

Thanks for the suggestions. We have added an introductory paragraph at the beginning of Section 2 and Section 4 as follows:

**Text added in Section 2** (lines 140-150)**:**
"This section describes the observational datasets and modelling framework and inputs used in this study. We first describe in Section 2.1-2.2 the two types of observations that are assimilated in the inversion framework: (i) TROPOMI satellite CO products and (ii) TCCON ground-based CO measurements. Then, in Section 2.3, we present several independent datasets (not assimilated) used for evaluation, including the Network for the Detection of Atmospheric Composition Change (NDACC, De Mazière et al., 2018) ground-based total column observations, in situ surface CO measurements from the World Data Centre for Greenhouse Gases (WDCGG) network and from Environment and Climate Change Canada's (ECCC) tall tower at ETL, and vertical profiles from in situ aircraft measurements. The same TCCON data are also used for comparisons between different experiments, although they are no longer independent information for the joint inversion. Finally, Section 2.4 provides a description of the GEOS-Chem model and emissions inventories that are used as the a priori estimate in our inversion's setup. The priors include three global inventories, including QFED, GBBEPx, and GFAS, and one regional inventory, CFFEPS, for North America."

**Text added in Section 4** (lines 396-410)**:**
"This section presents our main results by first assessing the CO emissions on a global scale (Section 4.1) and then focusing on North American emissions (Section 4.2), where the most extreme fire events took place during the study period in summer 2023. A series of experiments are conducted as listed in Table 2 for both global and regional analysis. For the global analysis (Sections 4.1.1-4.1.3), we use three biomass burning emission inventories (QFED, GBBEPx, GFAS) as priors, and compare how assimilating TROPOMI satellite observations alone or jointly with TCCON ground-based measurements affects our emissions estimates. A method to measure and compare error variance and information content is presented. For evaluation, two types of independent data, including NDACC total column and surface WDCGG measurements, in addition to the same TCCON observations (non-independent) are used. Focusing on North America analysis (Sections 4.2.1-4.2.2), we include an additional regional prior emissions inventory from CFFEPS (Chen et al., 2019) provided by ECCC and additional experimental InSb XCO data from TCCON at ETL to our analysis. First, our analysis explores the spatiotemporal variability of a posteriori and a priori CO fields during extreme fire episodes. Then, as a case study at ETL, we assess the impact of assimilating additional information from

experimental TCCON data at ETL, with unique retrieval characteristics, on local emissions constraints. The evaluations are performed using independent aircraft and tall tower measurements. The discussion presented in these two sections allows us to highlight both the broad and local impacts of our approach and the specific improvements achieved in areas most affected by fires."

C) Thought the results, there are several locations where the word "uncertainty" or "error" is used without a clear specification on the actual measure. Since different measures are used in the evaluation, it is important so use specific wording.
Uncertainty is used for the ensemble system == error variance or standard deviation.

Thank you for the feedback. We have revised the manuscript, by carefully defining the terms, uncertainty and error, identically throughout the entire text. All figures and tables are also consistent with this definition.
More specifically in the revised manuscript:

- Where we refer to "uncertainty", we explicitly state that it denotes the ensemble standard deviation or spread of ensembles from mean. This ensemble spread is the same size as the state vector (i.e., computed for each grid cell at each assimilation time window) based on LETKF analysis. Note that this definition was originally used in the entire results and discussions referring to the posterior emissions standard deviation (i.e., the caption of Table 3), except in Section 4.1.2.
- In Section 4.1.2, the uncertainty is originally defined to represent the ensemble error variance. However, to avoid confusion with the definition of uncertainty in other sections of the results and discussions as stated above, we replace the term "uncertainty" by "error variance" throughout the Section 4.1.2.
- Where we refer to "error", we specify that it denotes the standard deviation of the model−measurement residuals or the observation – forecast/analysis mismatch.

Below we provide the explicit changes we made for the specific lines labelled by the reviewer:

o l.53: how is the uncertainty quantified? Eg ensemble spread or difference to independent data?

Please see our response above about the definition of uncertainty. We revised the Abstract.

**Text revised in the Abstract** (lines 52-54)**:**
"The joint assimilation of TROPOMI+TCCON reduced the posterior $1\sigma$ uncertainty on the North American emission estimates by up to about 30%, while showing only a modest impact ($< 5\%$) on the mean estimate of the inferred emissions."

o l.358: How is the "a posteriori" uncertainty estimated? from LETKF analysis scheme? According to the table description for Tab.3, it is the a postriori ensemble standard deviation. Please specify in the text.

A posteriori uncertainty or ensemble standard deviation can be either obtained during LETKF analysis after computing posterior error covariance (i.e., $\widetilde{P^a}$ in Eq.1), or it can be obtained once the analysis increment applied to each ensemble member, by explicitly computing the ensemble spread around the mean.

**Text revised and added in Section 4.1.1** (lines 413-420)**:**
"Table 3 shows the total regional BB CO emissions from the a priori bottom-up inventories and the a posteriori emission estimates (i.e., ensemble mean) obtained from the TROPOMI-only assimilation and the joint TROPOMI and TCCON (i.e., TROPOMI + TCCON) assimilation. The standard deviations shown in this table are posterior ensemble spread based on the LETKF assimilation, which is referred to as posterior uncertainty throughout the discussion in this study. The vector of posterior uncertainties has the same size as the state vector, computed for each grid cell after each assimilation window. Accordingly, the posterior uncertainty in an inversion region (i.e., regional or global scale)

is the ensemble standard deviation of the total emissions. The total emissions per ensemble is obtained by summing the emissions across all grid cells in that region."

- o l.400: (as above?): Please define in the text how the uncertainties are estimated, ie as posterior ensemble spread after the assimilation as stated in the description of Fig.3? This is particularly important to interpret the large reduction of uncertainty when additionally assimilating TCCON compared to a small change in the posterior emission estimates themself (on a global scale). Please also define how the global and regional uncertainty estimated are calculated from the spatially resolved ensemble spread (or equivalent).

We added an explanation that defines the uncertainties and how it is computed at global and regional scales, at the beginning of Section 4.1. Please see our responses above to General comments B and C

- o l.586: The word "uncertainties" is unspecific in this case. Are you referring to correlations, biases? Or to the investigation of uncertainties as given by the ensemble spread in Sec.4.2? Please formulate more specific.

The uncertainties here refer to the error standard deviations between posterior/prior and independent data used for validation, shown as error-bar in Fig. 8b,d,f. This is an estimate of the model/analysis−measurements errors. Thus, to avoid a confusion between this and the uncertainty definition used earlier for ensemble spread, we revised the text.

**Text revised and added in Section 4.1.3** (lines 652-653)**:**
"In most of the cases, adding TCCON to the assimilation reduces the error standard deviations (i.e., posterior−measurement errors) while the mean bias remains almost identical to the bias of TROPOMI-only inversion."

- o l.664, l.666, l.667: please specify "uncertainty" in terms of the quantities that are given in Fig.11 or state where this statement is coming from elsewhere.

For the same reason as explained above, we replace the term "uncertainty" with "standard deviation" for the explanations of Fig. 11 in the revised manuscript. (please see our response to General comment C). We also added a brief description for Fig. 11a and Fig 11b. The markers and the error bars in Fig. 11b represent the mean bias and standard deviations of the prior/posterior−measurement residuals.

**Text revised and added in Section 4.2.1:**
(lines 722-725):
"Fig. 11a presents a Taylor diagram comparing the standard deviation and correlation of the a priori or a posteriori against TCCON measurements at ETL, and Fig. 11b shows the mean bias and standard deviations of the prior/posterior−measurement residuals. The a priori XCO are shown in circles, the a posteriori from the TROPOMI-only assimilation in squares, and the a posteriori from the joint TROPOMI+TCCON assimilation in triangles."
(line 729)**:**
"lowering the mean bias and standard deviation"
(line 731)**:**
"smaller mean bias and standard deviation"
(line 733)**:**
"the highest correlation, lowest standard deviation"

- D) From my knowledge figure descriptions like color-coding should only be given in the figure caption because it's hindering the reading flow in the text (eg. l.560ff, l.606ff, l.630ff, l.658ff, l.692ff, l.717ff, …, please check also other locations, especially before)

Thank you for this observation. Based on ACP guidelines, we could not find any explicit restriction stating that color-coding, marker types, or line styles must only be described in the figure captions and not in the main text. We appreciate the point about improving the reading flow, and we will consult the editor or the ACP support team for guidance in this regard.

**Smaller comments:**

1. Abstract: Wildfires play an important role on a wide range of scales, from very local impacts on individual hotspots to global scale. Please state more clearly in the abstract that this study focuses on the large-scale.

The information about the scale and spatiotemporal resolution is included in the revised manuscript.

**Text revised and added in the Abstract:**
(line 44):
**"We perform a global inverse modelling analysis to"**
(lines 45-47):
**"Using the GEOS-Chem model, we assimilated observations at 3-day temporal and 2° × 2.5° horizontal resolution from the Tropospheric Monitoring Instrument (TROPOMI) separately and then jointly with Total Carbon Column Observing Network (TCCON) measurements."**

2. Sec.2.4: can you please explain a bit more in detail how the aircraft data was sampled? Are the sensors mounted on commercial aircraft or was there a measurement campaign?

We added more details about aircraft measurements to the revised manuscript.

**Text revised and added in Section 2.3** (lines 230-241):
**"We use in situ aircraft CO measurements from the National Oceanic and Atmospheric Administration (NOAA) air sampling network (McKain, K. et al., 2024) taken as another independent source to evaluate our inversion results. The data product is freely available to public via https://gml.noaa.gov/ccgg/aircraft/. The aircraft program aims to capture temporal variability (i.e., seasonal and interannual changes) of the greenhouse gases in the lower atmosphere. Dry air mole fractions of CO are measured using flask air samples at different fixed altitude levels. It provides measurements at different sites across the United States and Canada and at different altitudes, descending from a maximum of 8000 m to the lowest sampling level at ~750 m (a.s.l). These data have been commonly used in previous studies to explore the large-scale changes in horizontal and vertical distribution of CO and greenhouse gases (Sweeney et al., 2015), to serve as benchmark for validating forward and inverse modelling analysis (Stephens et al., 2007; Yang et al., 2007), and to calibrate remote sensing retrievals (Wunch et al., 2010). Focusing on the impact of the experimental TCCON InSb data used in the inversion to constrain surface CO emissions, we use aircraft profiles at ETL during multiple time events (details are discussed in Section 4.2.2). Table 1 shows the list and geographical information of all observations used for evaluation."**

3. Sec.3: Please give a bit more details on the ensemble generation:
   o   Are the emissions from all sources perturbed or BB-only?

Yes, CO emissions in the prior are perturbed randomly from all sources using multiplicative factors. This results in the updating of the total CO emissions from all source sectors after LETKF inversion. We know that there is a chance that CO from other sources become misattributed to BB, and vice versa. However, since BB emissions are often spatially distinct from other source types (e.g., fossil fuel, biogenic) and given the fact that during the study period between May and September, BB contributed to the largest fraction of CO emissions (e.g., over 80% in North America), the level of misattribution is in fact negligible. This point is added to the revised manuscript (see below).

     o   How was the width of lognormal-distribution (ie perturbation variance) for the emission perturbations determined?

The reason for perturbing based on lognormal distribution is to ensure the positivity of the estimated emissions and to reflect the asymmetry of emission uncertainties more realistically (Hancock et al., 2025; Plant et al., 2022) (i.e., lognormal error better captures the skewed tails of emissions than the normal errors). To achieve this, we initialize the perturbation by sampling emissions scaling factors based on a lognormal distribution centered on 1 with $\sigma = 0.2$. This indicate that most ensemble members are initialized by scaling emissions using multiplicative factors in the range [0.67, 1.49] at 95% confidence (i.e., $2\sigma$ range in log-space or $[e^{-2\sigma}, e^{2\sigma}]$ in linear space). Also, the perturbation width $1\sigma$ (68% confidence) and $3\sigma$ (99% confidence) becomes [0.82, 1.22] and [0.55, 1.82], respectively. Figure below shows the width of distribution for lognormal perturbation of emissions based on three confidence intervals:

[Figure]

As suggested, we added a bit more details to the revised manuscript regarding the two points above:

**Text revised and added in Section 3:**
(lines 328-338):
"To obtain a posteriori estimate of CO emissions, we begin by initializing the ensemble scaling factors using a multiplicative random perturbation to the prior estimates from all emission sources based on a lognormal distribution. We assume lognormal errors on the prior emissions to ensure the positivity of the solution (i.e., prevent unrealistic negative scaling factors) and to better capture the skewed tails of the emissions distribution (Maasakkers et al., 2019; Plant et al., 2022). In the next step, CHEEREIO first runs GEOS-Chem for each ensemble member over the assimilation window and then applies Eq. 2 to those ensembles in the LETKF process. This process further scales the emissions based on the observation increments (i.e., $y^o - H(x^b)$), and the observation and the prior error covariances (Eq.1). Note that during the construction of error covariances, a logarithmic transform of the scaling factor distributions to a normal distribution is required to satisfy the assumptions by LETKF (Hunt et al., 2007), which can be transformed back to the lognormal distribution after LETKF process. Finally, gridded total CO emissions from all sources, including biomass burning (BB), fossil fuel, and biogenic emissions, are updated through the inversion process."

(lines 349-355):
"Then, an ensemble spin-up without assimilating observations is performed, where we perturbed emissions based on a lognormal distribution to create an ensemble spread. We assume a lognormal standard deviation of $\sigma = 0.2$, centered around 1, that provides ensemble members scaling factors between 0.55 to 1.82 with 99% confidence. This perturbation level is sufficient to generate meaningful ensemble spread while avoiding unrealistically high or low values for constructing the prior error covariance. We use a spin-up of about three months, comparable to the CO

lifetime during summer, not only to provide a reasonable spread in the ensemble members but also to ensure the concentrations will reflect the perturbation of emissions."

4. Sec.4.1, l.386-388: This suggests that you are including all global emissions in the comparison. Since your assimilated observations were limited to below 60°N (&S), excluding this region from the comparison would provide more isolated insights on the actual impact of assimilation. Did you look at this as well?

We did not look into this because we expect improvements even in regions where there are no observations since the Kalman filter transports the information from the observations throughout the domain. We can see this effect in the improvements in the evaluation against the Arrival NDACC site (Fig. 8) located in Antarctica (77.8° S). Of course, the improvements will be greater in regions where observations are available. We have added an explanation of this to the text (please see our response to Comment 12 from Referee 2). We revised the manuscript accordingly:

**Text revised and added in Section 4.1.1** (lines 442-446):
"The discrepancy between the a posteriori Siberian emissions could be because the assimilation did not ingest TROPOMI observations poleward of 60°N, and there are large emission sources poleward of 60°N, as can be seen in Fig. 3. Note that although the emissions beyond 60° may not be directly corrected from local observations at the current assimilation step, they can still be updated using observations between 60°S-60°N and through model transport and cycling of the assimilation that propagate information globally."

5. Sec.4.1: l.412f: I don't understand the importance of this sentence in the given context. Why should the a priori emissions and error estimates be different between the experiments?

(lines 467-471):
"lower overall uncertainty from the joint TCCON and TROPOMI assimilation suggests an improved handling of the spatial under-sampling error in TROPOMI-only assimilation, which is reflected in the uncertainty estimates. This improvement could be particularly important, since the a priori emissions and their errors were kept fixed between the joint and TROPOMI-only assimilations."

The last sentence is to emphasize the fact that the uncertainty reduction that we observed is only because of adding TCCON data to the assimilation, not other factors such as a priori emissions or assumptions on their errors. We clarify this in the revised manuscript.

**Text revised and added in Section 4.1.1** (lines 469-471):
"This improvement could be particularly important, as it shows the isolated effect of adding TCCON data to the inversions, when other factors such as a priori emissions and their errors were kept fixed between the joint and TROPOMI-only assimilations."

6. Sec.4.1, l.427-429: Is this "improvement" w.r.t. to the posterior? Please state clearly right when mentioning the 1st time in L.427.

Thanks for the feedback. Yes, the improvements here are relative to the posterior. We revised the manuscript accordingly.

**Text revised and added in Section 4.1.1** (lines 484-487):
"In North America, QFED shows only slight variations in the a priori emissions (Fig. 3a). GBBEPx (Fig. 3f) shows some degree of improvement over QFED relative to TROPOMI posterior during a few fire episodes. For GFAS, however, this improvement over QFED (Fig. 3k) is quite significant, so that the GFAS prior exhibits reasonable

agreement in both magnitude and temporal variability with the a posteriori emissions from the TROPOMI-only assimilation."

7. Sec.4.1, l.455-447: You mentioned the impact of TCCON already before (eg L.443). If the previous comparison refers to a priori vs joint a posteriori, please explicitly state so. From L.431ff, I was assuming you were comparing a priori vs TROPOMI a posteriori above. If this is the case, then I don't understand the explanation with few TCCON stations in L.443ff. Please clarify.

The last paragraph in Section 4.1.1 (lines 518-524) highlights the main implication or a conclusion of the earlier discussions. That is focused on the overall impact of TCCON in the joint inversion, although we have already described the results for joint inversion earlier in a few places (lines 508-517). We clarified it in the revised manuscript. Please also see our responses to Comments 5 and 6 above.

**Text revised and added in Section 4.1.1** (lines 518-524)**:**
"Overall, according to the discussion above and the results from the Observing System Simulation Experiments (OSSEs) demonstrated in Appendices A and B, we find that although TCCON alone may not significantly constrain spatiotemporal variability in the major inversion regions—likely due to the limited number of measurement sites— it is still clear that adding TCCON to TROPOMI in the joint inversion reduces the posterior uncertainty estimates everywhere  compared to the TROPOMI-only inversion."

8. Sec.4.1, l.455ff: That's a reasonable conclusion, is it possible to verify that, e.g. with the OSSE where you only assimilate TCCON in Apx.B?

Yes, it is possible to verify that assimilating TCCON data alone may not provide the same level of constraint on the spatiotemporal distribution of emissions as when they are assimilated jointly with TROPOMI. Although we did not include a separate evaluation of the TCCON-only assimilation in the manuscript, we initially tested this scenario and found that emissions in the major inversion region can still be constrained. This is because TCCON provides a high temporal density of information that can be propagated over long distances within the 3-day assimilation window and globally through assimilation cycling.
However, since TCCON sites are spatially sparse, the strength of the local constraint depends on the observational footprint, which is influenced by the geographic distribution of the sites and the duration over which the measurements are assimilated. For example, constraints are typically stronger and more rapid over North America and Europe compared to Africa (see Table B1 and Comments 12–14 from Referee 2). As a result, assimilating data from a station over a longer period allows its influence to spread more broadly through the system, leading to a stronger overall constraint on emissions. In contrast, in the joint inversion with TROPOMI, the broader spatial coverage of TROPOMI enables more immediate constraint of the spatial distribution of emissions. In this case, TCCON provides complementary information that primarily helps reduce the posterior uncertainty, rather than significantly correcting the prior emissions.

9. Sec.4.2: For understanding, I would suggest adding a note that the normalized error reduction (and maybe also the DOFS?) increases with increasing weight of observations compared to the model a priori, even for overfitting to the observations, and that a small error reduction could also be due to a good model a priori.

**Text revised and added in Section 4.1.2** (lines 546-555)**:**
"We find greater reduction of error variance in the joint inversion compared to the inversion with TROPOMI-only data, primarily in NA and in the vicinity of TCCON stations. In fact, the reductions correlate with the weight of observations compared to the model a priori, so that increasing the weight of observations with respect to the model a priori could result in higher reductions of the error variances. For example, with QFED as the a priori emissions, the rate of reduction is greater in boreal Canada, central and southern part of the United States, western Europe, eastern

Asia, Siberia, and Australia. This indicates where TCCON provide additional information to further constrain the emissions based on the QFED BB inventory. In addition, comparing the cases with different inventories suggests that there could be differences in error variance reduction due to the model a priori. Accordingly, the slightly greater reduction with GFAS, compared to QFED or GBBEPx, is likely due to its better spatiotemporal variability, which enables the inversion to better exploit the information from observations.

10. Sec.4.3: Does the validation use the same TCCON data as used in joint inversion? If so (as stated in L.539f) state explicitly right from the beginning (e.g. l.528) because it is an important information that the validation data is not independent for the joint inversion.

Yes, TCCON data used for evaluation here is not independent information. We explicitly stated this in the introductory paragraphs of Sections 2 and 4. Please see our responses to General comments A and B. As suggested, we also stated this in Section 4.1.3

**Text revised and added in Section 4.1.3** (lines 593-596)
"We evaluate the inversion against TCCON, NDACC total column retrievals, and in situ WDCGG measurements to better understand the constraint from each measurement type used in the inversion. NDACC and WDCGG are independent data while TCCON are the same data as those used in the assimilation, so they are not independent for evaluating the joint inversions."

11. Sec.4.4: The title is rather long which makes it difficult to extract the specific content if this subsection: North America. Please shorten the title to stress it's specific purpose more clearly. (and see also general comments above)

We revised the title of subsections in Section 4 in the revised manuscript. Please see our responses to General Comments A and B.

12. Sec.4.4: Which timer period is used in the statistical evaluation inf Fig.11? The whole period or some of the episodes? Please explain in the figure caption and the main text.

Thanks for the feedback. We added a description of the time period to the caption and the main text.

**Text revised and added in Fig. 11 caption** (lines 716-717):
"Figure 11: Taylor diagram for evaluation of the assimilation and a priori model using four different biomass burning inventories against TCCON XCO measurements at East Trout Lake (ETL) between May and September 2023."

**Text revised and added in Section 4.2.1** (lines 721-722)
"We also evaluated the a priori and a posteriori XCO with TCCON data from East Trout Lake (ETL) for the entire simulation period between May and September 2023."

13. Sec.4.5:
  o  Is the InSb retrieval really independent from the original retrieval, even when from same instrument? Without deeper knowledge about the retrievals: If its about different averaging kernels, it's probably more an "valuable" additional information rather than an completely independent measurement in a mathematical sense? Please explain.

It is true that the InSb CO retrievals are from the same instrument as the standard CO retrievals, so the retrieval algorithm, the instrument optical alignment, and the a priori model atmosphere (including vertical profiles of pressure, temperature, humidity, and CO) are identical between the two retrievals. That said, the two retrievals use different, non-overlapping parts of the spectrum (near 2100 cm$^{-1}$ for the InSb measurements, and near 4300 cm$^{-1}$ for the standard

TCCON measurements), and the measurements are recorded on different detectors. The two CO retrievals are not less independent than CO and, say, $CO_2$ retrievals from the same instrument, except that they use the same a priori profile. From an assimilation/inversion perspective, we assume the InSb data are independent since they have different measurement characteristics, including different column retrievals, averaging kernels, and estimated errors. These factors are sufficient to obtain a distinct innovation and analysis increments (i.e., $\gamma X^b \widetilde{P^a}(Y^b)^T R^{-1}(y^o - \overline{H(x^b)})$ from Eq.1), which result in a different emissions estimate with unique spatiotemporal characteristics. For example, Fig. S3 in the supplement demonstrates how the estimated emissions are different spatially when we include or exclude InSb data from the assimilated observation.

In addition, we know that the TCCON InSb data has errors likely correlated with standard TCCON. The error correlations ideally could be constructed and taken into account in the $R^{-1}$, however due to a limited or missing accurate information, it is common practice in data assimilation/inversion problems to make an assumption of unaccounted correlated errors and then approximate this impact in the analysis increments; for example by tuning the relative weight of observation errors and prior errors using regularization parameter (see our response to Comment 8 from Referee 2).

Could you give some details about the combined TCCON observations? Were the two products combined, and how, before using in the assimilation? Or were both used jointly in the assimilation? If so, how are where treated, e.g. are they assumed to be 2 completely uncorrelated datasets? And is this a reasonable assumption, and why?

InSb measurements include distinct XCO retrievals, averaging kernels, and reported errors, but same a priori profiles as the standard TCCON data. When these data are jointly assimilated with TROPOMI, alone or together with standard TCCON, they are provided as separate data to the assimilation. This results in increasing the size of the observation vector and observation error covariance accordingly: $(R^{-1}(y^o - \overline{H(x^b)}))$. Please see the response above regarding the error correlations. We added explanations to the revised manuscript.

**Text revised and added in Section 4.2.2** (lines 738-743):
"Since the measurements provide us with an independent set of XCO, with distinct averaging kernels, we take those as separate pieces of information into our joint TROPOMI and TCCON inversion. Similar to the standard TCCON, we assume uncorrelated errors for the InSb data, however, the effect of possible error correlations can be approximated and taken into account by re-tuning the regularization factor. The InSb CO measurements are processed in a similar way to produce a separate set of hourly grided TCCON InSb data. Thus, when these data are added to the joint inversion, the size of the observation vector and observation error covariance increases accordingly."

14. Sec.4.5, l.701ff: To me, the differences appear to be quite small. I would suggest stressing this a bit more, e.g. by adding this information in the last sentence in l.712ff.

As suggested, we revised this point in the manuscript.

**Text revised and added in Section 4.2.2** (lines 780-783):
"However, the combined standard TCCON and InSb CO assimilation captures the variability slightly better than the standard TCCON CO assimilation, with improved correlations relative to independent data, and also provides us with more representative estimates of the background CO."

15. Sec.4.5, l.718ff: That's an interesting point. Did you look into how the different averaging kernels impact the a posterior emissions? Do they induce in different spatial or temporal distributions of emissions?

In the supplements, Fig. S3 illustrates the estimated gridded emissions' spatial distribution in North America using standard TCCON data assimilated with and without InSb XCO. These figures indicate that using TCCON data with

different averaging kernels, we obtain distinct spatial and temporal patterns in the posterior emissions. These differences occur not only near the TCCON stations but also across the broader regional domain, both upwind and downwind of the measurement sites (see our response to Comment 14 from Referee 2).

As mentioned in our response to General comment A, we do not intend to add further discussion on this topic in the manuscript, as it would divert the focus from the main objective and would require additional analysis and evaluation beyond the scope of this study.

**Technical and formulation-related comments:**

- l. 216: "aircraft directly samples the atmosphere" Is there a word missing? Do you mean aircraft measurements (then "sample" in plural)? The aircraft itself does not sample.

We fixed this in the revised manuscript. Please see our response to Smaller comment 2 for a revised paragraph about aircraft measurements.

- l.226 & 228: The word "driven" is used in two different contexts here.

We have modified these in the revised manuscript

1. I'm not aware of using word for assimilated observations in a model. Suggest replacing here.

**Text revised in Section 2.4** (lines 245-246):
"The GEOS-Chem model (http://www.geos-chem.org, last access: 1 July 2024) is a global 3D CTM that uses assimilated meteorological observations as input from the NASA Global Modelling and Assimilation Office (GMAO)."

2. Probably suitable word when referring to reanalysis, however formulation remains unclear. Is MERRA-2 used for initialization? Suggest more specific formulation.

MERRA-2 is not used for initialization. GEOS-Chem is a chemical transport model, so it does not simulate the meteorology. Instead, MERRA-2 meteorological fields are continuous read into the model to drive the transport.

**Text revised and added in Section 2.4** (lines 246-248):
"We use version 14.1.1 of the GEOS-Chem CTM driven by meteorological input from the Modern-Era Retrospective Analysis for Research and Applications, Version 2 (MERRA-2; Gelaro et al., 2017)."

- l.230-232: Please add a specific literature reference of the model description here.

We added (Bey et al., 2001) to the revised manuscript.
Bey, I., Jacob, D. J., Yantosca, R. M., Logan, J. A., Field, B. D., Fiore, A. M., Li, Q., Liu, H. Y., Mickley, L. J., and Schultz, M. G.: Global modeling of tropospheric chemistry with assimilated meteorology: Model description and evaluation, J. Geophys. Res. Atmospheres, 106, 23073–23095, https://doi.org/10.1029/2001JD000807, 2001.

**Text revised in Section 2.4** (lines 249-250):
",which is degraded to $2° \times 2.5°$ horizontal grid and 47 vertical levels (Bey et al., 2001)."

- l.293, Eq.(1): tilde over analysis error covariance matrix was not defined

The tilde represents the ensemble space (S), and $\widetilde{\boldsymbol{P^a}}$ is analysis error covariance matrix in ensemble space ($k \times k$) according to (Hunt et al., 2007).

**Text revised in Section 3** (lines 315-316):
"$\widetilde{\boldsymbol{P^a}}$ is the analysis error covariance matrix in ensemble space"

- l.301ff:
  - Please specify more clearly which variables are referring to emissions (ie in model space), and which to concentrations (ie in observation space), eg for apriori profile, simulated CO profile, …

We have already defined the analysis, $\boldsymbol{x^a}$ (solution) and background $\boldsymbol{x^b}$ state vector of emissions earlier in lines 314-315.

**Text revised in Section 3** (line 325)
" $\boldsymbol{c^{a\ priori}}$ is the a priori profile provided by measurement data"

  - the definition of M remains unclear: does it refer to the forecast model or emission-to-concentration mapping, or both? If it is mapped in observation space as denoted, shouldn't it include the operator H? Please clarify.

**Text revised in Section 3** (lines 325-327):
"$\boldsymbol{M}(\boldsymbol{x^b})$ represents a forward operator that operates on the emissions state vector and produces CO profiles that are spatially and temporally interpolated at the location and time of the measurements,"

- l.306: ", and the observation and the prior error covariances": It unclear where is this part of the sentence referring to? Are the observation and error covariances also scaled based on the obs increments? Or are they also passed on the LETKF processor? Please specify.

Here, we talk about Eq.1 where all the terms on the far right-side as mentioned in line 335 impact the increment in the scaling factors, which are all part of the LETKF processor. We clarified this point in the revised manuscript.

**Text revised in Section 3** (lines 333-334):
"This process further scales the emissions based on the observation increments (i.e., $\boldsymbol{yo} - H(\boldsymbol{xb})$), and the observation and the prior error covariances (Eq.1)."

- l.337: Spelling error? Do you mean "updated by"?

**Text revised in Section 3** (lines 369-370)
"We also use an inflation factor $\Delta = 0.08$ to compensate for a quick reduction of the ensemble spread, which may result in an inversion not being updated by subsequent observations."

- Fig.4: The subfigures are quite small, it's especially hard to see influences of TCCON assimilation with a low spatial coverage

Thank you for this comment. We appreciate the concern about the visibility of individual subplots in Fig. 4. Our intention is to present all subpanels together so they can be directly compared. Given the current layout and number of subplots, there is unfortunately no additional space to enlarge each panel further without restructuring the entire figure. However, we have ensured that the figure is provided at high resolution (3,400 × 2,475 pixels, 330 dpi), so all details are clearly visible when viewed electronically or when zoomed in. The figure is a raster image file (.png format)

and the image pixels size is big enough to detect each grid cell in each subplot. We will consult with the editorial support team and provide the figure file at full resolution or separately for subplots for the final submission of the revised manuscript

- l.481f: "more reliable" compared to what? The a priori?

That is a general statement that the higher reduction of uncertainty (error variance in the revised manuscript) indicate that we are more reliable on our posterior estimation (compare to the case if the uncertainty reduction is lower). We revised the sentence to make it clear:

**Text revised in Section 4.1.2** (lines 545-546):
"The greater reduction in error variances implies higher confidence in the posterior estimate at those locations."

- l.588: "difference "unclear: Do you refer to differences between TROPOMI-only and joint inversion or between the different prior emissions?

Here, we refer to the differences between TROPOMI-only posteriors using QFED, GBBEPx, and GFAS; and similarly, the differences between joint inversions posteriors using QFED, GBBEPx, and GFAS.

- l. 593: "underestimation" w.r.t. what? Please specify and refer to section above if applicable

We clarified it in the revised manuscript.

**Text revised in Section 4.2.1** (lines 659-662):
"The emissions primarily came from the boreal forest across Canada, which were poorly estimated by the bottom-up emissions inventories with a 31% - 67% underestimation in the a priori relative to a posteriori from both TROPOMI-only and joint inversions in this region during the study time period (see Section 4.1 and Table 3)."

- Fig.9: The in-figure legend is confusing: Is "ensemble mean" the a posteriori ensemble mean, and "model a priori" the a priori ensemble mean? From my understanding, both are ensemble means, right? Then it would be clearer to label "a priori" and "a posteriori" (for each emission model) in the legend and the information "ensemble mean" can be added in the figure caption.

Thanks for the feedback. Yes, "Ensemble mean" in the figure caption refers to a posteriori ensemble mean of concentrations; however, there is no actual ensemble mean for a priori. This is in fact model forecast based on prior emissions, which is equivalent to a priori estimate of concentrations, and can also be referred to as model a priori. As suggested, we updated the in-figure legend of Fig. 9 and the caption to clarify the point.

[Figure]

**Figure 9**: Evaluation of the domain-averaged CO concentrations (XCO) of a priori model (dashed lines) and a posteriori using TROPOMI assimilations (solid lines) between four different inventories in North America, including QFED (blue), GBBEPx (green), GFAS (red), and CFFEPS (purple) against TROPOMI XCO measurements (black). A priori model refers to a model forecast using prior emissions and a posteriori is equivalent to the ensemble mean from LETKF. Three extreme wildfire episodes across boreal regions are chosen for comparison of the assimilation results using different inventories and comparison of the assimilation with a priori model.

- l.622: "even though the … ." What is the massage of this part of the sentence? You already compare with the a priori of GFAS and CFFEPS before. So they are closer to TROPOMI compared to what? Maybe the a priori of QFED and GBBEPx? Please reformulate and make the specific massage of the sentence more clear.

We removed the part of the sentence starting with "even though" to make the message clearer.

**Text revised in Section 4.2.1** (lines 659-691):
"Despite their poorer a priori estimates, QFED and GBBEPx provides a posteriori XCO that agrees with TROPOMI measurements better than the a priori of GFAS and CFFEPS."

- Fig.12: in-figure labels are very small, hardly readable

We have enlarged in-figure labels in Fig. 12 in the revised manuscript.

[Figure]

**Figure 12:** Evaluation of the assimilation and a priori model using CFFEPS biomass burning inventories against tall tower measurements at ~ 60 m above sea level altitude at East Trout Lake (ETL). (a-b) a priori model (blue), (c-d) assimilation using joint TROPOMI and TCCON standard XCO TCCON data (red), (e-f) assimilation using joint TROPOMI and InSb XCO TCCON data (yellow), and (g-h) assimilation using joint TROPOMI and the standard and InSb TCCON data (orange) are compared with tall tower measurements (green).

---------------------------------------------------------------------------------------------------------------------------- --------------

**Additional updates:**

1. We updated an affiliation in the revised manuscript (line 15). We changed "[5]National Institute of Water & Atmospheric Research Ltd (NIWA), Lauder, New Zealand" to "[5]New Zealand Institute for Earth Science Limited, Lauder, NZ".
2. We updated Fig. 1a legend to clarify the number of aggregated TCCON observations. We also added the terms "quality controlled and aggregated" and "non-aggregated" to the caption of Fig. 1 (lines 207-208): "The number of quality controlled and aggregated TROPOMI and TCCON observations (XCO) used in the global inversion as described in Sections 2.1 and 2.2. (b) The variability of the number of non-aggregated TROPOMI".
3. We changed "Takatsuji Shinya" to "Takatsuji, S." in Table 1 and the References.
4. We added "XCO" in line 189: "TCCON XCO data are no longer scaled to the WMO trace gas scale".
5. We added "19,733" in line 204: "providing 19,733 median hourly averaged observations".

**References**

Belikov, D. A., Maksyutov, S., Ganshin, A., Zhuravlev, R., Deutscher, N. M., Wunch, D., Feist, D. G., Morino, I., Parker, R. J., Strong, K., Yoshida, Y., Bril, A., Oshchepkov, S., Boesch, H., Dubey, M. K., Griffith, D., Hewson, W., Kivi, R., Mendonca, J., Notholt, J., Schneider, M., Sussmann, R., Velazco, V. A., and Aoki, S.: Study of the footprints of short-term variation in $XCO_2$ observed by TCCON sites using NIES and FLEXPART atmospheric transport models, Atmospheric Chem. Phys., 17, 143–157, https://doi.org/10.5194/acp-17-143-2017, 2017.

Bey, I., Jacob, D. J., Yantosca, R. M., Logan, J. A., Field, B. D., Fiore, A. M., Li, Q., Liu, H. Y., Mickley, L. J., and Schultz, M. G.: Global modeling of tropospheric chemistry with assimilated meteorology: Model description and evaluation, J. Geophys. Res. Atmospheres, 106, 23073–23095, https://doi.org/10.1029/2001JD000807, 2001.

Borsdorff, T., Aan de Brugh, J., Hu, H., Aben, I., Hasekamp, O., and Landgraf, J.: Measuring Carbon Monoxide With TROPOMI: First Results and a Comparison With ECMWF-IFS Analysis Data, Geophys. Res. Lett., 45, 2826–2832, https://doi.org/10.1002/2018GL077045, 2018.

Borsdorff, T., aan de Brugh, J., Schneider, A., Lorente, A., Birk, M., Wagner, G., Kivi, R., Hase, F., Feist, D. G., Sussmann, R., Rettinger, M., Wunch, D., Warneke, T., and Landgraf, J.: Improving the TROPOMI CO data product: update of the spectroscopic database and destriping of single orbits, Atmospheric Meas. Tech., 12, 5443–5455, https://doi.org/10.5194/amt-12-5443-2019, 2019.

Byrne, B., Liu, J., Bowman, K. W., Pascolini-Campbell, M., Chatterjee, A., Pandey, S., Miyazaki, K., van der Werf, G. R., Wunch, D., Wennberg, P. O., Roehl, C. M., and Sinha, S.: Carbon emissions from the 2023 Canadian wildfires, Nature, 633, 835–839, https://doi.org/10.1038/s41586-024-07878-z, 2024a.

Byrne, B., Liu, J., Bowman, K. W., Yin, Y., Yun, J., Ferreira, G. D., Ogle, S. M., Baskaran, L., He, L., Li, X., Xiao, J., and Davis, K. J.: Regional Inversion Shows Promise in Capturing Extreme-Event-Driven CO2 Flux Anomalies but Is Limited by Atmospheric CO2 Observational Coverage, J. Geophys. Res. Atmospheres, 129, e2023JD040006, https://doi.org/10.1029/2023JD040006, 2024b.

Chen, J., Anderson, K., Pavlovic, R., Moran, M. D., Englefield, P., Thompson, D. K., Munoz-Alpizar, R., and Landry, H.: The FireWork v2.0 air quality forecast system with biomass burning emissions from the Canadian Forest Fire Emissions Prediction System v2.03, Geosci. Model Dev., 12, 3283–3310, https://doi.org/10.5194/gmd-12-3283-2019, 2019.

Chen, Y., Shen, H., Kaiser, J., Hu, Y., Capps, S. L., Zhao, S., Hakami, A., Shih, J.-S., Pavur, G. K., Turner, M. D., Henze, D. K., Resler, J., Nenes, A., Napelenok, S. L., Bash, J. O., Fahey, K. M., Carmichael, G. R., Chai, T., Clarisse, L., Coheur, P.-F., Van Damme, M., and Russell, A. G.: High-resolution hybrid inversion of IASI ammonia columns to constrain US ammonia emissions using the CMAQ adjoint model, Atmospheric Chem. Phys., 21, 2067–2082, https://doi.org/10.5194/acp-21-2067-2021, 2021.

Chen, Z., Jacob, D. J., Gautam, R., Omara, M., Stavins, R. N., Stowe, R. C., Nesser, H., Sulprizio, M. P., Lorente, A., Varon, D. J., Lu, X., Shen, L., Qu, Z., Pendergrass, D. C., and Hancock, S.: Satellite quantification of methane emissions and oil–gas methane intensities from individual countries in the Middle East and North Africa: implications for climate action, Atmospheric Chem. Phys., 23, 5945–5967, https://doi.org/10.5194/acp-23-5945-2023, 2023.

Copernicus Sentinel-5P: (processed by ESA), TROPOMI Level 2 Carbon Monoxide total column products (02), https://doi.org/10.5270/S5P-bj3nry0, 2024.

De Mazière, M., Thompson, A. M., Kurylo, M. J., Wild, J. D., Bernhard, G., Blumenstock, T., Braathen, G. O., Hannigan, J. W., Lambert, J.-C., Leblanc, T., McGee, T. J., Nedoluha, G., Petropavlovskikh, I., Seckmeyer, G., Simon, P. C., Steinbrecht, W., and Strahan, S. E.: The Network for the Detection of Atmospheric Composition Change (NDACC): history, status and perspectives, Atmospheric Chem. Phys., 18, 4935–4964, https://doi.org/10.5194/acp-18-4935-2018, 2018.

Eskes, H. J., Velthoven, P. F. J. V., Valks, P. J. M., and Kelder, H. M.: Assimilation of GOME total-ozone satellite observations in a three-dimensional tracer-transport model, Q. J. R. Meteorol. Soc., 129, 1663–1681, https://doi.org/10.1256/qj.02.14, 2003.

Gaubert, B., Edwards, D. P., Anderson, J. L., Arellano, A. F., Barré, J., Buchholz, R. R., Darras, S., Emmons, L. K., Fillmore, D., Granier, C., Hannigan, J. W., Ortega, I., Raeder, K., Soulié, A., Tang, W., Worden, H. M., and Ziskin, D.: Global Scale Inversions from MOPITT CO and MODIS AOD, Remote Sens., 15, 4813, https://doi.org/10.3390/rs15194813, 2023.

Gelaro, R., McCarty, W., Suárez, M. J., Todling, R., Molod, A., Takacs, L., Randles, C. A., Darmenov, A., Bosilovich, M. G., Reichle, R., Wargan, K., Coy, L., Cullather, R., Draper, C., Akella, S., Buchard, V., Conaty, A., Silva, A. M. da, Gu, W., Kim, G.-K., Koster, R., Lucchesi, R., Merkova, D., Nielsen, J. E., Partyka, G., Pawson, S., Putman, W.,

Rienecker, M., Schubert, S. D., Sienkiewicz, M., and Zhao, B.: The Modern-Era Retrospective Analysis for Research and Applications, Version 2 (MERRA-2), J. Clim., 30, 5419–5454, https://doi.org/10.1175/JCLI-D-16-0758.1, 2017.

Hakami, A., Henze, D. K., Seinfeld, J. H., Chai, T., Tang, Y., Carmichael, G. R., and Sandu, A.: Adjoint inverse modeling of black carbon during the Asian Pacific Regional Aerosol Characterization Experiment, J. Geophys. Res. Atmospheres, 110, https://doi.org/10.1029/2004JD005671, 2005.

Hancock, S. E., Jacob, D. J., Chen, Z., Nesser, H., Davitt, A., Varon, D. J., Sulprizio, M. P., Balasus, N., Estrada, L. A., Cazorla, M., Dawidowski, L., Diez, S., East, J. D., Penn, E., Randles, C. A., Worden, J., Aben, I., Parker, R. J., and Maasakkers, J. D.: Satellite quantification of methane emissions from South American countries: a high-resolution inversion of TROPOMI and GOSAT observations, Atmospheric Chem. Phys., 25, 797–817, https://doi.org/10.5194/acp-25-797-2025, 2025.

Hasekamp, O., Lorente, A., Hu, H., and Butz, A.: Algorithm Theoretical Baseline Document for Sentinel-5 Precursor Methane Retrieval, 2022.

Hunt, B. R., Kostelich, E. J., and Szunyogh, I.: Efficient data assimilation for spatiotemporal chaos: A local ensemble transform Kalman filter, Phys. Nonlinear Phenom., 230, 112–126, https://doi.org/10.1016/j.physd.2006.11.008, 2007.

Janjić, T., Bormann, N., Bocquet, M., Carton, J. A., Cohn, S. E., Dance, S. L., Losa, S. N., Nichols, N. K., Potthast, R., Waller, J. A., and Weston, P.: On the representation error in data assimilation, https://doi.org/10.1002/qj.3130, 2018.

Jiang, Z., Jones, D. B. A., Kopacz, M., Liu, J., Henze, D. K., and Heald, C.: Quantifying the impact of model errors on top-down estimates of carbon monoxide emissions using satellite observations, J. Geophys. Res. Atmospheres, 116, https://doi.org/10.1029/2010JD015282, 2011.

Jiang, Z., Worden, J. R., Worden, H., Deeter, M., Jones, D. B. A., Arellano, A. F., and Henze, D. K.: A 15-year record of CO emissions constrained by MOPITT CO observations, Atmospheric Chem. Phys., 17, 4565–4583, https://doi.org/10.5194/acp-17-4565-2017, 2017.

Jones, D. B. A., Bowman, K. W., Logan, J. A., Heald, C. L., Liu, J., Luo, M., Worden, J., and Drummond, J.: The zonal structure of tropical $O_3$ and CO as observed by the Tropospheric Emission Spectrometer in November 2004 – Part 1: Inverse modeling of CO emissions, Atmospheric Chem. Phys., 9, 3547–3562, https://doi.org/10.5194/acp-9-3547-2009, 2009.

Kiel, M., Hase, F., Blumenstock, T., and Kirner, O.: Comparison of XCO abundances from the Total Carbon Column Observing Network and the Network for the Detection of Atmospheric Composition Change measured in Karlsruhe, Atmospheric Meas. Tech., 9, 2223–2239, https://doi.org/10.5194/amt-9-2223-2016, 2016.

Knapp, M., Kleinschek, R., Hase, F., Agustí-Panareda, A., Inness, A., Barré, J., Landgraf, J., Borsdorff, T., Kinne, S., and Butz, A.: Shipborne measurements of $XCO_2$, $XCH_4$, and XCO above the Pacific Ocean and comparison to CAMS atmospheric analyses and S5P/TROPOMI, Earth Syst. Sci. Data, 13, 199–211, https://doi.org/10.5194/essd-13-199-2021, 2021.

Lahoz, W. A. and Schneider, P.: Data assimilation: making sense of Earth Observation, Front. Environ. Sci., 2, https://doi.org/10.3389/fenvs.2014.00016, 2014.

Landgraf, J., aan de Brugh, J., Scheepmaker, R., Borsdorff, T., Hu, H., Houweling, S., Butz, A., Aben, I., and Hasekamp, O.: Carbon monoxide total column retrievals from TROPOMI shortwave infrared measurements, Atmospheric Meas. Tech., 9, 4955–4975, https://doi.org/10.5194/amt-9-4955-2016, 2016.

Lorente, A., Borsdorff, T., Butz, A., Hasekamp, O., aan de Brugh, J., Schneider, A., Wu, L., Hase, F., Kivi, R., Wunch, D., Pollard, D. F., Shiomi, K., Deutscher, N. M., Velazco, V. A., Roehl, C. M., Wennberg, P. O., Warneke, T., and

Landgraf, J.: Methane retrieved from TROPOMI: improvement of the data product and validation of the first 2 years of measurements, Atmospheric Meas. Tech., 14, 665–684, https://doi.org/10.5194/amt-14-665-2021, 2021.

Lu, X., Jacob, D. J., Wang, H., Maasakkers, J. D., Zhang, Y., Scarpelli, T. R., Shen, L., Qu, Z., Sulprizio, M. P., Nesser, H., Bloom, A. A., Ma, S., Worden, J. R., Fan, S., Parker, R. J., Boesch, H., Gautam, R., Gordon, D., Moran, M. D., Reuland, F., Villasana, C. A. O., and Andrews, A.: Methane emissions in the United States, Canada, and Mexico: evaluation of national methane emission inventories and 2010–2017 sectoral trends by inverse analysis of in situ (GLOBALVIEWplus CH$_4$ ObsPack) and satellite (GOSAT) atmospheric observations, Atmospheric Chem. Phys., 22, 395–418, https://doi.org/10.5194/acp-22-395-2022, 2022.

Maasakkers, J. D., Jacob, D. J., Sulprizio, M. P., Scarpelli, T. R., Nesser, H., Sheng, J.-X., Zhang, Y., Hersher, M., Bloom, A. A., Bowman, K. W., Worden, J. R., Janssens-Maenhout, G., and Parker, R. J.: Global distribution of methane emissions, emission trends, and OH concentrations and trends inferred from an inversion of GOSAT satellite data for 2010–2015, Atmospheric Chem. Phys., 19, 7859–7881, https://doi.org/10.5194/acp-19-7859-2019, 2019.

McKain, K., Sweeney, C., Baier, B., Crotwell, A., Crotwell, M., Handley, P., Higgs, J., Legard, T., Madronich, M., Miller, J. B., Moglia, E., Mund, J., Newberger, T., Wolter, S., and NOAA Global Monitoring Laboratory.: NOAA Global Greenhouse Gas Reference Network Flask-Air PFP Sample Measurements of    CO2, CH4, CO, N2O, H2, SF6 and isotopic ratios collected from aircraft vertical   profiles (2024-08-12), https://doi.org/doi.org/10.15138/39HR-9N34, 2024.

Miyazaki, K., Eskes, H. J., Sudo, K., Takigawa, M., van Weele, M., and Boersma, K. F.: Simultaneous assimilation of satellite NO$_2$, O$_3$, CO, and HNO$_3$ data for the analysis of tropospheric chemical composition and emissions, Atmospheric Chem. Phys., 12, 9545–9579, https://doi.org/10.5194/acp-12-9545-2012, 2012.

Miyazaki, K., Eskes, H. J., and Sudo, K.: A tropospheric chemistry reanalysis for the years 2005–2012 based on an assimilation of OMI, MLS, TES, and MOPITT satellite data, Atmospheric Chem. Phys., 15, 8315–8348, https://doi.org/10.5194/acp-15-8315-2015, 2015.

Pendergrass, D. C., Jacob, D. J., Nesser, H., Varon, D. J., Sulprizio, M., Miyazaki, K., and Bowman, K. W.: CHEEREIO 1.0: a versatile and user-friendly ensemble-based chemical data assimilation and emissions inversion platform for the GEOS-Chem chemical transport model, Geosci. Model Dev., 16, 4793–4810, https://doi.org/10.5194/gmd-16-4793-2023, 2023.

Plant, G., Kort, E. A., Murray, L. T., Maasakkers, J. D., and Aben, I.: Evaluating urban methane emissions from space using TROPOMI methane and carbon monoxide observations, Remote Sens. Environ., 268, 112756, https://doi.org/10.1016/j.rse.2021.112756, 2022.

Qu, Z., Jacob, D. J., Shen, L., Lu, X., Zhang, Y., Scarpelli, T. R., Nesser, H., Sulprizio, M. P., Maasakkers, J. D., Bloom, A. A., Worden, J. R., Parker, R. J., and Delgado, A. L.: Global distribution of methane emissions: a comparative inverse analysis of observations from the TROPOMI and GOSAT satellite instruments, Atmospheric Chem. Phys., 21, 14159–14175, https://doi.org/10.5194/acp-21-14159-2021, 2021.

Schneising, O., Buchwitz, M., Reuter, M., Bovensmann, H., and Burrows, J. P.: Severe Californian wildfires in November 2018 observed from space: the carbon monoxide perspective, Atmospheric Chem. Phys., 20, 3317–3332, https://doi.org/10.5194/acp-20-3317-2020, 2020.

Schneising, O., Buchwitz, M., Hachmeister, J., Vanselow, S., Reuter, M., Buschmann, M., Bovensmann, H., and Burrows, J. P.: Advances in retrieving XCH$_4$ and XCO from Sentinel-5 Precursor: improvements in the scientific TROPOMI/WFMD algorithm, Atmospheric Meas. Tech., 16, 669–694, https://doi.org/10.5194/amt-16-669-2023, 2023.

Stanevich, I., Jones, D. B. A., Strong, K., Parker, R. J., Boesch, H., Wunch, D., Notholt, J., Petri, C., Warneke, T., Sussmann, R., Schneider, M., Hase, F., Kivi, R., Deutscher, N. M., Velazco, V. A., Walker, K. A., and Deng, F.: Characterizing model errors in chemical transport modeling of methane: impact of model resolution in versions v9-

02 of GEOS-Chem and v35j of its adjoint model, Geosci. Model Dev., 13, 3839–3862, https://doi.org/10.5194/gmd-13-3839-2020, 2020.

Stanevich, I., Jones, D. B. A., Strong, K., Keller, M., Henze, D. K., Parker, R. J., Boesch, H., Wunch, D., Notholt, J., Petri, C., Warneke, T., Sussmann, R., Schneider, M., Hase, F., Kivi, R., Deutscher, N. M., Velazco, V. A., Walker, K. A., and Deng, F.: Characterizing model errors in chemical transport modeling of methane: using GOSAT XCH$_4$ data with weak-constraint four-dimensional variational data assimilation, Atmospheric Chem. Phys., 21, 9545–9572, https://doi.org/10.5194/acp-21-9545-2021, 2021.

Stephens, B. B., Gurney, K. R., Tans, P. P., Sweeney, C., Peters, W., Bruhwiler, L., Ciais, P., Ramonet, M., Bousquet, P., Nakazawa, T., Aoki, S., Machida, T., Inoue, G., Vinnichenko, N., Lloyd, J., Jordan, A., Heimann, M., Shibistova, O., Langenfelds, R. L., Steele, L. P., Francey, R. J., and Denning, A. S.: Weak Northern and Strong Tropical Land Carbon Uptake from Vertical Profiles of Atmospheric CO2, Science, 316, 1732–1735, https://doi.org/10.1126/science.1137004, 2007.

Sweeney, C., Karion, A., Wolter, S., Newberger, T., Guenther, D., Higgs, J. A., Andrews, A. E., Lang, P. M., Neff, D., Dlugokencky, E., Miller, J. B., Montzka, S. A., Miller, B. R., Masarie, K. A., Biraud, S. C., Novelli, P. C., Crotwell, M., Crotwell, A. M., Thoning, K., and Tans, P. P.: Seasonal climatology of CO2 across North America from aircraft measurements in the NOAA/ESRL Global Greenhouse Gas Reference Network, J. Geophys. Res. Atmospheres, 120, 5155–5190, https://doi.org/10.1002/2014JD022591, 2015.

Varon, D. J., Jacob, D. J., Sulprizio, M., Estrada, L. A., Downs, W. B., Shen, L., Hancock, S. E., Nesser, H., Qu, Z., Penn, E., Chen, Z., Lu, X., Lorente, A., Tewari, A., and Randles, C. A.: Integrated Methane Inversion (IMI 1.0): a user-friendly, cloud-based facility for inferring high-resolution methane emissions from TROPOMI satellite observations, Geosci. Model Dev., 15, 5787–5805, https://doi.org/10.5194/gmd-15-5787-2022, 2022.

Voshtani, S., Ménard, R., Walker, T. W., and Hakami, A.: Use of Assimilation Analysis in 4D-Var Source Inversion: Observing System Simulation Experiments (OSSEs) with GOSAT Methane and Hemispheric CMAQ, Atmosphere, 14, 758, https://doi.org/10.3390/atmos14040758, 2023.

Wunch, D., Wennberg, P. O., Toon, G. C., Keppel-Aleks, G., and Yavin, Y. G.: Emissions of greenhouse gases from a North American megacity, Geophys. Res. Lett., 36, https://doi.org/10.1029/2009GL039825, 2009.

Wunch, D., Toon, G. C., Wennberg, P. O., Wofsy, S. C., Stephens, B. B., Fischer, M. L., Uchino, O., Abshire, J. B., Bernath, P., Biraud, S. C., Blavier, J.-F. L., Boone, C., Bowman, K. P., Browell, E. V., Campos, T., Connor, B. J., Daube, B. C., Deutscher, N. M., Diao, M., Elkins, J. W., Gerbig, C., Gottlieb, E., Griffith, D. W. T., Hurst, D. F., Jiménez, R., Keppel-Aleks, G., Kort, E. A., Macatangay, R., Machida, T., Matsueda, H., Moore, F., Morino, I., Park, S., Robinson, J., Roehl, C. M., Sawa, Y., Sherlock, V., Sweeney, C., Tanaka, T., and Zondlo, M. A.: Calibration of the Total Carbon Column Observing Network using aircraft profile data, Atmospheric Meas. Tech., 3, 1351–1362, https://doi.org/10.5194/amt-3-1351-2010, 2010.

Wunch, D., Toon, G. C., Blavier, J.-F. L., Washenfelder, R. A., Notholt, J., Connor, B. J., Griffith, D. W. T., Sherlock, V., and Wennberg, P. O.: The Total Carbon Column Observing Network, Philos. Trans. R. Soc. Math. Phys. Eng. Sci., 369, 2087–2112, https://doi.org/10.1098/rsta.2010.0240, 2011.

Wunch, D., Jones, D. B. A., Toon, G. C., Deutscher, N. M., Hase, F., Notholt, J., Sussmann, R., Warneke, T., Kuenen, J., Denier van der Gon, H., Fisher, J. A., and Maasakkers, J. D.: Emissions of methane in Europe inferred by total column measurements, Atmospheric Chem. Phys., 19, 3963–3980, https://doi.org/10.5194/acp-19-3963-2019, 2019.

Yang, Z., Washenfelder, R. A., Keppel-Aleks, G., Krakauer, N. Y., Randerson, J. T., Tans, P. P., Sweeney, C., and Wennberg, P. O.: New constraints on Northern Hemisphere growing season net flux, Geophys. Res. Lett., 34, https://doi.org/10.1029/2007GL029742, 2007.

Zhou, M., Langerock, B., Vigouroux, C., Sha, M. K., Hermans, C., Metzger, J.-M., Chen, H., Ramonet, M., Kivi, R., Heikkinen, P., Smale, D., Pollard, D. F., Jones, N., Velazco, V. A., García, O. E., Schneider, M., Palm, M., Warneke,

T., and De Mazière, M.: TCCON and NDACC $X_{CO}$ measurements: difference, discussion and application, Atmospheric Meas. Tech., 12, 5979–5995, https://doi.org/10.5194/amt-12-5979-2019, 2019.

---

## Author Comment (AC2)

**Response to reviewers (2nd round of revisions)**

**Quantifying CO emissions from boreal wildfires by assimilating TROPOMI and TCCON observations**

Our responses to the referee comments and updates in the revised manuscript are listed below. Referee comments are in black and author responses are in blue. The line numbers in our responses refer to the 2nd revised manuscript.
* * *
**Referee 3 (RC3):**

The revised manuscript was significantly improved by the authors. All comments were sufficiently replied to and the manuscript was modified accordingly. I'm happy to accept the manuscript for publications with just a few very minor suggestions with all except one being purely technical:

minor:
- l.350-355: Thank you for adding additional information on the ensemble generation. For me it is not entirely clear if the perturbations of each member are sampled uniformly (i.e. one emission perturbation factor for each member applied globally throughout the whole time period) or vary in space and time (if so, are they spatio-temporally correlated?)? Please describe.

Thank you for your question and the opportunity to clarify. The perturbation of the emission scaling factors for ensemble generation is performed within the CHEEREIO framework. In our setup, the scaling factors are initialized randomly at the grid scale for each ensemble member, meaning that perturbations vary spatially rather than being applied uniformly or globally. These perturbations are sampled from a multivariate lognormal distribution, with spatial correlations imposed using an exponential decay function with a correlation distance of 500 km. We assume no explicit temporal correlations. Therefore, the spatially varying and correlated scaling factors are initialized at the beginning of the assimilation window for each ensemble member and are subsequently updated by observation increments at every assimilation cycle (3-day).
Further details on ensemble generation can be found in Section 3.2 of Pendergrass et al. (2023), and a description of the scaling factor perturbations is provided in the CHEEREIO documentation:
https://cheereio.readthedocs.io/en/latest/The-ensemble-configuration-file.html?highlight=perturb#scaling-factor-settings.

**Text revised and added in Section 3**

(lines 326-329):
"To obtain a posteriori estimate of CO emissions, we begin by initializing the ensemble scaling factors using a multiplicative random perturbation to the prior estimates from all emission sources at the grid scale (i.e., spatially varying perturbations), sampled from a multivariate lognormal distribution (see the detailed description of ensemble generation in Pendergrass et al. (2023), Section 3.2)."

(lines 348-354):
"Then, an ensemble spin-up without assimilating observations is performed, where emissions are randomly perturbed at each grid point based on a lognormal distribution to create an ensemble spread. We assume a lognormal standard deviation of $\sigma = 0.2$, centered around 1, that provides ensemble member scaling factors between 0.55 and 1.82 with 99% confidence. This perturbation level is sufficient to generate meaningful ensemble spread while avoiding unrealistically high or low values for constructing the prior error covariance. The scaling factors are assumed to be spatially correlated using an exponential decay function with a correlation distance of 500 km, while no explicit temporal correlations are imposed."

technical:
- l.145: Does "same TCCON data" refer to the assimilated data? I would suggest replacing "same" with "assimilated"
Yes, "same TCCON data" refers to the assimilated data. We have revised the text as suggested in Section 2 (line 146).

- l.154: "a local overpass solar times" is there a word missing? Or inconsistency between singular ("a") and plural ("times")

Corrected. We have revised the text in Section 2.1 (line 154): "a local overpass solar time of 13:30 UTC"

- l.330ff and 355ff: Due to the changes in l.330ff, this sentence is now doubled with almost identical content. Please remove one of them.

Thanks for pointing this out. We have removed the second occurrence of similar content regarding the use of the lognormal distribution in Section 3 (line 356): "The choice of a lognormal distribution not only maintains the positivity of the solution (i.e., prevent an unrealistic negative scaling factors), but also better represents the uncertainty of CO emission estimates from inventories (Maasakkers et al., 2019; Plant et al., 2022)."
* * *
**Additional minor updates:**

1. Updated Fig. 4c and 4i to remove a duplicate label in the subfigures.
2. Line 108: replaced ", and therefore" with "; therefore,".
3. Line 116: replaced "The TCCON" with "TCCON".
4. Line 149: replaced "inversion's setup" with "inversion setup".
5. Line 167: replaced "within" with "in".
6. Lines 179-180: replaced "reports" with "report shows that".
7. Line 180: added "data" after "XCO".
8. Figure 1 caption: replaced "The number" with "Number"; "quality controlled" with "quality-controlled"; "The variability" with "Variability"; "Temporal series" with "Time series"; and removed repeated "East Trout Lake" and repeated "column averaging kernels".
9. Line 222: removed "(see Table 2)".
10. Line 288: replaced "vegetation-type" with "vegetation type".
11. Line 289: replaced "Global" with "the Global".
12. Line 300: replaced "averaged" with "average".
13. Line 306: replaced "has" with "have".
14. Line 303: replaced "satellite driven" with "satellite-driven".
15. Line 308: replaced "global gridded" with "globally gridded".
16. Line 312: added "the" before "background".
17. Line 315: replaced "detail" with "detailed".
18. Line 324: replaced "location and time" with "locations and times".
19. Line 335: replaced "by" with "of".
20. Line 336: replaced "LETKF" with "the LETKF".
21. Line 339: removed "the" before "BB".
22. Line 351: replaced "between 0.55 to 1.82" with "between 0.55 and 1.82".
23. Line 356: replaced "perturbation of emissions" with "perturbations in emissions".
24. Line 367: replaced "$\Delta = 0.08$" with "($\Delta = 0.08$)"; and "quick" with "rapid".
25. Line 368: replaced "which may result in an inversion not being updated" with "which may otherwise prevent the inversion from being updated".
26. Line 370: replaced "impacted" with "affected".
27. Line 403: replaced "into" with "in".
28. Line 449: added "the" before "joint inversion".
29. Line 450: replaced "assimilation" with "inversion".
30. Line 452: added "could" before "benefit".

31. Line 454: replaced "emission" with "emissions"; and "between all the inventories" with "across all inventories".
32. Line 458: removed "of the".
33. Line 459: replaced "increases to" with "reaches".
34. Line 460: removed 'of uncertainty".
35. Line 502: replaced "Fig. 3d,i,n" with "Fig. 3b,g,l,d,i,n".
36. Figure 6 caption: replaced "inversion (red)" with "(red) inversions"
37. Line 573: replaced "the vector of dimension, m" with "a vector of dimension m"; and added "the" before "number".
38. Line 584: replaced "inversion" with "inversions".
39. Line 596: replaced "inversion" with "inversions".
40. Line 598: replaced "improves" with "improve".
41. Line 600: replaced "assimilation" with "assimilations".
42. Line 601: replaced "timeseries with "time series"; and "shows" with "show".
43. Line 605: replaced "show" with "shows".
44. Figure 7 caption: replaced "of all sites" with "for all sites"; "a timeseries and scatter plot" with "time series and scatter plots"; "however" with "but"; and "estimate" with "estimates".
45. Figure 8 caption: replaced "from bottom-up inventories" with "inventory"; added "the" before "prior" and "posterior".
46. Line 628: added "a" before "further".
47. Line 638: added "the" before "model"; and replaced "are" with "is".
48. Line 641: added "estimates" after "a posteriori"; and replaced "we see" with "observed".
49. Line 637 and 644: replaced "2°×2.5°" with "2° × 2.5°".
50. Line 644: replaced "for" with "at the".
51. Line 649: replaced "most of the cases" with "most cases".
52. Line 650: replaced "to the bias of the" with "to that of the".
53. Line 651: replaced "an equivalent … estimate" with "equivalent … estimates".
54. Line 658: added "the" before "a posteriori"; removed "time" before "period".
55. Figure 9 caption: added "the" before "a priori" and "a posteriori"; replaced "between" with "for"; replaced "of" with "between"; and added "for" before "comparison".
56. Line 672: replaced "assimilation" with "estimate"; and replaced "with" with "for".
57. Line 676: added "estimates" after "XCO"; removed "i.e.," in the parentheses.
58. Lines 676-677: shifted "than those with QFED and GBBEPx" to the end of the sentence.
59. Line 677: replaced "mode estimated" with "model-estimated"; replaced "has" with "shows".
60. Line 680: replaced "while" with "whereas".
61. Line 693: removed "of extreme wildfires".
62. Line 703: replaced "CFEEPS" with "CFFEPS".
63. Figure 10 caption: removed "the" before "prior"; added "profiles" after "a priori".
64. Figure 11 caption: replaced "preformed" with "performed"; added "TCCON" before "XCO"; added "data" after "TROPOMI-only" and replaced "a joint TROPOMI and TCCON" with "joint TROPOMI and TCCON data".
65. Lines 723-724: added "estimates" after "a priori"; removed "have".
66. Line 727: replaced "a closer estimate" with "closer estimates".
67. Line 731: replaced "with matching variability with the measurements" with "with variability matching the measurements".
68. Line 739: replaced "grided" with "gridded".
69. Line 748: replaced "Colombia" with "Columbia".
70. Figure 12 caption: added "the" before "CFFEPS"; replaced "TROPOMI and TCCON standard XCO TCCON" with "TROPOMI and standard TCCON"; and removed "XCO" before "TCCON".

71. Lines 758-760: removed "in" before the color names in parentheses.
72. Line 761: removed "the" before "inversion cases".
73. Figure 13 caption: replaced "Evaluating" with "Evaluation of"; replaced "of" with "from"; added "the" before "a priori".
74. Line 773: replaced "due" with "due to".
75. Lines 791-792: replaced "plume" and "altitude" with "plumes" and "altitudes".
76. Line 805: replaced "column averaged" with "column-averaged".
77. Line 818: removed "between" after "by".
78. Lines 826: replaced "deference" and "standard deviation" with "differences" and "standard deviations"; added "the" before "assimilation".
79. 847-848: replaced "inversions" with "inversion"; and "improve" with "improves".
80. Lines 860: replaced "an idealized condition" with "idealized conditions".
81. Figure A1 caption: replaced "true − prior" with "prior − true"; replaced the second occurrence of "(c)" with "(d)".
82. Line 876: replaced "and" with "under".
83. Line 878: added "the" before "assimilation parameter".
84. Lines 883: replaced "posterior" with "posteriors".
85. Line 894: added "only" before "TCCON".
86. Table B1 caption and footnotes: added "the" before "true emissions"; replaced "priror" with "prior"; and replaced "is" with "are".
87. Line 940: replaced "is" with "are".
88. Line 952: replaced "that" with ", which".
89. Line 956: replaced "QFEDv2.5r1" with "QFED v2.5r1".